

# On higher-dimensional Carrollian and Galilean conformal field theories

**Bin Chen[1,2,3], Reiko Liu[1] and Yu-fan Zheng[1]**

**1** School of Physics, Peking University, No.5 Yiheyuan Rd,
Beijing 100871, P. R. China
**2** Collaborative Innovation Center of Quantum Matter,
No.5 Yiheyuan Rd, Beijing 100871, P. R. China
**3** Center for High Energy Physics, Peking University,
No.5 Yiheyuan Rd, Beijing 100871, P. R. China

## Abstract

In this paper, we study the Carrollian and Galilean conformal field theories (CCFT and GCFT) in $d > 2$ dimensions. We construct the highest weight representations (HWR) of Carrollian and Galilean conformal algebra (CCA and GCA). Even though the two algebras have different structures, their HWRs share similar structure, because their rotation subalgebras are isomorphic. In both cases, we find that the finite dimensional representations are generally reducible but indecomposable, and can be organized into the multiplets. Moreover, it turns out that the multiplet representations in $d > 2$ CCA and GCA carry not only the simple chain structure appeared in logCFT or $2d$ GCFT, but also more generally the net structures. We manage to classify all the allowed chain representations. Furthermore we discuss the two-point and three-point correlators by using the Ward identities. We mainly focus on the two-point correlators of the operators in chain representations. Even in this relative simple case, we find some novel features: multiple-level structure, shortage of the selection rule on the representations, undetermined 2-pt coefficients, etc.. We find that the non-trivial correlators could only appear for the representations of certain structure, and the correlators are generally polynomials of time coordinates for CCFT (spacial coordinates for GCFT), whose orders depend on the levels of the correlators.


doi:10.21468/SciPostPhys.14.5.088

# 1 Introduction

Conformal bootstrap provides a nonperturbative way to read the spectrum and the OPE coefficients of the theory by imposing conformal symmetry, unitarity and crossing symmetry. It was firstly proposed in 1970s [1,2], and it had been successfully applied to the study of 2D conformal field theory(CFT), in particular 2D minimal models in 1980s [3]. It is experiencing a renaissance, starting from the seminal work of [4]. Various numerical techniques and analytical methods have been developed. For nice reviews, see [5,6]. The conformal bootstrap not only allows us to study the properties of various models with conformal symmetry, for example the critical 3d Ising model [7,8], but also sheds light on the study of AdS/CFT correspondence [9–12] and S-matrix bootstrap.

Conformal bootstrap relies very much on the conformal symmetry. It would be interesting to investigate the bootstrap program if the conformal symmetry is replaced by other conformal-like symmetries. These symmetries often partially break the Lorentz symmetry but keep some kind of scale invariance. They include Schrödinger symmetry, Lifshitz symmetry, Galilean conformal symmetry and Carrollian conformal symmetry etc. In two dimensions, the global symmetries can be enhanced to local ones under suitable conditions, and the resulting algebras lead to more restrictions on the dynamics [13–16]. There have been some efforts in developing bootstrap program for conformal-like symmetries. In [17–22], conformal bootstrap has been studied in the theories with Schrödinger conformal symmetry. In [23, 24], the Bondi-Metzner-Sachs(BMS) bootstrap was studied. In [25–27], two-dimensional Galilean conformal bootstrap has been initiated.

In this work, we would like to study Carrollian conformal field theory (CCFT) and Galilean conformal field theory (GCFT) in dimension $d > 2$. The Carrollian conformal symmetry can be obtained by taking the ultra-relativistic limit of conformal symmetry. The Carrollian symmetry was first introduced by Levy-Leblond in 1965 [28, 29]. In the ultra-relativistic limit, the speed of light turns to zero, and the lightcones close up. It has been discussed in various physical systems, see [30–43]. Very recently, the Carrollian symmetry appears in the study of fractons [44–48]. Remarkably, the infinitely extended Carrollian conformal algebras in $d = 2, 3$ are isomorphic to the algebra $\text{BMS}_{d+1}$, which generate the asymptotic symmetry groups in flat space-time [32]. On the other hand, the Galilean conformal symmetry can be read by taking the non-relativistic, i.e. the speed of light $c \to \infty$ limit, of the conformal group. Its physical implications have been widely studied, see for example [49–51]. In particular, it turns out that the two-dimensional Galilean conformal symmetry is closely related to the flat holography, as the $\text{GCA}_2$ is isomorphic to $\text{BMS}_3$ [52]. It is also worth mentioning that there exist BMS-like extension of Carrollian algebra, as the asymptotic symmetry of four-dimensional Carrollian gravities [43, 53].

The Carrollian conformal symmetry is generated Carrollian conformal algebra (CCA), with the generators being $\{D, P^\mu, K^\mu, B^i, J^{ij}\}$ $i, j = 1, \ldots, d-1$, $\mu = 0, 1, \ldots, d-1$, where $B^i$ are Carrollian boost operators. The commutation relations can be found in (2.1). The highest weight representations of CCA, analogous to the ones in conformal field theory (CFT), are the eigenstates of the dilation generator $D$, and are annihilated by the generators $K^\mu$. This means that although the Lie algebra $CCA_d \cong \mathfrak{iso}(d, 1)$, its highest weight representations are not constructed in the same way as the famous Wigner's classification.[1] One major difficulty is to construct the representations for the "CCA rotation", which includes the spatial rotations $\{J^{ij}\}$ and the CCA boosts $\{B^i\}$. Different from the $SO(d)$ group and its algebra, the CCA rotation is not semi-simple, thus the finite dimensional representations are generally reducible but indecomposable, and can be organized as multiplet representations. The multiplet representations for $d > 2$ CCA have much complicated structures. They are made up of non-trivial $SO(d-1)$ representations joined by the action of Carrollian boosts, and this generally leads to net representations rather than just chain-like ones in LogCFT or $2d$ GCFT. The general structure of net representations is beyond the scope of this work. We manage to work out all the allowed chain representations, and find that they can be categorized into a few classes: the decreasing chain, the increasing chain and two exceptional cases, besides rank-1 singlets. With the representations of the CCA rotation being well defined, the highest weight representations and the definitions of local operators follow immediately, which complete the discussions of local operators in CCFT.

In contrast, the Galilean conformal symmetry can be obtained by taking non-relativistic limit, and is also generated by $\{D, P^\mu, K^\mu, B^i, J^{ij}\}$ but with $B^i$ being the Galilean boost oper-

---

[1]In this work, we focus on the highest weight representation. The representations from Wigner's classification may be useful for other purposes.

ators. The Galilean conformal algebra (GCA) has a different algebraic structure (2.3) from CCA, and the symmetry generators of GCA act differently as the symmetries on the space-time. Nevertheless, the highest weight representations we need have similar structures with the ones in CCA. The reason is that the GCA rotation $\{J^{ij}, B^i\}$ shares the same algebraic structure with the CCA rotation, thus the finite dimensional representations of GCA rotation and further the highest weight representations and local operators have similar structures with the ones in CCA. The essential difference originates from the action of the $B^i$ generators. Consequently the covariant tensor representations of the CCA rotation become the contravariant tensor representations of the GCA rotation.

Furthermore, we discuss the two-point and three-point functions in CCFT and GCFT. In principle, these correlators can be constrained by the Ward identities, just as in CFT. In this work, we mainly focus on the correlators of chain representations in CCFT$_4$. Even in this relatively simple case, there present some novel features. First of all, as the time coordinate and spacial coordinates behave differently under symmetry transformation, the time-dependence of the correlators is absent in many cases. We find that the non-trivial correlators with time dependence could only appear for the representations of certain structure, and the correlators are generally polynomials of time coordinates for CCFT (spacial coordinates for GCFT). Secondly, due to the multiplet structure, the 2-pt correlators present multi-level structures. At each level, there could be 2-pt coefficients, which generally cannot be fixed by the Ward identities. Moreover, as the representations are reducible, Schur lemma cannot be applied such that the selection rules on the representations are absent. Consequently the 2-pt correlators of the operators in different representations in CCFT could be nonvanishing. For the 2-pt correlators of net representations and the 3-pt correlators of the operators in chain representation, the discussion by using the Ward identities is similar, but becomes more tedious. In this work, we only briefly discuss these cases.

The rest of the paper is organized as follows: in section 2, we review the Carrollian and Galilean conformal symmetry; in section 3 we construct the highest weight representations of CCA and GCA; and further in section 4 we calculate the 2-pt and 3-pt correlation functions in 4d CCFT; in 5 we briefly discuss the correlators in 4d GCFT; finally in section 6 we conclude this paper with some discussions. In a few appendices, we not only present some technical details in calculations, and some discussions on 3d CCFT, but also show how the Carrollian conformal algebra can be induced on the null hypersurface from a Lorentzian CFT.

## 2 Carrollian and Galilean conformal symmetries

In this section, we introduce the Carrollian and Galilean conformal symmetries. They can be obtained by taking special limits of the relativistic conformal symmetry [54]: the Carrollian case corresponds to the ultra-relativistic limit $c \to 0$, while the Galilean one corresponds to the non-relativistic limit $c \to \infty$. Another intrinsic way of deriving them is to consider the conformal transformations in the framework of Carrollian geometry and Newton-Cartan geometry [30, 31]. Let us discuss them case by case.

### 2.1 Carrollian conformal symmetry

One can obtain the Carrollian Conformal Algebra (CCA) in $d$ dimensions by taking the ultra-relativistic limit $c \to 0$ from the usual $d$-dimensional conformal algebra [54]. The generators are labeled by $\{D, P^\mu, K^\mu, B^i, J^{ij}\}$ with $\mu = 0, 1, \ldots, d-1$, $i, j = 1, \ldots, d-1$, where the Carrollian boost generators $B^i$ come from the rotation generators: $J^{i0} \xrightarrow{c \to 0} B^i$. The commutation

relations are

$$[D, P^\mu] = P^\mu, \quad [D, K^\mu] = -K^\mu, \quad [D, B^i] = [D, J^{ij}] = 0,$$
$$[J^{ij}, G^k] = \delta^{ik} G^j - \delta^{jk} G^i, \quad G \in \{P, K, B\}$$
$$[J^{ij}, P^0] = [J^{ij}, K^0] = 0,$$
$$[J^{ij}, J^{kl}] = \delta^{ik} J^{jl} - \delta^{il} J^{jk} + \delta^{jl} J^{ik} - \delta^{jk} J^{il},$$
$$[B^i, P^j] = \delta^{ij} P^0, \quad [B^i, K^j] = \delta^{ij} K^0, \quad [B^i, B^j] = [B^i, P^0] = [B^i, K^0] = 0,$$
$$[K^0, P^0] = 0, \quad [K^0, P^i] = -2B^i, \quad [K^i, P^0] = 2B^i, \quad [K^i, P^j] = 2\delta^{ij} D + 2J^{ij}. \tag{2.1}$$

We may re-list the commutators of $K$ and $P$ in the following matrix:

$$[K^\mu, P^\nu] = \begin{pmatrix} 0 & -2B^1 & -2B^2 & -2B^3 & \dots \\ 2B^1 & 2D & 2J^{12} & 2J^{13} & \dots \\ 2B^2 & 2J^{21} & 2D & 2J^{23} & \dots \\ 2B^3 & 2J^{31} & 2J^{32} & 2D & \dots \\ \vdots & \vdots & \vdots & \vdots & \ddots \end{pmatrix}. \tag{2.2}$$

The actions of these generators as the symmetries of space-time are listed in Table 1. Notice that the commutators of the symmetry charges differ from the ones of the vector fields by a minus sign due to the definition of symmetry generators $Q_\epsilon = -\int dS_\mu \epsilon_\nu T^{\mu\nu}$.

Table 1: Action of CCA symmetry generators as the vector fields on the space-time.

| generator | vector field | finite transformation |
|---|---|---|
| $d$ | $t\partial_t + x^i \partial^i$ | $\lambda x^\mu$ |
| $p^\mu$ | $(\partial_t, \vec{\partial})$ | $x^\mu + a^\mu$ |
| $k^\mu$ | $(-\vec{x}^2 \partial_t, 2\vec{x} x^\mu \partial^\mu - \vec{x}^2 \vec{\partial})$ | $\left( \frac{t - a^0 \vec{x}^2}{1 - 2\vec{a}\cdot\vec{x} + \vec{a}^2 \vec{x}^2}, \frac{\vec{x} - \vec{a}\vec{x}^2}{1 - 2\vec{a}\cdot\vec{x} + \vec{a}^2 \vec{x}^2} \right)$ |
| $b^i$ | $\vec{x}\partial_t$ | $(t + \vec{v}\cdot\vec{x}, \vec{x})$ |
| $m^{ij}$ | $x^i \partial^j - x^j \partial^i$ | $(t, \mathbf{M}\cdot\vec{x})$ |

Another interesting thing is that the CCA is isomorphic to the Poincare algebra, $\mathfrak{cca}_d \simeq \mathfrak{ca}_{d-1} \ltimes \mathbb{R}^{d+1} \simeq \mathfrak{iso}(d, 1)$, where $\mathfrak{ca}_{d-1} \simeq \mathfrak{so}(d, 1)$ is the $(d-1)$-dimensional Euclidean conformal algebra. One can reorganize the generators to see this relation

$$\tilde{P}^{-2} \equiv \frac{1}{2}(K^0 - P^0), \qquad \tilde{J}^{-2,i} \equiv \frac{1}{2}(K^i - P^i),$$
$$\tilde{P}^{-1} \equiv -\frac{1}{2}(K^0 + P^0), \qquad \tilde{J}^{-1,i} \equiv \frac{1}{2}(K^i + P^i),$$
$$\tilde{P}^i \equiv -B^i, \qquad \tilde{J}^{-2,-1} \equiv D, \quad \tilde{J}^{ij} \equiv -J^{ij}. \tag{2.3}$$

This reorganization keeps the anti-symmetric relation $\tilde{J}^{ab} = -\tilde{J}^{ba}$, where $a, b = -2, -1, 1, \dots, d-1$ (here we skipped label 0 to avoid miss-leading). The commutation relations are then

$$[\tilde{J}^{ab}, \tilde{J}^{cd}] = \eta^{ac} \tilde{J}^{bd} - \eta^{ad} \tilde{J}^{bc} + \eta^{bd} \tilde{J}^{ac} - \eta^{bc} \tilde{J}^{ad},$$
$$[\tilde{J}^{ab}, \tilde{P}^c] = \eta^{ac} \tilde{P}^b - \eta^{bc} \tilde{P}^a, \qquad \text{with } \eta = \mathrm{diag}\{\overset{-2}{-1}, \overset{-1}{1}, \overset{1}{1}, \dots, \overset{d-1}{1}\}, \tag{2.4}$$

which is exactly the Poincaré algebra. Thus one can easily find the Casimir operators of the algebra. Let us consider for example the $d = 4$ case. From the algebra $\mathfrak{iso}(4,1)$, we know that there are three independent Casimir operators $C_2, C_3', C_4$, which have the following forms respectively

$$
\begin{aligned}
C_2 =\,& P^0 K^0 + B^i B^i \,, \\
C_3' =\,& \epsilon_{ijk}(2-D)J^{ij}B^k + \frac{1}{2}\epsilon_{ijk}P^0 J^{ij}K^k - \frac{1}{2}\epsilon_{ijk}P^i J^{jk}K^0 + \epsilon_{ijk}P^i B^j K^k \,, \\
C_4 =\,& 4D(D-4)B^i B^i - \epsilon_{ijk}\epsilon_{mnl}J^{ij}J^{mn}B^k B^l \\
& + 6P^0(D-4)K^0 + P^0 P^0 K^i K^i + P^i P^i K^0 K^0 - 2P^0 P^i K^i K^0 \\
& + 2P^0(3-2D)B^i K^i + 4P^i(2D-7)B^i K^0 + 4P^i B^i B^j K^j - 4P^i B^j B^j K^i \\
& + 4P^0 J^{ij}B^i K^j - 4P^i J^{ij}B^j K^0 + 2P^0 J^{ij}J^{ji}K^0 \,.
\end{aligned}
\tag{2.5}
$$

We can obtain these Casimir operators by taking ultra-relativistic limit from the ones of $\mathfrak{ca}_4$. Since $\mathfrak{ca}_4 \simeq \mathfrak{so}(d,2)$, its Casimir operators $\tilde{C}_2, \tilde{C}_3'$ and $\tilde{C}_4$ are given by standard formalism. The two sets of Casimir operators are related as

$$
\tilde{C}_2 \xrightarrow{c\to 0} C_2 \,, \qquad \tilde{C}_3' \xrightarrow{c\to 0} C_3' \,, \qquad \frac{1}{2}(\tilde{C}_4 - \tilde{C}_2^2) \xrightarrow{c\to 0} C_4 \,.
\tag{2.6}
$$

It is also remarkable that there exists an infinite extension of $d$-dimensional CCA which is isomorphic to BMS$_{d+1}$ algebra [54]. For $d = 3$, the extended algebra is $\mathfrak{bms}_4 = \{L_n, \bar{L}_n, M_{rs}\}$ with the commutation relations

$$
\begin{aligned}
[L_n, L_m] &= (n-m)L_{n+m}\,, & [\bar{L}_n, \bar{L}_m] &= (n-m)\bar{L}_{n+m}\,, \\
[L_m, M_{r,s}] &= \left(\frac{m+1}{2} - r\right)M_{r+m,s}\,, & [\bar{L}_m, M_{r,s}] &= \left(\frac{m+1}{2} - s\right)M_{r,s+m}\,, \\
[M_{r,s}, M_{t,u}] &= 0\,, & m,n,r,s,t,u &\in \mathbb{Z}\,.
\end{aligned}
\tag{2.7}
$$

The algebra $\mathfrak{cca}_3$ can be identified to the global part of $\mathfrak{bms}_4$

$$
\begin{aligned}
D &= L_0 + \bar{L}_0\,, & J^{12} &= -i(L_0 - \bar{L}_0)\,, \\
B^1 &= -\frac{1}{2}(M_{0,1} + M_{1,0})\,, & B^2 &= \frac{i}{2}(M_{0,1} - M_{1,0})\,, \\
P^0 &= -M_{0,0}\,, & P^1 = L_{-1} + \bar{L}_{-1}\,, & P^2 &= -i(L_{-1} - \bar{L}_{-1})\,, \\
K^0 &= M_{1,1}\,, & K^1 = L_1 + \bar{L}_1\,, & K^2 &= i(L_{-1} - \bar{L}_{-1})\,.
\end{aligned}
\tag{2.8}
$$

The corresponding vector fields on the space-time are[2]

$$
l_n = z^{n+1}\partial_z + \frac{n+1}{2}z^n t\partial_t \,, \quad \bar{l}_n = \bar{z}^{n+1}\partial_{\bar{z}} + \frac{n+1}{2}\bar{z}^n t\partial_t \,, \quad m_{r,s} = -z^r \bar{z}^s \partial_t \,, \quad n,r,s \in \mathbb{Z}, \tag{2.9}
$$

where $z = x^1 - ix^2, \bar{z} = x^1 + ix^2$ are the complex coordinates of the celestial sphere.

For $d \geq 4$, the infinitely extended algebra is $\mathfrak{bms}_{d+1} = \{D, P^i, K^i, J^{ij}, M^{\vec{m}}\}$, where $M^{\vec{m}}$ are infinite generators with $\vec{m} = (m_1, \ldots, m_{d-1}), m_i \in \mathbb{Z}$. The $M^{\vec{m}}$ generators are commuting with each other, $[M^{\vec{m}_1}, M^{\vec{m}_2}] = 0$, and the rest parts make up a $(d-1)$-dimensional conformal

---

[2]Notice once again the minus sign caused by the definition $Q_\epsilon = -\int dS_\mu \epsilon_\nu T^{\mu\nu}$.

algebra $\mathfrak{ca}_{d-1} = \{D, P^i, K^i, J^{ij}\}$. The commutation relations of $\mathfrak{ca}_{d-1}$ with $M$'s are

$$
\begin{aligned}
[D, M^{\vec{m}}] &= \left(1 - \sum_i m_i\right) M^{\vec{m}}, \\
[P^i, M^{\vec{m}}] &= -m_i M^{\vec{m}-\vec{e}_i}, \\
[K^i, M^{\vec{m}}] &= \left(2 + m_i - \sum_j 2m_j\right) M^{\vec{m}+\vec{e}_i} + \left(\sum_{j \neq i} m_j\right) M^{\vec{m}-\vec{e}_i+2\vec{e}_i}, \\
[J^{ij}, M^{\vec{m}}] &= m_i M^{\vec{m}-\vec{e}_i+\vec{e}_j} - m_j M^{\vec{m}+\vec{e}_i-\vec{e}_j}.
\end{aligned}
\tag{2.10}
$$

The identifications of the generators of $\mathfrak{cca}_d$ with the ones of the global part of $\mathfrak{bms}_{d+1}$ are

$$
P^0 = M^{\vec{0}}, \quad B^i = M^{\vec{e}_i}, \quad K^0 = -\sum_i M^{2\vec{e}_i}, \quad \vec{e}_i = (0, \ldots, \overset{i}{1}, \ldots, 0), \tag{2.11}
$$

plus the obvious ones $D, P^i, K^i, J^{ij}$. And the vector fields corresponding to $M^{\vec{m}}$ act on the space-time as

$$
m^{\vec{m}} \equiv \left(\prod_{i=1}^{d-1}(x^i)^{m_i}\right) \partial_t, \quad \vec{m} = (m_1, \ldots, m_{d-1}), \quad m_i \in \mathbb{Z}. \tag{2.12}
$$

## 2.2 Galilean conformal symmetry

To obtain the Galilean Conformal Algebra (GCA), one takes the non-relativistic limit $c \to \infty$ [55]. The generators of the algebra can also be denoted by $\{D, P^\mu, K^\mu, B^i, J^{ij}\}$ with $i, j = 1, \ldots, d-1$, $\mu = 0, 1, \ldots, d-1$. They obey different commutation relations, especially for $B$'s,

$$
\begin{aligned}
&[D, P^\mu] = P^\mu, \quad [D, K^\mu] = -K^\mu, \quad [D, B^i] = [D, J^{ij}] = 0, \\
&[J^{ij}, G^k] = \delta^{ik} G^j - \delta^{jk} G^i, \quad G \in \{P, K, B\}, \\
&[J^{ij}, P^0] = [J^{ij}, K^0] = 0, \\
&[J^{ij}, J^{kl}] = \delta^{ik} J^{jl} - \delta^{il} J^{jk} + \delta^{jl} J^{ik} - \delta^{jk} J^{il}, \\
&\mathbf{[B^i, P^0] = -P^i}, \quad \mathbf{[B^i, K^0] = -K^i}, \quad [B^i, B^j] = \mathbf{[B^i, P^j] = [B^i, K^j] = 0}, \\
&\mathbf{[K^0, P^0] = 2D}, \quad [K^0, P^i] = -2B^i, \quad [K^i, P^0] = 2B^i, \quad \mathbf{[K^i, P^j] = 0}.
\end{aligned}
\tag{2.13}
$$

Note that the commutators in bold are different from the ones in CCA. We re-list the commutators of $K$'s and $P$'s in the following matrix

$$
[K^\mu, P^\nu] = \begin{pmatrix}
2D & -2B^1 & -2B^2 & -2B^3 & \ldots \\
2B^1 & 0 & 0 & 0 & \ldots \\
2B^2 & 0 & 0 & 0 & \ldots \\
2B^3 & 0 & 0 & 0 & \ldots \\
\vdots & \vdots & \vdots & \vdots & \ddots
\end{pmatrix}. \tag{2.14}
$$

The generators can be understood as the vector fields acting on the space-time, as listed in Table 2.

The structure of this algebra is $\mathfrak{gca}_d \simeq (\mathfrak{so}(3) \times \mathfrak{so}(d-1)) \ltimes \mathbb{R}^{3(d-1)}$, where $\mathfrak{so}(3) = \{P^0, D, K^0\}$, $\mathfrak{so}(d-1) = \{J^{ij}\}$, and $\{P^i, B^i, K^i\} = \mathbb{R}^{3(d-1)}$, and the semi-direct product is slightly non-trivial as shown in Figure 1.

We can construct the Casimir operators of GCA by using the standard method: first construct the combinations of the generators in the universal enveloping algebra and then require

Table 2: GCA symmetry generators as the vector fields on the space-time.

| generator | vector field | finite transformation |
|---|---|---|
| $d$ | $t\partial_t + x^i\partial^i$ | $\lambda x^\mu$ |
| $p^\mu$ | $\left(\partial_t\,,\,\vec{\partial}\right)$ | $x^\mu + a^\mu$ |
| $k^\mu$ | $\left(2tx^\mu\partial^\mu - t^2\partial_t, -t^2\vec{\partial}\right)$ | $\left(\frac{t}{1-a^0 t}, \frac{\vec{x}-\vec{a}t^2}{(1-a^0 t)^2}\right)$ |
| $b^i$ | $-t\vec{\partial}$ | $(t, \vec{x} - t\vec{v})$ |
| $m^{ij}$ | $x^i\partial^j - x^j\partial^i$ | $(t, \mathbf{M}\vec{x})$ |

$$\mathfrak{so}(3) \ltimes \begin{cases} \mathbb{R} \times \mathbb{R} \times \cdots \times \mathbb{R} \\ \times \quad\; \times \qquad\qquad \times \\ \mathbb{R} \times \mathbb{R} \times \cdots \times \mathbb{R} \\ \times \quad\; \times \qquad\qquad \times \\ \mathbb{R} \times \mathbb{R} \times \cdots \times \mathbb{R} \end{cases}$$
$$\underbrace{\phantom{\mathbb{R} \times \mathbb{R} \times \cdots \times \mathbb{R}}}_{\rtimes}$$
$$\mathfrak{so}(d-1)$$

Figure 1: Structure of Galilean conformal algebra.

them to be invariant under the action of the generators to fix the relative coefficients. After some tedious work, we manage to find the Casimir operators for 4-dimensional GCA

$$\begin{aligned}
C_2 &= P^i K^i + B^i B^i\,,\\
C_3' &= \epsilon_{ijk} P^i B^j K^k\,,\\
C_4 &= P^i P^i K^j K^j - P^i P^j K^j K^i + 4P^i B^i B^j K^j - 4P^i B^j B^j K^i\,.
\end{aligned} \tag{2.15}$$

Similar to the case of CCA, the Casimir operators of GCA can be obtained by taking non-relativistic limit of the Casimir operators $\tilde{C}_2, \tilde{C}_3', \tilde{C}_4$ of conformal algebra as well,

$$\tilde{C}_2 \overset{c\to\infty}{\longrightarrow} C_2\,, \qquad \tilde{C}_3' \overset{c\to\infty}{\longrightarrow} C_3'\,, \qquad \tilde{C}_4 \overset{c\to\infty}{\longrightarrow} C_4\,. \tag{2.16}$$

There exists an infinite extension $\{L_n, M_n^i, J^{ij}\}$ for general $d$ dimension as well, with the commutation relations

$$\begin{aligned}
[L_n, L_m] &= (n-m)L_{n+m}\,, \quad [L_n, M_m^i] = (n-m)M_{n+m}^i\,, \quad [M_n^i, M_m^j] = 0\,,\\
[J^{ij}, L_n] &= 0\,, \quad [J^{ij}, M_n^k] = \delta^{ik}M_n^j - \delta^{jk}M_n^i\,, \qquad n \in \mathbb{Z}\,.
\end{aligned} \tag{2.17}$$

Its global part is the same as the $\mathfrak{gca}_d$, with the following nontrivial identifications

$$D = L_0\,, \quad B^i = M_0^i\,, \quad P^0 = L_{-1}\,, \quad P^i = -M_{-1}^i\,, \quad K^0 = L_1\,, \quad K^i = M_1^i\,. \tag{2.18}$$

The corresponding vector fields are of the forms

$$l_n = t^{n+1}\partial_t + (n+1)t^n x^i \partial_i\,, \quad m_n^i = -t^{n+1}\partial_i\,, \quad n \in \mathbb{Z}\,, \tag{2.19}$$

besides the ones $m^{ij}$ given in table 2.

## 2.3 Space-time structure

In the discussions above, we interpret the Carrollian/Galilean conformal symmetry as ultra/non-relativistic limit of relativistic conformal symmetry. In fact, there exists an intrinsic way to understand Carrollian/Galilean group as the symmetries of the underlying space-time structure [30–32, 37]. The two space-times are united into the Bargmann manifold: the Carrollian space-time is a null hyper-surface of the Bargmann manifold, while the Galilean space-time is the base space of the Bargmann manifold. The corresponding (extended) conformal group is the conformal extension of space-time symmetry group with isotropic scaling.

To be more specific, the Carrollian group is the space-time symmetry of Carrollian manifold $(\mathcal{C}, g^{\mathcal{C}}, \xi)$, where $\mathcal{C}$ is a $d$-dimensional smooth manifold endowed with a degenerate metric $g^{\mathcal{C}}$ and a vector $\xi$ which generates the kernel of $g^{\mathcal{C}}$. In a modern language, the manifold $\mathcal{C}$ is described as a fiber bundle with an 1-dimensional fiber of the coordinate $t$ and $(d-1)$-dimensional base space $\mathcal{B}^{\mathcal{C}}$ of the coordinates $x^i$. The simplest Carrollian manifold is the flat Carrollian space-time with $\mathcal{B}^{\mathcal{C}} = \mathbb{R}^{d-1}$, $\mathcal{C} = \mathcal{B}^{\mathcal{C}} \times \mathbb{R}^1$, $g^{\mathcal{C}} = \delta_{ij} dx^i \otimes dx^j$ and $\xi = \partial_t$. The Carrollian group generated by $\{P^\mu, B^i, J^{ij}\}$ is naturally the space-time symmetry of the flat Carrollian manifold $\mathcal{C}$.

However, to make the finite conformal extension of Carrollian symmetry close under $k^i$ transformations, we should compactify the space-time, which requires us to define the "infinity" properly. As the inversion $\mathbf{i}_E : x^\mu \to \frac{x^\mu}{x^2}$ plays an important rule in the case of Euclidean conformal symmetry, the *spacial inversion* $\mathbf{i}$ in the Carrollian space-time is essential to define the "infinity". The definition of the spacial inversion and its adjoint actions for the flat Carrollian space-time are[3]

$$\begin{aligned}
\mathbf{i} : (t, \vec{x}) &\to \left(\frac{t}{\vec{x}^2}, \frac{\vec{x}}{\vec{x}^2}\right), \quad \mathbf{i}^2 = id, \\
\mathbf{i} d \mathbf{i} &= -d, \quad \mathbf{i} p^\mu \mathbf{i} = -k^\mu, \\
\mathbf{i} k^\mu \mathbf{i} &= -p^\mu, \quad \mathbf{i} b^i \mathbf{i} = b^i, \quad \mathbf{i} m^{ij} \mathbf{i} = m^{ij},
\end{aligned} \tag{2.20}$$

where $id$ is the identity transformation. The above action of $\mathbf{i}$ on the finite transformations can be checked easily, for example:

$$\mathbf{i} p^\mu \mathbf{i} : \ (t, \vec{x}) \xrightarrow{\mathbf{i}} \left(\frac{t}{\vec{x}^2}, \frac{\vec{x}}{\vec{x}^2}\right) \xrightarrow{p^\mu} \left(\frac{t}{\vec{x}^2} + a^0, \frac{\vec{x}}{\vec{x}^2} + \vec{a}\right) \xrightarrow{\mathbf{i}} \frac{(t + a^0 \vec{x}^2, \vec{x} + \vec{a} \vec{x}^2)}{1 + 2\vec{a} \cdot \vec{x} + \vec{a}^2 \vec{x}^2}. \tag{2.21}$$

The spacial inversion $\mathbf{i}$ is not in the connected component of the Carrollian conformal group, but acting even times of $\mathbf{i}$ on a finite transform keeps it in the same connected component of the Carrollian conformal group.

It is expected that the spacial inversion $\mathbf{i}$ should map the "origin" to the "infinity". This requires us to handle the fiber bundle structure carefully. Firstly, we deal with the compactification of the base space $\mathcal{B}^{\mathcal{C}}$. Projecting to $\mathcal{B}^{\mathcal{C}}$, the Carrollian conformal group is reduced to the $(d-1)$-dimensional conformal group, and the spacial inversion $\mathbf{i}$ is reduced to the inversion $\mathbf{i}_E$ in $(d-1)$-dimensional space:

$$\begin{aligned}
d|_{\mathcal{B}^{\mathcal{C}}} &= d_E, \quad p^i|_{\mathcal{B}^{\mathcal{C}}} = p^i_E, \quad k^i|_{\mathcal{B}^{\mathcal{C}}} = k^i_E, \quad m^{ij}|_{\mathcal{B}^{\mathcal{C}}} = m^{ij}_E, \\
p^0|_{\mathcal{B}^{\mathcal{C}}} &= K^0|_{\mathcal{B}^{\mathcal{C}}} = b^i|_{\mathcal{B}^{\mathcal{C}}} = id_E, \quad \mathbf{i}|_{\mathcal{B}^{\mathcal{C}}} = \mathbf{i}_E,
\end{aligned} \tag{2.22}$$

where the symmetries with the label "$E$" represent the symmetries in $(d-1)$-dimensional Euclidean space. Thus we can add the infinity point $\vec{\infty}$ to $\mathbb{R}^{d-1}$ and define the compactification of the base space $\mathcal{B}^{\mathcal{C}}$ as a $(d-1)$-dimensional sphere $S^{d-1}$.

---

[3]There are actually four different well-applied choices for $\mathbf{i}$: $(t, \vec{x}) \to \left(\pm \frac{t}{\vec{x}^2}, \pm \frac{\vec{x}}{\vec{x}^2}\right)$, while the adjoint actions are the same up to a minus sign. However, different choices lead to the same result for the discussions in this sub-section.

We further consider $\mathbf{i}^2 = id$ acting successively on the fiber near the origin of the base space

$$(t, \vec{\epsilon}) \xrightarrow{\mathbf{i}} (t/\vec{\epsilon}^2, \vec{\epsilon}/\vec{\epsilon}^2) \xrightarrow{\mathbf{i}} (t, \vec{\epsilon}). \tag{2.23}$$

Taking $\vec{\epsilon} \to \vec{0}$, we have

$$(t, \vec{0}) \xrightarrow{\mathbf{i}} (\mathbf{t}, \vec{\infty}) \xrightarrow{\mathbf{i}} (t, \vec{0}), \tag{2.24}$$

where $\mathbf{t} = \lim_{\vec{\epsilon} \to \vec{0}} t/\vec{\epsilon}^2$ (with $\mathbf{t}_1 = \lim_{\vec{\epsilon} \to \vec{0}} t_1/\vec{\epsilon}^2 \neq \mathbf{t}_2 = \lim_{\vec{\epsilon} \to \vec{0}} t_2/\vec{\epsilon}^2$ for $t_1 \neq t_2$) generates a 1-dimensional space: $\mathbf{t} \in \tilde{\mathbb{R}} \cong \mathbb{R}$, such that the second $\mathbf{i}$ acts properly. Thus the compactified Carrollian space-time is $\mathcal{C} = S^{d-1} \times \mathbb{R}$ with the base space $\mathcal{B}^\mathcal{C} = S^{d-1}$. It is possible to choose other set of coordinates for $\mathcal{B}^\mathcal{C}$ to avoid awkward definition of $\mathbf{t} \in \tilde{\mathbb{R}}$, but in this paper we do not need to make such choice.

The Galilean group is the space-time symmetry of a Newton-Cartan manifold $(\mathcal{N}, g^\mathcal{N}, \theta)$, where $\mathcal{N}$ is a $d$-dimensional smooth manifold endowed with a symmetric $(2,0)$-tensor $g^\mathcal{N}$ and a 1-form $\theta$ which generates the kernel of $g^\mathcal{N}$. $\mathcal{N}$ is a fiber bundle with $\mathcal{B}^\mathcal{N}$ being its 1-dimensional base space. For the flat case we have $\mathcal{B}^\mathcal{N} = \mathbb{R}$, $\mathcal{N} = \mathcal{B}^\mathcal{N} \times \mathbb{R}^{d-1}$, $g^\mathcal{N} = \delta^{ij} \partial_i \otimes \partial_j$ and $\theta = dt$. The Galilean group is generated by $\{P^\mu, B^i, J^{ij}\}$, where $B^i$ is the Galilean boost. Similar to the Carrollian case, the Galilean conformal symmetry is the conformal extension of space-time symmetry on the compactified manifold $\mathcal{N} = S^1 \times \mathbb{R}^{d-1}$. In this case, it is temporal inversion $\mathbf{i}: (t, \vec{x}) \to \left(\pm\frac{1}{t}, \pm\frac{\vec{x}}{t^2}\right)$ relating the "origin" to the "infinity". The discussions are quite similar and we omit the details.

## 2.4 Conformal invariants

It is very different from the $d = 2$ case that the conformal invariants of 4-point insertions in higher-$d$ ($d \geq 3$) Carrollian space-time do not depend on the time-like degrees of freedom. We can always firstly fix the temporal coordinates $t_n = 0$ by using $d + 1 \geq 4$ differential operators: $\{p^0 = \partial_t, \ b^i = x^i \partial_t, \ k^0 = \vec{x}^2 \partial_t\}$. The rest symmetries are just $(d-1)$-dimensional conformal symmetry and it is standard to fix the insertions in the configuration[4]

$$(0, \vec{0}) \ (0, \vec{1}) \ (0, \vec{z}) \ (0, \vec{\infty}). \tag{2.25}$$

This leaves us with two independent conformal invariants $(z, \bar{z})$, which are exactly the same as the 4-point conformal invariants in CFT$_{d-1}$.

In fact, the number of time-like degrees of freedom of $n$-insertions is $n - (d + 1)$. Thus, when the number of insertions is not too large, the conformal invariants would be independent of time-like coordinates, and are the same as the ones in CFT$_{d-1}$. The dependence of time-like coordinates appears only in the conformal invariants of higher-point insertions. To be specific, we have

$$CCFT_d \ n\text{-pt invariants} = CFT_{d-1} \ n\text{-pt invariants}, \quad 4 \leq n \leq d + 1.$$

This is indeed the case for $n \leq d + 1$ insertions, since the $d + 1 \geq 4$ differential operators $\{p^0, \ b^i, \ k^0\}$ always fix all the temporal coordinates. Taking $d = 4$ for example, the 4-point and 5-point invariants do not depend on the time-like coordinates, being the same as the ones in 3-dim CFT, while the time-like dependence appears in the 6-point and even higher-point conformal invariants

$$\begin{aligned}
&\text{4-pt:} \quad (u, v) = (u, v)|_{(\text{CFT}_3)}, \\
&\text{5-pt:} \quad z_5^a = z_5^a|_{(\text{CFT}_3)}, \quad a = 1, \ldots, 5, \\
&\text{6-pt:} \quad z_6^a = z_6^a|_{(\text{CFT}_3)}, \quad a = 1, \ldots, 8, \quad z_6^9 \ \text{contains time-like d.o.f.}, \\
&n\text{-pt:} \quad z_n^a = z_n^a|_{(\text{CFT}_3)}, \quad a = 1, \ldots, 3n - 10, \\
&\qquad\qquad z_n^a \quad a = 3n - 9, \ldots, 4n - 15, \quad \text{contain time-like d.o.f.}.
\end{aligned} \tag{2.26}$$

---

[4] Recall that $\mathbf{t}_4$ is defined in the sense that $\mathbf{0} = \lim_{\vec{\epsilon} \to \vec{0}} 0/\vec{\epsilon}^2$.

It is obvious that this feature does not occur in $d = 2$, because there are only three relating differential equations, not enough to constrain all the time-like degrees of freedom.

There is no such strangely behaved invariants for the insertions in the Galilean space-time. It can be checked that for general $d > 2$ GCA, there are only two invariants which look similar to the ones in $2d$ GCA [25]. For 4-pt insertions in $4d$ GCA, the two invariants are $t$ and $r$, with

$$
\begin{aligned}
t &= \frac{t_{12}t_{34}}{t_{13}t_{24}}, \\
r &= |\vec{x}| = \left| t\left(\frac{\vec{x}_{12}}{t_{12}} + \frac{\vec{x}_{34}}{t_{34}}\right) - t\left(\frac{\vec{x}_{13}}{t_{13}} + \frac{\vec{x}_{24}}{t_{24}}\right) \right|.
\end{aligned}
\tag{2.27}
$$

Obviously the invariants are not independent of the space-like coordinates. The key difference is that the base space of Galilean space-time is 1-dimensional $S^1$ and the fiber is $(d-1)$-dimensional $\mathbb{R}^{d-1}$. The differential equations are not enough to fix all the space-like coordinates.

At first looking, the conformal invariants in the Carrollian space-time are unusual. Another way to understand them is from taking the limits on the usual conformal invariants in Minkowski space-time. Actually the conformal invariants for the Carrollian or Galilean space-time could be obtained by taking ultra- or non-relativistic limit of the conformal invariants in usual space-time.

# 3 Representations and Local Operators

The systematic way of constructing and classifying local operators in relativistic CFTs dates back to Mack and Salam's work [56], and the resulting highest weight representations[5] contain only primary operators $\mathcal{O}^a$ and their descendants (derivative operators) $\partial^n \mathcal{O}^a$, without other unidentified operators to close the action of the symmetry algebra. We first briefly review this method of constructing local operators, and then apply it to discuss finite-component field operators in CCFT and GCFT. Since the structures of the involved algebras are similar, we consider the case of CCA and CCFT in details and leave the discussions of GCA and GCFT to section 3.5.

In section 3.1 we explain the induction method of constructing local operators. In section 3.2 we introduce the tensor representations of CCA rotation as motivating examples. In section 3.3 we discuss the constraints on general finite-dimensional representations of CCA rotation. In section 3.4 we give the definition of the highest-weight representations and local operators of CCA.

The highest-weight representations for GCFT and CCFT have been discussed to some extent in the literature [36, 54]. In particular, the scale-spin representations were studied and were nicely applied to the study of specific Galilean/Carrollian field theories [36]. The finite-dimensional scale-spin representation in fact fits in the multiplet representation in section 3.4, and we compare them in 3.6.

For concreteness, the discussions in this section mainly focus on $d = 4$, and can be applied to other dimensions $d \geq 3$ as well. In Appendix C, we discuss the cases of other dimensions, especially for the special $d = 3$ case.

## 3.1 Construction of local operators

In the following we use the terms "group" and "algebra" interchangeably. Similar to the relativistic case, to include the spinors in non-Lorentzian theories we need to replace the conformal

---

[5]More precisely, they are parabolic Verma modules, see e.g. the explanations in [10].

group by its covering group. For example, the Carrollian case is $ISO(d,1) \rightarrow \mathrm{Spin}(d,1) \ltimes \mathbb{R}^{d+1}$. The method of constructing local operators is as follows:

1. Denoting the conformal group as $G$ with Lie algebra $\mathfrak{g}$, a symmetry transformation $g \in G$ is represented as a unitary operator $U(g)$ on the Hilbert space, and acts on other operators adjointly, $\mathcal{O} \rightarrow \mathcal{O}' = U(g)\mathcal{O}U(g)^{-1}$. A local operator $\mathcal{O}(x)$ depends only on its insertion point and should respect the symmetry, hence $\mathcal{O}(x)'$ is an operator located at $g(x)$. Now assuming a complete basis of local operators $\{\mathcal{O}^a(x)\}$, by the logic above $\mathcal{O}^a(x)'$ can be expanded into linear combinations of $\mathcal{O}^a(g(x))$,

$$\mathcal{O}^a(x)' = R^a_b(g^{-1}, x)\mathcal{O}^b(g(x)), \tag{3.1}$$

where $R^a_b(g^{-1}, x)$ are the combination coefficients, and the inverse $g^{-1}$ is to preserve the composition $g_1 g_2$.

2. To determine $R^a_b(g, x)$, consider the transformations $g_0$, which keep the origin $x = 0$ intact. These transformations compose a stabilizer subgroup (little group) $G_0$ with Lie algebra $\mathfrak{g}_0$. By (3.1) they act on $\mathcal{O}^a := \mathcal{O}^a(0)$ as

$$U(g_0)\mathcal{O}^a U(g_0)^{-1} = R^a_b(g_0^{-1}, 0)\mathcal{O}^b. \tag{3.2}$$

Hence $R^a_b(g_0, 0)$ is a representation of $G_0$, and we need to construct and classify representations of $\mathfrak{g}_0$.

3. Choosing a representation $R$ of $\mathfrak{g}_0$ on vector space $V_R$, let the operators $\mathcal{O}^a \in V_R$ freely move away from the origin, by the action of translation operator $U(x) = \exp x^\mu P_\mu$,

$$\mathcal{O}^a(x) = U(x)\mathcal{O}^a(0)U(x)^{-1}, \tag{3.3}$$

then the action of the conformal group $G \ni g$ on $\mathcal{O}^a(x)$ is induced by the representation $R$. To be concrete, the action of $g$ on $\mathcal{O}(x)$ is equivalent to the action of $(gx) \in G$ on $\mathcal{O}(0)$. Using the coset decomposition $gx = x'g_0$ with $g_0 \in G_0$, the action turns to locate the action of $g_0$ on $\mathcal{O}$ at $x'$:

$$\begin{aligned} U(g)\mathcal{O}^a(x)U(g)^{-1} &= U(gx)\mathcal{O}^a(0)U(gx)^{-1} \\ &= U(x')(U(g_0)\mathcal{O}^a(0)U(g_0)^{-1})U(x')^{-1} \\ &= R^a_b(g_0^{-1})\mathcal{O}^b(x'). \end{aligned} \tag{3.4}$$

In practice, we use the BCH formula to derive the infinitesimal transformations of $G$.

In the cases of CFT, CCFT and GCFT, the stabilizer algebras $\mathfrak{g}_0$ share a similar structure that helps us to simplify the discussion. They are all made up of three subalgebras: dilation $D$, generalized rotations[6] $M = \{J, B\}$ and special conformal transformations (SCTs) $K$ respectively. And the commutation relations are: $[D, M] = 0$, and $[D, K] \subset K$, $[M, K] \subset K$, i.e., $K$ is a representation of $D$ and $M$. The commutativity of the dilatation and the rotations implies that the local operators $\mathcal{O}^a$ can be diagonalized into the eigenstates of the dilation,[7] $[D, \mathcal{O}] = \Delta_\mathcal{O}\mathcal{O}$, and simultaneously into a representation of the rotations, $[M, \mathcal{O}^a] = M^a_b\mathcal{O}^b$.

Following the terminology of [56], the finite-dimensional representations of $G_0$ are called as type I, describing finite-component field operators, and infinite-dimensional ones are type II.

---

[6]Here $J$ is the spatial rotation, and $B$ is the Lorentzian, Carrollian or Galilean boost respectively.

[7]The local operators can be generalized eigenstates of $D$, accounting for the logarithmic multiplets in logarithmic CFTs.

Furthermore, the representations satisfying the primary-like conditions $[K, \mathcal{O}^a] = 0$ are called type a, otherwise are called type b.

In compact CFTs where the dilatation spectrum is discrete and bounded below, $[K, \mathcal{O}] = 0$ can always be satisfied. However, in non-unitary CFTs, CCFT and GCFT, a priori there is no physical reason guaranteeing this condition. For simplicity, in this work we focus on the type Ia case. Hence the remaining task is to construct and classify finite-dimensional representations of the rotation subalgebra $M$.

For CFT$_2$ with only global symmetries $SO(3,1)$ or $SO(2,2)$, and Galilean/Carrollian CFT$_2$ with $ISO(2,1)$, the rotation groups are $SO(2)$ and $\mathbb{R}$ respectively and hence are non-semisimple. The finite dimensional representations give logarithmic multiplets, e.g. [57] and boost multiplets [25] respectively.

For CFT$_d$, with $d \geq 3$, the rotation group $SO(d)$ or $SO(d-1,1)$ is semisimple, and finite dimensional representations are completely reducible. However for $d \geq 3$ CCFT and GCFT, the generalized rotation group, CCA and GCA rotation group respectively, is the Euclidean group $ISO(d-1)$. The finite dimensional representations are not completely reducible, and the building blocks are indecomposable representations.

## 3.2 Tensor representations of CCA rotation

To construct all representations of CCA rotation $ISO(3)$ is difficult. In this subsection, we start from the examples of tensor representations, and get some hints of the constraints on general representations. The tensor representations can be found in two ways: the first is taking ultra-relativistic limit $c \to 0$ of $SO(4)$ tensor representations; the second is defining the vector representation and then using tensor product to get higher-rank tensor representations. In the following, we mainly use the second approach and leave the detailed discussions of taking limit to Appendix A.

The simplest case is the scalar representation: the primary operator $\mathcal{O}$ is invariant under CCA rotations

$$[J^{ij}, \mathcal{O}] = [B^i, \mathcal{O}] = 0. \tag{3.5}$$

The interesting structures appear in the following non-trivial representations.

### 3.2.1 Vector representations

The simplest non-trivial case is the vector representation, denoted as $V$. The vector operators $\mathcal{O}^\mu \in V$ transform as covariant vectors $P^\mu$ under the CCA rotation, and thus from (2.1) the actions of CCA rotation are

$$[J^{ij}, \mathcal{O}^k] = \delta^{ik}\mathcal{O}^j - \delta^{jk}\mathcal{O}^i, \quad [B^i, \mathcal{O}^j] = \delta^{ij}\mathcal{O}^0, \quad [J^{ij}, \mathcal{O}^0] = [B^i, \mathcal{O}^0] = 0. \tag{3.6}$$

From the inclusion $SO(3) \subset ISO(3)$, this vector representation can be organized as spin-1 and spin-0 $SO(3)$ representations, which are related to each other by the boost operators. The explicit relations are shown in Figure 2, where for simplicity, we organize the CCA rotation generators as

$$J = -iJ^{12}, \quad J^\pm = \frac{1}{\sqrt{2}}(\mp J^{23} + iJ^{31}), \quad B^\pm = \frac{1}{\sqrt{2}}(iB^1 \pm B^2), \tag{3.7}$$

with the commutation relations:

$$
\begin{aligned}
&[J, J^\pm] = \pm J^\pm, \quad [J^+, J^-] = J, \\
&[J, B^\pm] = \pm B^\pm, \quad [J, B^3] = 0, \\
&[J^+, B^+] = 0, \quad [J^+, B^3] = B^+, \quad [J^+, B^-] = B^3, \\
&[J^-, B^+] = B^3, \quad [J^-, B^3] = B^-, \quad [J^-, B^-] = 0.
\end{aligned} \tag{3.8}
$$

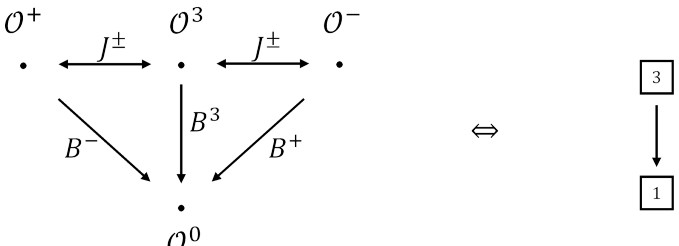

Figure 2: The vector representation of CCA rotation group. The meaning of this Young diagram structure will be introduced in section 3.2.2.

The spin-1 $SO(3)$ part $\mathcal{O}^i$ is organized as $\mathcal{O}^\pm = \frac{1}{\sqrt{2}}(i\mathcal{O}^1 \pm \mathcal{O}^2)$, being the eigen-operators of $J$ and related to each other by $J^\pm$.

From Figure 2 it is immediately noticed that $\mathcal{O}^0$ spans a subrepresentation $V_0 \subset V$ of CCA rotation and there is no other sub-representation $V^\perp$ such that $V_0 \oplus V^\perp = V$. Hence the vector representation $V$ is reducible but indecomposable.

The reducibility of vector representation can also be seen from the taking-limit procedure. For simplicity consider the two operators $(\mathcal{O}^0, \mathcal{O}^3)$, after taking limit we have

$$\left[-iJ^{03}, \frac{1}{\sqrt{2}}(i\mathcal{O}^0 \pm \mathcal{O}^3)\right] = \pm \frac{1}{\sqrt{2}}(i\mathcal{O}^0 \pm \mathcal{O}^3) \xrightarrow{c \to 0} i\left[B^3, \frac{1}{\sqrt{2}}(i\mathcal{O}^0 \pm \mathcal{O}^3)\right] = \pm \frac{i}{\sqrt{2}}\mathcal{O}^0 . \quad (3.9)$$

Namely, the representation matrix of $B^3$ becomes non-diagonalizable after taking limit:

$$J^{03} = \begin{pmatrix} 0 & 1 \\ -1 & 0 \end{pmatrix} \xrightarrow{c \to 0} -B^3 = \begin{pmatrix} 0 & 0 \\ -1 & 0 \end{pmatrix}. \quad (3.10)$$

Other $B$-matrices also contain Jordan blocks, hence the operators cannot be organized as the eigen-operators of the $B$ generators. And from the Jordan blocks we can find the nontrivial subrepresentation.

Not only for the vector representation, the breakdown of complete reducibility is an unavoidable feature for generic representations of the CCA rotation $ISO(d-1)$. To conveniently describe this kind of representations we firstly introduce some terminologies. We call the *multiplet* representations as such type of reducible but indecomposable representations: they are finite direct sums of subspaces $V = \bigoplus_{n=1}^N V_n$, where $V_n$ are irreducible representations of $SO(3)$ and are connected by $B$ generators. These subspaces $V_n$ are named as *sub-sectors* of a multiplet. Due to the finiteness of $N$, the operators in each sub-sector can be annihilated by finite times of $B$'s actions, and the minimal number of times is called the *order* of the sub-sector.[8] The *rank* of a multiplet representation is defined as the maximal order of all sub-sectors. The rank-1 multiplet representations are also called *singlet* representations, which are in fact $SO(3)$ irreducible representations. For example, the vector representation is a rank 2 multiplet with $\{\mathcal{O}^+, \mathcal{O}^3, \mathcal{O}^-\}$ and $\{\mathcal{O}^0\}$ being the irreducible $SO(3)$ sub-sectors of order 2 and 1 respectively. We will elaborate these points in section 3.3, and now turn to the construction of tensor representations.

### 3.2.2 Higher rank tensor representations

Taking tensor product of vector representations $V$ and decomposing it into indecomposable ones, we get tensor representations of CCA rotation, $V^{\otimes k} = \bigoplus_{n=1}^N V_n$. And to describe the

---

[8]The definition of order here is good enough for tensor representations, but not for general representations. A self-consistent definition can be found in section 3.3.

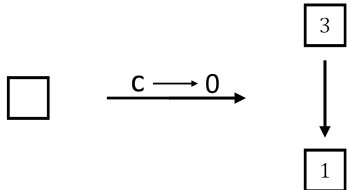

Figure 3: The vector representation of CCA by taking the limit from vector representation of $SO(4)$.

structure of $V_n$, we need to generalize the Young diagram to label the mixed symmetry of indices and the boost action. We briefly recall the idea of Young diagram and then generalize it to the CCA rotation $ISO(d-1)$.

The idea of Young diagram is as follows. Starting from the vector representation $V$ of $GL(d)$ (or its compact form $U(d)$), the general linear group $GL(d)$ and the symmetric group $S_k$ simultaneously act on the the tensor representation $V^{\otimes k}$. The joint actions of $GL(d) \times S_k$ commute, hence $V^{\otimes k}$ should split into $V^{\otimes k} = \bigoplus_{n=1}^{N} V_n$, where $V_n$ are irreducible representations of $S_k$ and are in one-to-one correspondence with Young diagrams. Moreover by the Schur-Weyl theorem $V_n$ are also irreducible with respect to $GL(d)$. But some representations of $GL(d)$ are missing. For example, the determinant $R: g \mapsto \det(g)$ corresponds to the Young diagram $[n]$ with $n$ rows and 1 column, but $R_m: g \mapsto \det(g)^m, m \in \mathbb{Z}$ with $m < 0$ cannot be characterised by any Young diagram.

Descending to $SL(d) \subset GL(d)$ (or the compact ones $SU(d) \subset U(d)$), the determinant representations $R_m$ are trivial and all the $V_n$ remain irreducible, hence we arrive at the familiar fact: the Young diagrams with the rows less than $d$ are in one-to-one correspondence with the representations of $SL(d)$ or $SU(d)$.[9] Then for $SO(d) \subset SL(d)$, $V_n$ become reducible and can be decomposed into irreducible ones after splitting the trace parts. The correspondence between the Young diagrams and the irreducible representations could be broken at two levels: firstly the spinor representations are missed; secondly there are redundancies of Young diagrams. For $SU(d), d \geq 3$ the quark $[1]$ and anti-quark $[d-1]$ are not isomorphic, but for $SO(d)$ they are isomorphic due to the metric tensor. Only the Young diagrams with $\leq \frac{d}{2}$ rows gives non-isomorphic irreducible tensors.

Now for the CCA rotation $ISO(d-1)$, in the previous subsection we use $SO(d-1) \subset ISO(d-1)$ to label the spins of sub-sectors in the vector representation $V$ and add the arrows to characterise the boost actions between the sub-sectors. We can still decompose the tensors $V_n$ in terms of the Young diagrams, $SO(d-1) \subset ISO(d-1) \subset SL(d)$, as we show below.

In the vector case $V$, before taking the limit $c \to 0$ the index of $v^a \in V$ is labeled by a $SO(4)$ box, and the 4 components of $v^a$ correspond to the box filled by indices $a = 0, 1, 2, 3$. After taking the limit, the $SO(4)$ symmetry is broken and by $ISO(d-1) \subset SL(4)$ we need $SL(4)$ or equivalently $SU(4)$ boxes instead. The 4 components of the vector are decomposed to 3 spacial components as an $SO(3)$ vector denoted by $\boxed{3}$, and 1 temporal component as an $SO(3)$ scalar denoted by $\boxed{1}$.[10] Then the arrows connecting different sub-sectors represent the action of the boosts. This is illustrated in Figure 3.

For higher-rank tensors we need to bookmark the contractions of the spatial indices of $SO(3)$ in the $SU(4)$ Young diagrams. The contraction of $SO(3)$ indices is denoted by $\boxed{3}\!\!-\!\!\boxed{3}$

---

[9]Strictly speaking, for $SL(d)$ and $GL(d)$, $V_n$ are all holomorphic and there are also anti-holomorphic representations by taking complex conjugate. For these groups, the complex conjugate representations are not related to the dual representations.

[10]We apologize to the notation here: the 3 or 1 in boxes are not the third or first component, but the dimension of $SO(3)$ representations.

explicitly. The results of the rank-2 and rank-3 tensor representations are shown in Figure 4 and 5 respectively. The rank-2 tensor representation of CCA rotations is decomposed into a 10-dimensional representation and a 6-dimensional representation, and the rank-3 tensor representation is decomposed into three 20-dimensional representations and a 4-dimensional representation.

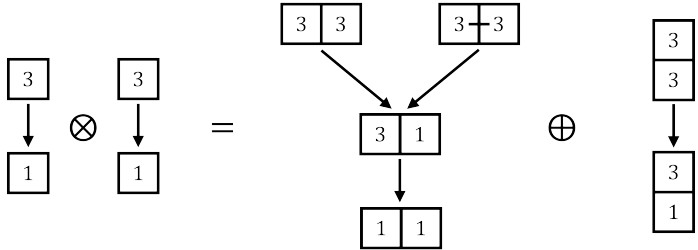

Figure 4: The rank-2 tensor representation of CCA.

From the above examples we find that for the decomposition $V^{\otimes k} = \bigoplus_{n=1}^{N} V_n$ of $SL(d) \subset GL(d)$, all the $V_n$ remain indecomposable with respect to $ISO(d-1) \subset SL(d)$. We believe this is true for arbitrary rank $k$ and dimension $d$. Then the algorithm of decomposition can be summarized as follows:

1. Write down all the possible $SU(d)$ Young diagram corresponding to $V_n$. Every diagram corresponds to an indecomposable sector of $ISO(d-1)$;

2. For every diagram, fill every box in the first $d-1$ rows with label $\boxed{d-1}$ representing the $(d-1)$ spacial indices. Then write down all possible contraction of $SO(d-1)$ indices using notation similar to $\boxed{3\!\!+\!\!3}$;

3. Considering the action of the boosts, replace one $\boxed{d-1}$ label by $\boxed{1}$ to get a new sub-sector and draw an arrow to this new sub-sector from the old one. Repeat this step until there is one $\boxed{1}$ label for every column (since there is only one temporal index and it can not be anti-symmetric with itself);

4. Take other $SU(d)$ Young diagrams in the step 1 and repeat the steps 2 and 3. The dimensions of the sub-sectors can be read by peeling all boxes filled with $\boxed{1}$ and the contractions $\boxed{3\!\!+\!\!3}$, and then view the rest as the Young diagram of $SO(d-1)$.

In this diagrammatic method, each of the $SU(d)$ Young diagram in the net is an irreducible $SO(d-1)$ representation, and corresponds to the projector $P = P_{\text{trace}}P_s P_t P_0$, where $P_0$ is the standard Young projector of $SU(d)$, $P_s, P_t$ are projectors to spatial and temporal components, and $P_{\text{trace}}$ is the projector to ensure the traceless condition.

Since each sub-sector is a representation of $SO(d-1)$, this generalized $SU(d)$ Young diagrams can be equivalently replaced by $SO(d-1)$ Young diagrams. There are several advantages of $SU(d)$ Young diagrams comparing with $SO(d-1)$ Young diagrams: the number of boxes in every sub-sector of is equal to the rank of the tensor, and this fact gets lost if using the Young diagram of $SO(d-1)$ directly; it is convenient for computation since the indices and contractions are kept explicitly. The disadvantage is that there can be redundancies: different generalized $SU(d)$ Young diagrams can correspond to the same $SO(d-1)$ representation.

Back to $d = 4$, for future convenience we also introduce new notation in the lower panel of Figure 5, where $(j)$ labels a spin-$j$ $SO(3)$ sub-sector. As explained in Appendix A, one major

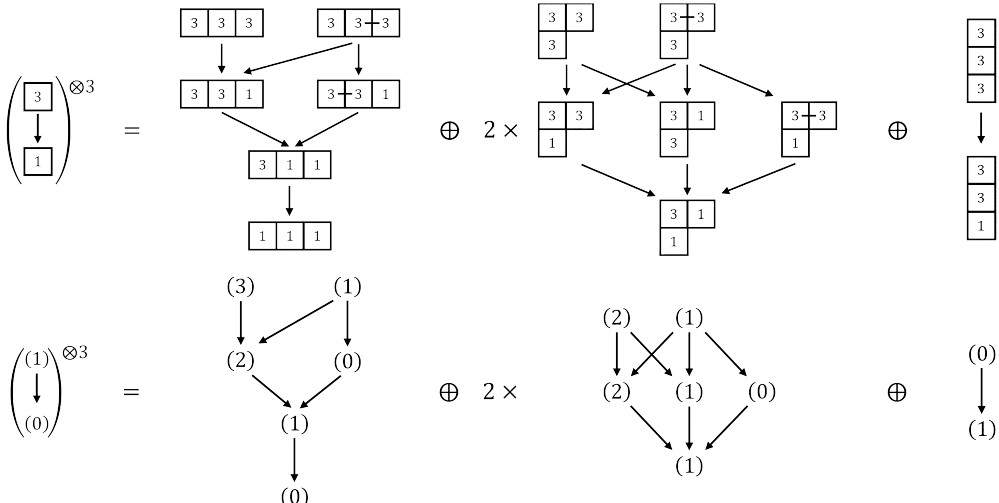

Figure 5: The rank-3 tensor representation of CCA. The upper panel shows the decomposition in terms of Young diagram, with the number in the box representing the spacial indices and temporal indices respectively. The lower panel represents the same decomposition, but now the subsectors are labeled by the representations of $SO(3)$ directly, where $(j)$ labels the spin-$j$ representation of dimension $2j + 1$. The last summand $(0) \to (1)$ is isomorphic to the dual vector $V^\vee$. Since $ISO(d-1)$ is not in $SU(d)$, the dual of $V$ is not isomorphic to $V$ itself.

difference between the tensor representations of CCA rotation and $SO(4)$ is that after taking the limit some decomposable representations of $SO(4)$ become indecomposable in CCA. For example the symmetric part of the rank-2 tensor representation of $SO(4)$ can be decomposed into a symmetric traceless part and a trace part, as shown in Figure 14 in Appendix A, while the symmetric part of the rank-2 tensor representation of the CCA rotation group can not be decomposed into the traceless and the trace parts, which instead are connected by the $B^i$ generators as shown in Figure 4.

Besides the direct sum decomposition of the tensor product, a more broad class of examples are the subrepresentations of tensors, which can be found by selecting some $SO(3)$ sub-sectors and collecting all sub-sectors along the arrows till the end. This is feasible because the arrows of $B$ are one-way arrows - there is no generator sending the lower sub-sectors back upwards (strictly upper triangular matrix in the sense of representation matrix), and thus the lower sub-sectors form a sub-representation. For example, one can get a 5-dimensional representation: $(0) \to (1) \to (0)$ starting from the $\boxed{3\ 3}$ sub-sector in the first part of the rank-2 tensor representation.

## 3.3 General representations of CCA rotation

The general representations of CCA rotation are rather complicated, but they are worthy studying. Firstly they appear in the concrete models. For example, the Carrollian $U(1)$ gauge fields $A$ [40] are in a chain representation, and as will be shown in a subsequent work the stress tensor $F$ is in a net representation. In some other papers, the operators other than the singlet representation have been studied, see e.g. [36, 38]. Secondly, if the operator product expansion (OPE) exists in higher dimensional CCFT, the operators in all possible representations can appear in the OPE even if the external operators are in some simple representations.

It turns out that not all finite-dimensional representations can be derived from the sub-representations of tensor representations, and we need a bottom-up method of constructing

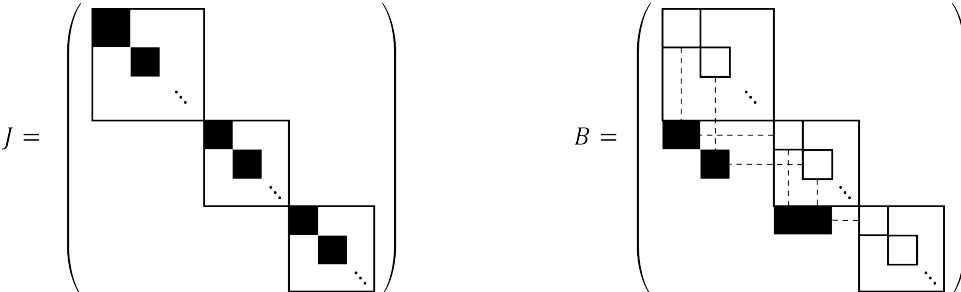

Figure 6: The matrix representation of the CCA rotations.

representations of the CCA rotation. The following theorem from [58] is useful for characterizing the structure of finite dimensional representations of the CCA rotation:

**Theorem 1** *Set a Lie algebra* $\mathfrak{g} = \mathfrak{g}_0 \ltimes \mathfrak{n}$, *where* $\mathfrak{g}_0$ *is a semi-simple Lie algebra and* $\mathfrak{n}$ *a nilpotent Lie algebra. The representation of* $\mathfrak{g}$ *on a finite dimensional vector space* $V$ *is such that there is a sequence of subspaces of* $V$: $0 = W_0 \subsetneq W_1 \subsetneq \cdots \subsetneq W_r = V$, *where each* $W_i$ *is invariant and completely decomposable under* $\mathfrak{g}_0$. *The* $\mathfrak{n}$ *elements maps subspace* $W_i$ *to* $W_{i-1}$ *for* $i = 1, \ldots, r$.

Applying this theorem to our case $\mathfrak{g}_0 = \mathfrak{so}(3)$ and $\mathfrak{n} = \{B^i\}$, the finite-dimensional representations of the CCA rotations are all multiplet representations with every sub-sectors being irreducible representations of $SO(3)$. The boost generators $\mathfrak{n}$ map sub-sectors to sub-sectors of one order lower since $(\mathrm{ad}_B)^1 B = 0$. Here we provide the rigorous definition of the term "order": the operators in $W_1$ have order 1, and the operators in $W_i/W_{i-1}$ have order $i$.

It should be stressed that this theorem implies the action of the $B^i$ generators on the operator $\mathcal{O}$ cannot give the terms proportional to $\mathcal{O}$ itself. This means that the boost charge in [51] should vanish $\xi = 0$ for all finite-component field operators in $d \geq 3$.

To get full constraints, we consider the representation matrix. By the theorem 1, the general representation matrix of the CCA rotations would be like the form shown in Figure 6, where the black blocks are non-zero. The diagonal black blocks of $J$'s represent $SO(3)$ irreducible representations, and the blocks in the same big square are sub-sectors in the same order. The $B$ matrices, as discussed above, are off square-diagonal matrices mapping $SO(3)$ representations to $SO(3)$ representations of lower order.

Using the specific algebraic structure, one can further fix the matrix blocks. For $J^{ij}$'s, the matrix blocks are exactly the well-known matrices of irreducible $SO(3)$ representations, which we repeat here

$$\mathbf{J}_j |j, m\rangle = m |j, m\rangle \, ,$$

$$\mathbf{J}_j^+ |j, m\rangle = \sqrt{\frac{1}{2}(j+m+1)(j-m)} |j, m+1\rangle \, ,$$

$$\mathbf{J}_j^- |j, m\rangle = \sqrt{\frac{1}{2}(j-m+1)(j+m)} |j, m-1\rangle \, . \tag{3.11}$$

There are three types of $B$ matrix blocks. Since $B^i$'s form a spin-1 $SO(3)$ representation, by the tensor product decomposition $(j) \otimes (1) = (j-1) \oplus (j) \oplus (j+1)$, their actions on $SO(3)$ representation $(j)$ give $(j)$ or $(j \pm 1)$ representations. Using the Wigner-Eckart theorem we can determine the $B$ matrix blocks up to an overall coefficient, which further can be absorbed into

the $SO(3)$ representations $|j_i, m\rangle \to c_i |j_i, m\rangle$. The resulting matrix blocks are

$$
\begin{aligned}
(\mathbf{B}^{m=a}_{j \to j+1})_{m_1, m_2} &= \delta_{m_2, m_1+a} \langle j, m_1; 1, a | j+1, m_1+a \rangle \langle j+1, B, j \rangle \\
&= \delta_{m_2, m_1+a} \sqrt{(j+1)(2j+1)} \langle j, m_1; 1, a | j+1, m_1+a \rangle , \\
(\mathbf{B}^{m=a}_{j \to j})_{m_1, m_2} &= \delta_{m_2, m_1+a} \langle j, m_1; 1, a | j, m_1+a \rangle \langle j, B, j \rangle \\
&= \delta_{m_2, m_1+a} \sqrt{j(j+1)} \langle j, m_1; 1, a | j, m_1+a \rangle , \\
(\mathbf{B}^{m=a}_{j \to j-1})_{m_1, m_2} &= \delta_{m_2, m_1+a} \langle j, m_1; 1, a | j-1, m_1+a \rangle \langle j-1, B, j \rangle \\
&= \delta_{m_2, m_1+a} \sqrt{j(2j+1)} \langle j, m_1, 1, a | j-1; m_1+a \rangle ,
\end{aligned}
\tag{3.12}
$$

where $\langle j_2, B, j_1 \rangle$ is the reduced matrix element. Concretely they are[11]

$$
\begin{cases}
\mathbf{B}^3_{j \to j+1} |j, m\rangle = \sqrt{(j+m+1)(j-m+1)} |j+1, m\rangle , \\[2mm]
\mathbf{B}^+_{j \to j+1} |j, m\rangle = \sqrt{\dfrac{1}{2}(j+m+2)(j+m+1)} |j+1, m+1\rangle , \\[2mm]
\mathbf{B}^-_{j \to j+1} |j, m\rangle = \sqrt{\dfrac{1}{2}(j-m+2)(j-m+1)} |j+1, m-1\rangle ,
\end{cases}
\tag{3.13}
$$

$$
\begin{cases}
\mathbf{B}^3_{j \to j} |j, m\rangle = m |j, m\rangle , \\[2mm]
\mathbf{B}^+_{j \to j} |j, m\rangle = -\sqrt{\dfrac{1}{2}(j+m+1)(j-m)} |j, m+1\rangle , \\[2mm]
\mathbf{B}^-_{j \to j} |j, m\rangle = \sqrt{\dfrac{1}{2}(j-m+1)(j+m)} |j, m-1\rangle ,
\end{cases}
\tag{3.14}
$$

$$
\begin{cases}
\mathbf{B}^3_{j \to j-1} |j, m\rangle = -\sqrt{(j+m)(j-m)} |j-1, m\rangle , \\[2mm]
\mathbf{B}^+_{j \to j-1} |j, m\rangle = \sqrt{\dfrac{1}{2}(j-m)(j-m-1)} |j-1, m+1\rangle , \\[2mm]
\mathbf{B}^-_{j \to j-1} |j, m\rangle = \sqrt{\dfrac{1}{2}(j+m)(j+m-1)} |j-1, m-1\rangle .
\end{cases}
\tag{3.15}
$$

The remaining commutation relations $[B, B] = 0$ restrict the chain representations (which means $(j_1) \to (j_2) \to \cdots \to (j_3)$ without any branches) to be of the following forms

$$
\begin{aligned}
&(0) \to (1) \to (0) , \\
\cdots \to &(j) \to (j+1) \to (j+2) \to \cdots , \\
\cdots \to &(j) \to (j-1) \to (j-2) \to \cdots
\end{aligned}
\tag{3.16}
$$

For example, $(2) \to (1) \to (0)$ is an allowed representation, but $(0) \to (1) \to (2) \to (1) \to (0)$ is forbidden since $\cdots \to (1) \to (2) \to (1) \to \cdots$ is not an allowed pattern.

For more complicated net representations, the constraints by $[B, B] = 0$ are very weak. For example, one can construct four kinds of net representations as shown in Figure 7 with different middle level.

Formally, an $ISO(3)$ representation $R$ can be labelled by a directed graph $G_R$, including a set of the vertices $V$ each associated with an irreducible $SO(3)$ representation and a set of

---

[11]We relabel the magnetic quantum number of a $SO(3)$-vector $V^a$ by $V^{m=\pm1} = V^\pm$, $V^{m=0} = V^3$, to distinguish it from the temporal component $V^0$.

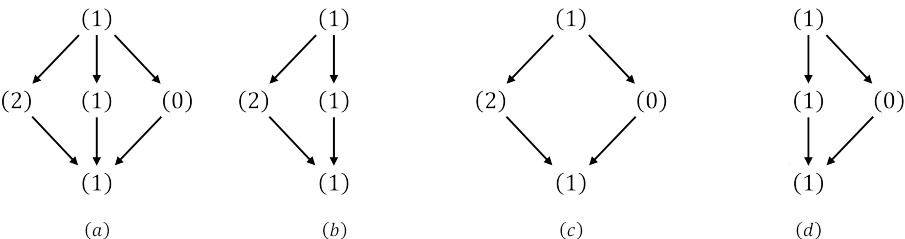

Figure 7: All the four net representations are legal although the middle level of the representations are different.

arrows $E$ showing the actions of $B$'s. To take all the constraints from $[B, B] = 0$ into account, we need to consider all directed-path between two vertices $(j_1)$, $(j_2)$ joined by two successive arrows $(j_1) \to (j_i) \to (j_2)$.

Since the representation is finite-dimensional, we can insert a complete basis into $[B^a, B^b] = 0$, $a, b = \pm, 3$, and find that there are only three possibly non-vanishing terms

$$
\begin{aligned}
\langle j_2, m_2 | [B^a, B^b] | j_1, m_1 \rangle &= \sum_n \langle j_2, m_2 | B^a | n \rangle \langle n | B^b | j_1, m_1 \rangle - (a \leftrightarrow b) \\
&= (c_1' \langle j_2, m_2 | B^a | j_1 - 1, m_1 + b \rangle \langle j_1 - 1, m_1 + b | B^b | j_1, m_1 \rangle \\
&\quad + c_2' \langle j_2, m_2 | B^a | j_1, m_1 + b \rangle \langle j_1, m_1 + b | B^b | j_1, m_1 \rangle \\
&\quad + c_3' \langle j_2, m_2 | B^a | j_1 + 1, m_1 + b \rangle \langle j_1 + 1, m_1 + b | B^b | j_1, m_1 \rangle) \\
&\quad - (a \leftrightarrow b),
\end{aligned}
\tag{3.17}
$$

where we have applied the Wigner-Eckart theorem to calculate the matrix elements of $B$'s, and $c_1', c_2', c_3'$ are normalization factors of $(j_1 - 1)$, $(j_1)$ and $(j_1 + 1)$ respectively. Then the equation (3.17) turns to

$$
c_1 \mathbf{B}^a_{j_1 - 1 \to j_2} \mathbf{B}^b_{j_1 \to j_1 - 1} + c_2 \mathbf{B}^a_{j_1 \to j_2} \mathbf{B}^b_{j_1 \to j_1} + c_3 \mathbf{B}^a_{j_1 + 1 \to j_2} \mathbf{B}^b_{j_1 \to j_1 + 1} - (a \leftrightarrow b) = 0.
\tag{3.18}
$$

This gives $j_1 \times j_2$ over-constrained equations for $c_i$'s. To solve them we firstly determine the possible values of $j_2$. The decomposition of $B^a B^b | j_1, m \rangle$ is

$$
(1) \otimes_{\text{sym}} (1) \otimes (j_1) = 2(j_1) \oplus (j_1 - 1) \oplus (j_1 - 2) \oplus (j_1 + 1) \oplus (j_1 + 2),
\tag{3.19}
$$

in which the symmetric tensor product $\otimes_{\text{sym}}$ is due to $[B, B] = 0$, hence there are five choices of $(j_2)$.

1. Case 1: $j_2 = j_1 \pm 2$. For $j_2 = j_1 + 2$, by the Wigner-Eckart theorem, the only non-vanishing coefficient is $c_3$, and there are no further constraints. And the case $j_2 = j_1 - 2$ is similar. This case leads to the chain representations

$$
\begin{aligned}
&\cdots \to (j) \to (j + 1) \to (j + 2) \to \cdots, \\
&\cdots \to (j) \to (j - 1) \to (j - 2) \to \cdots
\end{aligned}
\tag{3.20}
$$

2. Case 2: $j_2 = j_1 \pm 1$. For $j_2 = j_1 + 1$, the non-vanishing coefficients are $c_2, c_3$, and the equation (3.17) gives a linear relation of $c_2$ and $c_3$

$$
\begin{aligned}
c_1 &= j_1 c_2 - (j_1 + 2) c_3 = 0, \quad &&\text{for } j_1 \geq 1/2, \\
c_1 &= c_2 = c_3 = 0, \quad &&\text{for } j_1 = 0.
\end{aligned}
\tag{3.21}
$$

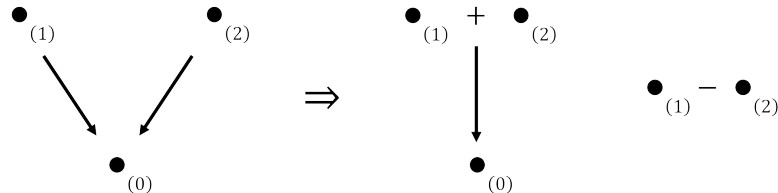

Figure 8: Basis change for a $2d$ GCA representation to chain representations.

For $j_2 = j_1 - 1$ we have

$$
\begin{aligned}
c_3 &= (j_1 - 1)c_1 - (j_1 + 1)c_2 = 0, \quad \text{for } j_1 \geq 3/2, \\
c_1 &= c_2 = c_3 = 0, \quad\quad\quad\quad\quad\quad\; \text{for } j_1 = 1.
\end{aligned}
\tag{3.22}
$$

3. Case 3: $j_2 = j_1$. The equation (3.17) gives a set of linear relations for $c_i$'s

$$
\begin{aligned}
(2j_1 - 1)c_1 &= c_2 + (2j_1 + 3)c_3, \quad \text{for } j_1 \geq 1, \\
c_1 &= c_2 + 4c_3 = 0, \quad\quad\quad\;\; \text{for } j_1 = 1/2, \\
c_1 &= c_2 = 0, \quad\quad\quad\quad\quad\;\; \text{for } j_1 = 0.
\end{aligned}
\tag{3.23}
$$

Notice that in the cases of $j_2 = j_1 \pm 1$ or $j_2 = j_1$, if two of $c_i$'s vanish, the other must vanish due to the linear relations, except the trivial one $(0) \rightarrow (1) \rightarrow (0)$. Hence the allowed chain representations can contain

$$
\cdots \rightarrow (j) \rightarrow (j+1) \rightarrow (j+2) \rightarrow \cdots, \quad \text{or} \quad \cdots \rightarrow (j) \rightarrow (j-1) \rightarrow (j-2) \rightarrow \cdots
\tag{3.24}
$$

In summary, the finite dimensional representations of CCA rotation $\{J, B\}$ are $SO(3)$ spin representations $(j)$ with spin $j$, unidirectionally connected by $B^i \colon (j) \rightarrow (j) \otimes (1)$, and consequently form a net or chain structure. The net representations are complicated and lack of limits, while the possible chain representations must take the following patterns:

**rank 2**

$$
\begin{aligned}
&(j) \rightarrow (j+1), \\
&(j) \rightarrow (j), \quad j \neq 0, \\
&(j) \rightarrow (j-1).
\end{aligned}
\tag{3.25}
$$

**rank 3 or more**

$$
\begin{aligned}
&(0) \rightarrow (1) \rightarrow (0), \\
\cdots \rightarrow &(j) \rightarrow (j+1) \rightarrow (j+2) \rightarrow \cdots, \\
\cdots \rightarrow &(j) \rightarrow (j-1) \rightarrow (j-2) \rightarrow \cdots,
\end{aligned}
\tag{3.26}
$$

where the patterns works for all possible values of $j \in \{0\} \cup \mathbb{Z}_+/2$. Note that the rank-2 case $(0) \rightarrow (0)$ is exceptional, because the representation matrices of CCA rotation $\{J, B\}$ are all zero matrices so that this representation reduces to two decoupled rank-1 $(0)$ representations.

The discussions above applies to all $d \geq 3$ CCA cases. However, the 2-dimensional case is special since the CCA rotation of $2d$ CCA or $2d$ GCA is simply $\{B^1\}$. Thus the theorem 1 can not be applied here and there exist finite dimensional representations with non-zero boost charges for $2d$ CCA or $2d$ GCA [55]. Besides, the representation for every order of the multiplet is trivially an 1-dimensional representation, and it is always possible for a complicated net representation to reduce to the chain representations with the help of basis change. See Figure 8 for an example.

It is worth noticing that the Casimir operators (2.5) acting on the multiplet representations have nonvanishing kernels. This also happens in $2d$ GCFT with the multiplets [25], leading to

$(\Delta, j = 2, r = 3, q = 3)$   $(\Delta, j = 1, r = 3, q = 3)$

$(\Delta, j = 2, r = 3, q = 2)$   $(\Delta, j = 1, r = 3, q = 2)$   $(\Delta, j = 0, r = 3, q = 2)$

$(\Delta, j = 1, r = 3, q = 1)$

Figure 9: The second part of the rank-3 tensor representation of CCA consists of a $\{\Delta, r = 3\}$ representation.

the multi-pole structure in the expansion of the 4-point functions by the conformal blocks. It would be interesting to investigate if such multi-pole structure appears in higher dimensions as well.

## 3.4 Highest weight module

With the representation of the CCA rotation well-defined, the following procedure is standard. A primary operator inserted at $x = 0$ is labeled by $\{\Delta, j, m, r, q\}$ with $\Delta$ and $m$ being the quantum numbers of $D$ and $J = -iJ^{12}$ respectively, $j$ being the label of sub-representation under $SO(3)$, $q$ being the multiplet order, and $r$ being the total rank of the multiplet. A conformal family (highest weight module) of finite dimensional representation is thus

$$
\begin{aligned}
\text{primary: } & \mathcal{O}^{(m,q)}, \quad [D, \mathcal{O}^{(m,q)}] = \Delta \mathcal{O}^{(m,q)}, \quad [J, \mathcal{O}^{(m,q)}] = m\mathcal{O}^{(m,q)}, \quad [K^\mu, \mathcal{O}^{(m,q)}] = 0, \\
\text{descendants: } & [P^\mu, \mathcal{O}^{(m,q)}] = \partial^\mu \mathcal{O}^{(m,q)}, \\
\text{spin index: } & [J^\pm, \mathcal{O}^{(m,q)}] \propto \mathcal{O}^{(m\pm 1,q)}, \\
\text{multiplet index: } & [B^3, \mathcal{O}^{(m,q)}] \propto \mathcal{O}^{(m,q-1)}, \quad [B^\pm, \mathcal{O}^{(m,q)}] \propto \mathcal{O}^{(m\pm 1,q-1)}.
\end{aligned}
\tag{3.27}
$$

For example, the second part of the rank-3 tensor representation in Figure 5 can be re-labeled in the notation introduced here as in Figure 9.

We reorganize the generators for simplicity

$$
J^\pm = \frac{1}{\sqrt{2}}(\mp J^{23} + iJ^{31}), \qquad G^\pm = \frac{1}{\sqrt{2}}(iG^1 \pm G^2), \quad G \in \{P, K, B\},
\tag{3.28}
$$

with the commutation relations

$$
\begin{aligned}
&[J, J^\pm] = \pm J^\pm, \quad [J^+, J^-] = J, \\
&[J, G^\pm] = \pm G^\pm, \quad [J, G^3] = 0, \\
&[J^+, G^+] = 0, \quad [J^+, G^3] = G^+, \quad [J^+, G^-] = G^3, \\
&[J^-, G^+] = G^3, \quad [J^-, G^3] = G^-, \quad [J^-, G^-] = 0.
\end{aligned}
\tag{3.29}
$$

And in the following table, we list how the symmetry generators change the quantum numbers.

Table 3: The way that symmetry generators change the quantum numbers.

|  | $P^0$ | $P^\pm$ | $P^3$ | $K^0$ | $K^\pm$ | $K^3$ | $J^\pm$ | $B^\pm$ | $B^3$ |
|---|---|---|---|---|---|---|---|---|---|
| $\Delta$ | $\Delta+1$ | $\Delta+1$ | $\Delta+1$ | $\Delta-1$ | $\Delta-1$ | $\Delta-1$ | $\Delta$ | $\Delta$ | $\Delta$ |
| $m$ | $m$ | $m\pm 1$ | $m$ | $m$ | $m\pm 1$ | $m$ | $m\pm 1$ | $m\pm 1$ | $m$ |
| $q$ | $q$ | $q$ | $q$ | $q$ | $q$ | $q$ | $q$ | $q-1$ | $q-1$ |

It is obvious that the multiplets also appear at the descendent level even if the primary is not multiplet, since $P^\mu$ naturally forms a vector representation and the tensor product with multiplet structure is still a multiplet.

Finally, the local operators at the point $x^\mu$ can then be defined as

$$\mathcal{O}(x) = U(x)\mathcal{O}(0)U(x)^{-1}, \quad U = \exp(x^\mu P^\mu). \tag{3.30}$$

The action of the generators on the local operators $\mathcal{O}(x)$ can be further evaluated by using the BCH formula, and the conformal family forms an induced representation of the CCA.

## 3.5 Representation and local operators of GCFT

As indicated earlier, since GCA rotation has exactly the same structure as the one in CCA, all the discussion about the representations of the CCA rotations and the local operators in CCFT apply to higher dimensional GCA and GCFT as well. By the Theorem 1, we know that the finite dimensional representations of GCA rotation must have the same multiplet structures and obey the same constraints. And then, we can define the highest weight modules and the local operators in GCFT.

The difference from the CCA case only appears when regarding the tensor representations. One finds that the tensor structures of GCA should be similar, but the covariant tensors of the GCA rotation become the contravariant tensors of the CCA rotation. For example, the covariant and contravariant vector representations in CCA and GCA are respectively

$$\begin{aligned} \text{CCA covariant vector: } (1) \to (0), \quad \text{contravariant vector: } (0) \to (1), \\ \text{GCA covariant vector: } (0) \to (1), \quad \text{contravariant vector: } (1) \to (0). \end{aligned} \tag{3.31}$$

This is not surprising if considering the covariant vectors and contravariant vectors in Euclidean space-time with explicit dependence on the speed of light $c$

$$x^\mu = (ct, \vec{x}), \qquad x_\mu = (t/c, \vec{x}), \quad \text{with } g = \text{diag}(c^2, 1, \dots). \tag{3.32}$$

It is obvious that the covariant vectors of the GCA rotation $x^\mu|_{c\to\infty}$ transform similarly as the contravariant vectors of the CCA rotation $x_\mu|_{c\to 0}$.

It is convenient to define the dual representation $\rho^{\text{dual}}$ of a given representation $\rho$. The dual representation $\rho^{\text{dual}}$ has similar structure but inverse direction to $\rho$, showing the inverse actions of $B$'s. For example, take $\rho = (1) \to (0)$, then $\rho^{\text{dual}} = (0) \to (1)$. The representation matrices are related as

$$\begin{aligned} \mathbf{J}_\rho = \mathbf{J}_{\rho^{\text{dual}}}, \qquad \mathbf{J}_\rho^+ = \mathbf{J}_{\rho^{\text{dual}}}^+, \qquad \mathbf{J}_\rho^- = \mathbf{J}_{\rho^{\text{dual}}}^-, \\ \mathbf{B}_\rho^3 = -(\mathbf{B}_{\rho^{\text{dual}}}^3)^\dagger, \quad \mathbf{B}_\rho^+ = (\mathbf{B}_{\rho^{\text{dual}}}^-)^\dagger, \quad \mathbf{B}_\rho^- = (\mathbf{B}_{\rho^{\text{dual}}}^+)^\dagger. \end{aligned} \tag{3.33}$$

One can easily check this result by plugging in (3.13), (3.14) and (3.15).[12]

Therefore, we say the representation of CCA and GCA rotation are dual to each other: the covariant CCA tensor representations $\rho_{\text{CCA}}$ are equivalent to the dual representations of covariant GCA tensor representation $\rho_{\text{GCA}}$: $\rho_{\text{CCA}} \cong (\rho_{\text{GCA}})^{\text{dual}}$, and vice versa. This feature is a result of the relation between two fiber bundle structures [31], i.e., $\mathcal{C} = \mathbb{R}^{d-1} \times \mathbb{R}^1$ for Carrollian case and $\mathcal{N} = \mathbb{R}^1 \times \mathbb{R}^{d-1}$ for Galilean case.

---

[12]For the representations containing $(j) \to (j)$ structure, the relation on the $\mathbf{B}$ matrices differs by a minus sign due to our convention. We can multiply $i$ in (3.14) to set all the $\mathbf{B}$ matrices fit in the relation (3.33).

## 3.6 Relation to scale-spin representation

In this subsection, we discuss the relation between the scale-spin representations proposed in [54] for GCFT and [36] for CCFT and the multiplet representations we constructed. As the scale-spin representations are actually similar in GCFT and CCFT, we here focus on the CCFT case.

The scale-spin representation is defined by

$$[B_k, \Phi] = a\varphi + b\sigma_k \chi + \tilde{b}\tilde{\sigma}_k \phi + sA_k + rA_t \delta_{\Phi k} + \cdots \tag{3.34}$$

The notation in this subsection follows the original paper, where $\varphi$ is the scalar field, $(\phi, \chi)$ are fermionic fields, $(A_t, A_i)$ are vector fields, "$\cdots$" represents possible higher spin fields, and $\delta_{\Phi k}$ means the possible tensor index of $\Phi$ being equal to $k$.

All the representations in (3.34) can be found in the multiplet representations, and furthermore, the theorem 1 indicates that $[B_k, \Phi] = \cdots + 0\,\Phi + \ldots$ since the diagonal blocks of $B$ matrices are vanishing blocks.

- The scalar $\varphi$ is trivially identified with the scalar representation here with $[B^i, \varphi] = 0$, and $\{a = 0, b = \tilde{b} = 0, s = r = 0, \ldots\}$.

- The spinor representations, although not being introduced in detail above, have the same restrictions. One may set $j$'s to be half integers, and the above discussions immediately gives the corresponding matrix blocks and other constraints. Since the $SO(3)$ representation $\left(\frac{1}{2}\right) \otimes (1) = \left(\frac{1}{2}\right) \oplus \left(\frac{3}{2}\right)$, $\left(\frac{1}{2}\right) \to \left(\frac{1}{2}\right)$ is obviously a representation, which the fermionic fields $\phi$ and $\chi$ are identified with the multiplet

$$
\begin{array}{ll}
\phi \in \left(\dfrac{1}{2}\right) & \\
\quad \downarrow & \phi : a = 0,\ b = 0,\ \tilde{b} = \dfrac{1}{2},\ s = r = 0, \ldots, \\
\chi \in \left(\dfrac{1}{2}\right) & \chi : a = 0,\ b = \tilde{b} = 0,\ s = r = 0, \ldots
\end{array}
\tag{3.35}
$$

- There is no difference between the covariant and the contravariant vector representations of $SO(4)$. However, in taking ultra- or non-relativistic limit, the covariant and contravariant vector representations behave very differently and not surprisingly, there are two corresponding choices when re-scaling the vector fields $A_\mu$, leading to electric and magnetic sectors. The electric sector is identified with the covariant vector representation $(1) \to (0)$

$$
\textbf{Electric sector:} \quad
\begin{array}{ll}
A_i \in (1) & \\
\quad \downarrow & A_i : a = 0,\ b = \tilde{b} = 0,\ s = 1, r = 0, \ldots, \\
A_t \in (0) & A_t : a = 0,\ b = \tilde{b} = 0,\ s = r = 0, \ldots
\end{array}
\tag{3.36}
$$

And the magnetic sector is identified with the contravariant vector representation $(0) \to (1)$

$$
\textbf{Magnetic sector:} \quad
\begin{array}{ll}
A_t \in (0) & \\
\quad \downarrow & A_t : a = 0,\ b = \tilde{b} = 0,\ s = 0, r = 1, \ldots, \\
A_i \in (1) & A_i : a = 0,\ b = \tilde{b} = 0,\ s = r = 0, \ldots
\end{array}
\tag{3.37}
$$

In conclusion, the finite dimensional scale-spin representation perfectly fits in the multiplet representations. Besides, the scale-spin representation should further obey the constraint $\xi = 0$ for finite dimensional representations.

# 4 Correlation Functions in 4d CCFT

In this section, we study the correlation functions in 4d CCFT. As one should expect, the time coordinate and the spacial coordinates behave differently in the correlators, due to the special space-time structure. In many cases, the correlators are independent of the time coordinates, and will be referred to as the trivial correlators. Otherwise they are called non-trivial correlators.

The Carrollian and Galilean conformal symmetries are powerful enough to fix the structures of two-point and three-point correlation functions, similar to the conformal symmetry. Due to the difficulties caused by the sophisticated representation structures, the correlators are not easy to calculate directly by using the constraints of the symmetry. To make things worse, the standard shortcuts in computing the correlators in the usual CFT are not applicable. Firstly, the representations of CCFT/GCFT are reducible, which means we cannot use the Schur lemma to read the selection rules on the representations. Secondly, the embedding formalism seems too complicated to use. Therefore, in the following we will take the brute-force approach and go through all the tedious calculations.

Our method of calculating correlation functions is to use the Ward identities. Supposing the uniqueness of the vacuum and the invariance of the vacuum under the symmetry transformations, we have the Ward identity of the symmetry generator $G$

$$0 = \langle [G, \mathcal{O}_1] \mathcal{O}_2 \dots \rangle + \langle \mathcal{O}_1 [G, \mathcal{O}_2] \dots \rangle + \dots \tag{4.1}$$

As we have already defined the local operators in (3.30), we get the action of $G$ on $\mathcal{O}$ easily

$$[G, \mathcal{O}(x)] = U(x) \left[ U^{-1}(x) G U(x), \mathcal{O} \right] U(x)^{-1}, \tag{4.2}$$

where by using the BCH formula, we have

$$U^{-1}(x) G U(x) \overset{\text{BCH}}{=} G + [G, x^\mu P^\mu] + \frac{1}{2} [[G, x^\mu P^\mu], x^\mu P^\mu] + \dots \tag{4.3}$$

Since $U(x)[P^\mu, \mathcal{O}] U^{-1}(x) = \partial^\mu \mathcal{O}(x)$, the Ward identities finally turn into a set of differential equations of the correlators. Taking some CCA symmetry generators for example, the BCH formula gives

$$
\begin{aligned}
U^{-1}(x) B^i U(x) &= B^i + x^i P^0, \\
U^{-1}(x) K^0 U(x) &= K^0 - 2x^i B^i - x^i x^i P^0, \\
U^{-1}(x) K^i U(x) &= K^i + 2(x^i D + t B^i + x^j J^{ij}) + 2x^i x^\mu P^\mu - (x^j x^j) P^i,
\end{aligned}
\tag{4.4}
$$

which lead to the partial differential equations (PDEs). The full set of Ward identities are:

$$
\begin{aligned}
W(P^\mu) &\equiv (\partial_1^\mu + \partial_2^\mu) \langle \mathcal{O}_1 \mathcal{O}_2 \rangle = 0, \\
W(D) &\equiv (x_1^\mu \partial_1^\mu + x_2^\mu \partial_2^\mu) \langle \mathcal{O}_1 \mathcal{O}_2 \rangle + \Delta_1 \langle \mathcal{O}_1 \mathcal{O}_2 \rangle + \Delta_2 \langle \mathcal{O}_1 \mathcal{O}_2 \rangle = 0, \\
W(J^{ij}) &\equiv ((x_1^i \partial_1^j - x_1^j \partial_1^i) + (x_2^i \partial_2^j - x_2^j \partial_2^i)) \langle \mathcal{O}_1 \mathcal{O}_2 \rangle + \langle (J^{ij} \mathcal{O}_1) \mathcal{O}_2 \rangle + \langle \mathcal{O}_1 (J^{ij} \mathcal{O}_2) \rangle = 0, \\
W(B^i) &\equiv (x_1^i \partial_{t_1} + x_2^i \partial_{t_2}) \langle \mathcal{O}_1 \mathcal{O}_2 \rangle + \langle (B^i \mathcal{O}_1) \mathcal{O}_2 \rangle + \langle \mathcal{O}_1 (B^i \mathcal{O}_2) \rangle = 0, \\
W(K^0) &\equiv (-x_1^i x_1^i \partial_{t_1} - x_2^i x_2^i \partial_{t_2}) \langle \mathcal{O}_1 \mathcal{O}_2 \rangle - 2x_1^i \langle (B^i \mathcal{O}_1) \mathcal{O}_2 \rangle - 2x_2^i \langle \mathcal{O}_1 (B^i \mathcal{O}_2) \rangle = 0, \\
W(K^i) &\equiv ((2x_1^i x_1^\mu \partial_1^\mu - x_1^j x_1^j \partial_1^i) + (2x_2^i x_2^\mu \partial_2^\mu - x_2^j x_2^j \partial_2^i)) \langle \mathcal{O}_1 \mathcal{O}_2 \rangle \\
&\quad + 2(\Delta_1 x_1^i \langle \mathcal{O}_1 \mathcal{O}_2 \rangle + t_1 \langle (B^i \mathcal{O}_1) \mathcal{O}_2 \rangle + x_1^j \langle (J^{ij} \mathcal{O}_1) \mathcal{O}_2 \rangle) \\
&\quad + 2(\Delta_2 x_2^i \langle \mathcal{O}_1 \mathcal{O}_2 \rangle + t_2 \langle \mathcal{O}_1 (B^i \mathcal{O}_2) \rangle + x_2^j \langle \mathcal{O}_1 (J^{ij} \mathcal{O}_2) \rangle) = 0.
\end{aligned}
\tag{4.5}
$$

Solving these PDEs gives the constraints on the correlators. In particular, for the 2-point and 3-point correlators, since the number of degrees of freedom in them are less than the number of the generators, the Ward identities can fix them completely, just as what happened in CFT.

Different from the usual CFT case, the correlators in CCFT often present multiple-level structure, due to the operators belong to some sophisticated indecomposable representations, as shown in the last section. The correlators can be classified by the levels with respect to the orders in multiplet representations. We will discuss such multiple-level structure in the two- and three-point functions carefully.

Moreover, as there is no selection rule on the representations, the 2-pt correlators for the operators in different representations are generally not vanishing. And generically one can not diagonalize these correlators by changing basis or absorb the coefficients by redefining the operators. Consequently the 2-pt coefficients are generally not fixed by the symmetry. Similarly, in the 3-pt correlators there could be multiple 3-pt coefficients as well. It should be point out that if one try to bootstrap CCFT, the propagating operator may be in various representations since there is no selection rule.

It should be stressed that the correlators are defined as the vacuum expectation values of operators

$$\langle \mathcal{O}_1 \mathcal{O}_2 \cdots \rangle \equiv \langle \text{vac} | \mathcal{O}_1 \mathcal{O}_2 \cdots | \text{vac} \rangle. \tag{4.6}$$

Because there has not been a quantization scheme which admits the operator-state correspondence, by this step we cannot interpret the correlators as the inner products of states. The issue of quantization is subtle and we leave this for further study.

In the remaining parts of this section, we discuss the correlators in 4d CCFT. We mainly focus on the 2-pt functions of the operators in chain representations, which already present some novel features. The discussions on the 2-pt correlators of the operators in net representations become tedious due to complicated structures of the representations. Moreover we present some observations on the 3-pt correlators of the operators in chain representation.

## 4.1  Correlators for singlet representations

In this subsection we discuss the 2-pt correlators of the singlets in 4d CCFT, and it turns out that there are two types of solutions of the conformal Ward identities. One of them depends only on the spatial coordinates and is the same as the 2-pt function in $\text{CFT}_3$. While the other is proportional to $\delta^{(3)}(\vec{x}_{12})|t_{12}|^\alpha$. In this paper, we focus on the structure of power law correlators, and the discussions of $\delta$ correlators are similar.

As an illustration, we consider the 2-pt functions of scalar operators $\mathcal{O}_n$, $n = 1, 2$ with scaling dimensions $\Delta_n$ and satisfying $[B^i, \mathcal{O}_n] = [J^{ij}, \mathcal{O}_n] = [K^\mu, \mathcal{O}_n] = 0$. The covariance under the translation, the spatial rotation and the bosonic symmetry implies

$$\langle \mathcal{O}_1(t_1, \vec{x}_1) \mathcal{O}_2(t_2, \vec{x}_2) \rangle = G(t, r), \tag{4.7}$$

where $t = |t_{12}|$, $r = |\vec{x}_{12}|$. The Ward identities of $\{B^i, K^0\}$ are

$$\begin{aligned} B^i: & \quad (x_1^i \partial_{t_1} + x_2^i \partial_{t_2}) G(t, r) = 0, \\ K^0: & \quad (\vec{x}_1^2 \partial_{t_1} + \vec{x}_2^2 \partial_{t_2}) G(t, r) = 0, \end{aligned} \tag{4.8}$$

and interestingly the above equations have two independent solutions[13]

$$G(t, r) = c_1 f(r) + c_2 \delta^{(3)}(\vec{x}_{12}) g(t). \tag{4.9}$$

---

[13]The existence of two kinds of solutions was also noticed in [41].

Then the Ward identity of $D$ gives

$$G(t,r) = c_1 \frac{1}{r^{\Delta_1 + \Delta_2}} + c_2 \delta^{(3)}(\vec{x}_{12}) \frac{1}{t^{\Delta_1 + \Delta_2 - 3}}. \tag{4.10}$$

If $c_1 \neq 0$, the Ward identities of $K^i$ will force $\Delta_1 = \Delta_2$, and the resulting 2-pt function coincides with the scalar 2-pt function in $CFT_3$. If $c_1 = 0$, there is no further constraint on $\Delta_1$ and $\Delta_2$.

Similar to the discussion in $2d$ [26], the two types of solutions in (4.10) can be understood in the following concrete models:

- $c_1 \neq 0, c_2 = 0$: the bilocal action of free scalar

$$S = \int d^3\vec{x}_1 dt_1 d^3\vec{x}_2 dt_2 \, \phi(\vec{x}_1, t_1) |\vec{x}_{12}|^{2\Delta_\phi - 8} \phi(\vec{x}_2, t_2). \tag{4.11}$$

This free action is Carrollian conformal invariant due to the chosen exponent of $|\vec{x}_{12}|$. By the field redefinition $\Phi(\vec{x}) := \int dt \, \phi(\vec{x}, t)$ we can eliminate the degeneracy and get back to the action of generalized free scalar in $3d$,

$$S = \int d^3\vec{x}_1 d^3\vec{x}_2 \, \Phi(\vec{x}_1) |\vec{x}_{12}|^{2\Delta_\phi - 8} \Phi(\vec{x}_2), \tag{4.12}$$

with $\Delta_\Phi = \Delta_\phi - 1$. The two-point function is

$$\langle \Phi\Phi \rangle = c_1 |\vec{x}_{12}|^{2 - 2\Delta_\phi}, \tag{4.13}$$

where $c_1$ is an unimportant constant. This fits into the first solution in (4.10) with $\Delta_1 = \Delta_2 = \Delta_\phi - 1$.

- $c_1 = 0, c_2 \neq 0$: the Carrollian free scalar [36, 40, 59] with the action

$$S = \int d^3\vec{x} dt \, \phi \partial_t^2 \phi. \tag{4.14}$$

The 2-pt function can be calculated by the path integral since the theory is free,

$$\langle \phi\phi \rangle = c_2 t \delta^{(3)}(\vec{x}_{12}), \tag{4.15}$$

where $c_2$ is an unimportant constant. This fits into the second solution in (4.10) with $\Delta_1 = \Delta_2 = 1$.[14]

In [25–27], the first type of correlation functions $|\vec{x}_{12}|^{-2\Delta}$ in $2d$ are shown to provide convergent and associative OPEs in concrete models. This suggests that the first type of correlation functions is suitable for discussing OPE and conformal block expansion. The second type of correlation functions $\delta^{(3)}(\vec{x}_{12})t^{-\Delta_1 - \Delta_2 + 3}$ appears in the Carrollian description of celestial CFTs [60, 61]. However, due to the lack of analyticity and selection rule on $\Delta$, it can be harder to establish the OPE relations between local operators. The action (4.14) is the electric sector of free scalar in [40], while there exists the magnetic sector of free scalar. The 2-point functions of the magnetic version take the form of $\delta^{(3)}(\vec{x}_{12})$ as well. A more explicit discussion on the correlators of Carrollian electric/magnetic sector of free scalar will appear in an upcoming paper.

In the rest of the paper we will set $c_2 = 0$ and focus on the first type of 2-pt functions for all other representations as well. We leave the discussion on the second type for further

---

[14]For the $2d$ Carrollian free scalar model, there is another quantization scheme which leads to correlation functions fitting into the first solution in (4.10) [42].

consideration. Repeating the above discussion, the 2-pt functions of spinning singlet operators $\mathcal{O}_n$ with $SO(3)$ spin $j_n$ coincide with those in $\text{CFT}_3$

$$\langle \mathcal{O}_1(x_1)\mathcal{O}_2(x_2)\rangle = \frac{C_{12}I_2(\vec{x}_1,\vec{x}_2)}{|\vec{x}_{12}|^{2\Delta_1}}\delta_{\Delta_1,\Delta_2}\delta_{j_1,j_2}, \qquad (4.16)$$

where the tensor indices have been suppressed and the $I_2$ is the 2-pt tensor structure.[15] The 3-pt functions of the singlet operators are also independent of time due to the Ward identities of $\{P^0, B^i, K^0\}$, and further coincide with the ones in $\text{CFT}_3$:

$$\langle \mathcal{O}_1(x_1)\mathcal{O}_2(x_2)\mathcal{O}_3(x_3)\rangle = \frac{C_{123}\,I_3}{|\vec{x}_{12}|^{\Delta_{123}}|\vec{x}_{23}|^{\Delta_{231}}|\vec{x}_{13}|^{\Delta_{312}}}, \quad \Delta_{ijk} = \Delta_i + \Delta_j - \Delta_k, \qquad (4.17)$$

where $I_3$ is 3-pt tensor structure and $C_{123}$ is the 3-pt coefficient.

One may suspect that the $\text{CCFT}_4$ correlators of the first type all reduce to those in $\text{CFT}_3$, and the answer is no for two reasons. Firstly for chain and net multiplets there are non-trivial temporal dependence in the correlators, as we will show in the following subsections. Secondly even for the singlets, the temporal dependence can appear in the higher-point correlators. When calculating the correlators, the Ward identities of $\{P^0, B^i, K^0\}$ give five differential equations of time coordinates. With $[B^i, \mathcal{O}] = 0$, the correlators with less than six insertions of the singlets would not depend on time coordinates, and other Ward identities make them further degenerate into the correlators of $\text{CFT}_3$. However, for the 6-pt and even higher-point correlators, there are only five independent differential equations of $t$'s, thus the $t$-dependence certainly appears in the conformal invariants in the "stripped correlators" discussed in section 2.4. For the kinematic part, since the differential equations from the Ward identities of $\{P^0, B^i, K^0\}$ are all of the forms $(\sum_i g(\vec{x}_i)\partial_{t_i})f_{\text{n-pt}} = 0$, the dependence on $t$'s does not show up in the kinematic parts. This means that the kinematic parts of the correlators of the singlet operators are exactly the same as the ones in the correlators of $\text{CFT}_3$.[16] The appearance of time-like dependence tells us we cannot treat singlet operators exactly the same as the $\text{CFT}_3$ operators.

## 4.2 Two-point functions for chain representations

It turns out that the correlators of net representations are rather sophisticated due to the complicated structures of limitless net representations. Thus we first focus on the case of chain representations here, and we will briefly discuss the net representations in section 4.3.

The calculation is nothing more but solving the Ward identities. It is easier to start from the lowest-order chain operators and to carry out the calculation level by level, since the correlators of higher-order chain operators depend on the lower-order ones by the action of $B$ and $K$ generators. Recall that the derivation of two-point correlators of the singlets is relatively easy, because the singlets are annihilated by the $B$ generators, so that the correlators are independent of the time coordinates and are reduced to the ones in $\text{CFT}_3$. For the same reason, the calculation for the lowest-order chain operators could also be reduced to the ones in $\text{CFT}_3$ since the $B$ generators also annihilate them. Besides, if the correlators of the lowest-order chain operators do not satisfy the selection rule, they are limited to be vanishing. Moreover, even though in the sense of operators $B^i\mathcal{O}^{(q)} \propto \mathcal{O}^{(q-1)} \neq 0$, the $B$ generators may annihilate the higher operators in the correlators, such as $\langle \dots (B^i\mathcal{O}^{(q)})\dots\rangle \propto \langle \dots \mathcal{O}^{(q-1)}\dots\rangle = 0$, so that

---

[15] For generic $j_1$ and $j_2$, the Ward identities of $J^{ij}$ fix the tensor structure $I^{m_1,m_2}_{j_1,j_2}$ up to a set of relative coefficients. We leave the detailed discussion on the relative coefficients for $I^{m_1,m_2}_{j_1,j_2}$ to Appendix B.

[16] The term "kinematic part" here refer to the part totally fixed by the Ward identities, in contrast to the definition in some literature where the kinematic part is defined up to multiplying some conformal invariants. In this sense, the kinematic part of the correlators for $\text{CCFT}_4$ singlets is _exactly_ the same with the correlators for $\text{CFT}_3$ operators.

the corresponding correlators of $\mathcal{O}^{(q)}$ would again behave similarly as the correlators in CFT$_3$. One can apply the same argument recursively before finding the lowest non-zero correlators, and using them one can further build the correlators of the higher-order chain operators.

To make a clearer expression, we introduce the term "level" for the correlators of the multiplets. The level in a $n$-pt correlator is defined as

$$\text{Level} = \sum_{i=1}^{n} q_i - n + 1, \qquad (4.18)$$

where $q_i$ is the order of the $i$-th operator in a multiplet. Then the correlator can be organized by different levels. For example, the lowest level one comes from the correlator of the lowest-order operators in every multiplets, while the highest level one comes from the correlator of the highest-order operators. For a 2-pt correlator of the operators in multiplet representations, we have the following structure:

$$\left\langle \mathcal{O}_1^{(m_1,q_1)}(x_1) \mathcal{O}_2^{(m_2,q_2)}(x_2) \right\rangle = f_{q_1,q_2}^{m_1,m_2}(x_{12}). \qquad (4.19)$$

The lowest-level correlator is of level number 1, corresponding to $f_{1,1}$, and the second lowest level correlator correspond to $f_{1,2}$ and $f_{2,1}$, etc.. For an explicit example, see the levels labeled in (4.22). Thus the general strategy calculating the 2-pt functions of $\mathcal{O}_i$ in the chain representations is as follows:

1. Use the Ward identities of $\{P, D, J\}$ generators to determine the scaling structures $|\vec{x}_{12}|^{-(\Delta_1+\Delta_2)}$ and the $SO(3)$ tensor structures $I$ for each level of the correlators. Note that here we would not impose any selection rules since they come from the Ward identities of $K$ generators. This means that there exist non-zero tensor structures for $j_1 \neq j_2$, and the relative coefficients in the tensor structures are generally not determined;

2. Starting from the lowest level correlators, use the Ward identities of $B$ generators to make them independent of $t$ coordinates, and further apply the Ward identities of $K$ generators to read the selection rules and fix the relative coefficients in the tensor structure $I$. If the lowest-level correlator do not satisfy the selection rule, the one-level-higher correlator would behave similarly as the lowest one. If the lowest level correlators are vanishing, repeat this procedure for one higher level correlators until find out the first non-zero correlators, which are independent of $t$ coordinates and have the same structures with the ones in CFT$_3$;

3. Use $B$ generators to find the power law of $(t_{12}/|\vec{x}_{12}|)^n$ structures[17] for the higher-level correlators. Then impose the $K$ Ward identities to check the selection rules for the solutions;

4. For some cases, the solutions to the Ward identities of the higher-level correlators are to set the coefficients of the lower-level correlators to zero. In these cases, these higher-level correlators become the lowest non-zero level, and we return to step 2 for these correlators. If the Ward identities are satisfied for all level correlators, we are done with the 2-pt correlators of the operators in given representations.

In what follows next, we consider some explicit examples of 2-pt functions for the short chain representations to get the restriction and the selection rules. Some details of the calculations are omitted, and the interested readers can find them in Appendix B.

---

[17]There may exist the solutions with the polynomials of $t_{12}/|\vec{x}_{12}|$ for the operators in some specific representations. However, the polynomial solutions can always be modified into the power laws with suitable change of the basis. The detailed discussions can be found in section 4.3.

### 4.2.1 Trivial 2-pt correlators

We start from an example where two primary operators are all in the vector representation. After considering the constraints from $P$ generators, the 2-point functions have the forms

$$\left\langle \mathcal{O}_1^{(m_1,q_1)}(x_1)\mathcal{O}_2^{(m_2,q_2)}(x_2)\right\rangle = f_{q_1,q_2}^{m_1,m_2}(x_{12}), \qquad \mathcal{O}_1, \mathcal{O}_2 \in (1) \to (0), \quad x_{12} = x_1 - x_2. \quad (4.20)$$

Hereafter, we will frequently omit some quantum numbers in the operators and the correlators for simplicity. Following the strategies above, the calculation on the lowest-level correlator $f_{1,1}^{0,0}$ requires that $f_{1,1}^{0,0} = C^{(0,0)}|\vec{x}_{12}|^{-2\Delta}$ with the constraint $\Delta_1 = \Delta_2 = \Delta$, and $C^{(j_1,j_2)}$ denoting the 2-pt coefficient.

Next, considering $f_{1,2}^{0,0}$, the Ward identity of $B^+$ and $B^3$ give rise to

$$\begin{aligned} B^+: &\quad (ix_{12}^1 + x_{12}^2)\partial_{t_{12}}f_{1,2}^{0,0} = 0, \\ B^3: &\quad x_{12}^3\partial_{t_{12}}f_{1,2}^{0,0} + f_{1,1}^{0,0} = 0. \end{aligned} \quad (4.21)$$

The first equation shows that $f_{1,2}^{0,0}$ is independent of time coordinates, and thus the second equation yields $f_{1,1}^{0,0} = 0$, leading to the 2-pt coefficient $C^{(1,1)} = 0$. Further, although the actions of $B$'s on $\mathcal{O}_2^{(m_2,2)}$ are not vanishing, $[B, \mathcal{O}_2^{(m_2,2)}] \propto \mathcal{O}_2^{(m_2,1)}$, the actions of $B$'s in the correlators are vanishing as $\left\langle \mathcal{O}_1^{(0,1)}(B^i\mathcal{O}_2^{(m_2,2)})\dots\right\rangle = 0$. This results in $f_{1,2}^{0,m_2}$ behaving as the correlators in CFT$_3$. Moreover, the selection rule $j_1 = j_2$ requires $f_{1,2}^{0,m_2} = 0$. The same argument applies to the other level-2 correlator and leaves $f_{2,1}^{m_1,0} = 0$. Finally, $f_{2,2}^{m_1,m_2}$ is just the 2-pt function of spin-1 operators in CFT$_3$. In the resulting 2-pt functions only the highest level survive and reduce to the ones in CFT$_3$:

$$\begin{aligned} &\text{Level 3:} &f_{2,2}^{m_1,m_2} &= \frac{C\, I_{1,1}^{m_1,m_2}}{|\vec{x}_{12}|^{2\Delta}}, & &(1) &(1) \\ & & & & \mathcal{O}_1 \in &\downarrow &\mathcal{O}_2 \in \downarrow \\ &\text{Level 2:} &f_{1,2}^{0,m_2} = 0, \quad f_{2,1}^{m_1,0} &= 0, & &(0) &(0) \\ &\text{Level 1:} &f_{1,1}^{0,0} &= 0, & &\text{with } \Delta_1 = \Delta_2 = \Delta. \end{aligned} \quad (4.22)$$

Here $C$ is the 2-pt coefficient, and $I_{j_1,j_2}^{m_1,m_2}$ is the 2-pt tensor structure with spin $j_1$ and $j_2$, whose explicit expression can be found in Appendix B. Here we do not hurry to diagonalize the 2-pt correlators for later convenience.

In the following for a 2-pt correlator, when only its top-level correlators are non-zero and independent of $t$ coordinates, we will call it "trivial", otherwise call it "nontrivial". The above 2-pt correlator is a trival one.

In fact, since there are only two time coordinates $t_1$ and $t_2$ for the 2-pt correlators, one can prove that for the 2-pt correlators being non-trivial, there must be at least one operator in the increasing chain representation, which is of the form $\cdots \to (j) \to (j+1) \to \dots$. The proof is as follows. Consider a specific correlator $f_{1,2}^{0,0}$. If there is one $B$ generator annihilating both operators in the sense of acting on the correlators, the Ward identity of this $B$ together with the one of $P^0$ ensure that the correlator is independent of $t$ coordinates and reduces to that in CFT$_3$. If furthermore there exist another $B$ generator relating this correlator to a lower-level one, say $B^3$ relating $f_{1,2}^{0,0}$ to $f_{1,1}^{0,0}$, it is clear by its Ward identity that this lower-level correlator must vanish, i.e. $f_{1,1}^{0,0} = 0$ and thus $C^{(0,0)} = 0$. Such kind of situations is not rare. In fact for the 2-pt functions of the operators both in the non-increasing chain representations, there always exist some $B$ generators which allow us to repeatedly use the above argument

and find the correlators trivial. This means that the following correlators for the rank-2 chain representations are all trivial

$$\begin{cases} \mathcal{O}_1 \in (1) \to (0), \\ \mathcal{O}_2 \in (1) \to (1), \end{cases} \qquad \begin{cases} \mathcal{O}_1 \in (1) \to (1), \\ \mathcal{O}_2 \in (1) \to (1). \end{cases} \tag{4.23}$$

Besides, this is not the only restriction on non-trivial 2-pt correlators. It turns out that the Ward identities of $B$'s and $K$'s can give more constraints. Consider the following 2-pt correlators

$$\begin{array}{cccc} \vdots & \vdots & & \mathcal{O}_1 \in \cdots \to (j), \\ \text{Level 2:} & f_{1,2}^{m_1,m_2}, & \ldots & \mathcal{O}_2 \in \cdots \to (j-1) \to (j), \\ \text{Level 1:} & f_{1,1}^{m_1,m_2}, & & \text{with } \Delta_1 = \Delta_2 = \Delta. \end{array} \tag{4.24}$$

where $\mathcal{O}_2$ is in an increasing chain. Following the bottom up algorithm, we first find $f_{1,1}^{m_1,m_2} = C^{(j,j)} I_{j,j}^{m_1,m_2} / |\vec{x}_{12}|^{2\Delta}$, with relative coefficients in $I_{j,j}^{m_1,m_2}$ being totally fixed by the Ward identities of $K$'s on $f_{1,1}^{m_1,m_2}$. Secondly, the Ward identities of $B$'s on $f_{1,2}^{m_1,m_2}$ lead to

$$x_{12}^i \partial_{t_{12}} f_{1,2}^{m_1,m_2} + 0 + \sum_m (\mathbf{B}^i)_{m_2,m}^{(j-1)\to j} f_{1,1}^{m_1,m} = 0, \tag{4.25}$$

which requires that $C^{(j,j)} = 0$ and $f_{1,2}^{m_1,m_2}$ is independent of time coordinates. In fact, the 2-pt correlators are non-trivial only if the actions of $B$'s on both operators give non-zero correlators. The proof is rather tedious, and the interested reader can refer Appendix B for details. Finally, due to the selection rule $j_1 = j_2$ for the 2-pt correlators in CFT$_3$, $f_{1,2}^{m_1,m_2} = 0$, resulting in

$$\begin{array}{cccc} \vdots & \vdots & & \mathcal{O}_1 \in \ldots \to (j), \\ \text{Level 2:} & f_{1,2}^{m_1,m_2} = 0, & \ldots & \mathcal{O}_2 \in \cdots \to (j-1) \to (j), \\ \text{Level 1:} & f_{1,1}^{m_1,m_2} = 0, & & \text{with } \Delta_1 = \Delta_2 = \Delta. \end{array} \tag{4.26}$$

The following are two explicit examples of rank-2 multiplets with one operator in an increasing chain. Their 2-pt correlators are both trivial. The first one is

$$\begin{array}{ccccc} \text{Level 3:} & f_{2,2}^{m_1,0} = 0, & & (1) & (0) \\ \text{Level 2:} & f_{1,2}^{m_1,0} = 0, & f_{2,1}^{m_1,m_2} = 0, & \mathcal{O}_1 \in \downarrow, & \mathcal{O}_2 \in \downarrow. \\ \text{Level 1:} & f_{1,1}^{m_1,m_2} = 0, & & (1) & (1) \end{array} \tag{4.27}$$

In this case, all 2-pt correlators are vanishing. The second case is the 2-pt correlators of two contravariant vectors

$$\begin{array}{ccccc} \text{Level 3:} & f_{2,2}^{0,0} = \dfrac{C}{|\vec{x}_{12}|^{2\Delta}}, & & (0) & (0) \\ & & & \mathcal{O}_1 \in \downarrow, & \mathcal{O}_2 \in \downarrow, \\ \text{Level 2:} & f_{1,2}^{m_1,0} = 0, & f_{2,1}^{0,m_2} = 0, & (1) & (1) \\ \text{Level 1:} & f_{1,1}^{m_1,m_2} = 0, & & \text{with } \Delta_1 = \Delta_2 = \Delta. \end{array} \tag{4.28}$$

### 4.2.2 The simplest non-trivial example: covariant and contravariant vectors

The simplest non-trivial case is the correlators of two operators in the vector and contravariant vector representations, respectively:

$$\left\langle \mathcal{O}_1^{(m_1,q_1)}(x_1) \mathcal{O}_2^{(m_2,q_2)}(x_2) \right\rangle = f_{q_1,q_2}^{m_1,m_2}(x_{12}), \qquad \mathcal{O}_1 \in (1) \to (0), \mathcal{O}_2 \in (0) \to (1). \tag{4.29}$$

Following the algorithm, we know that the lowest-level correlator is determined to be vanishing by the selection rule $j_1 = j_2$, i.e $f_{1,1}^{m_1,0} = 0$, and thus the two second-lowest-level correlators $f_{2,1}^{m_1,m_2}$ and $f_{1,2}^{0,0}$ are independent of $t$

$$f_{1,2}^{0,0} = \frac{C^{(0,0)}}{|\vec{x}_{12}|^{2\Delta}}, \qquad f_{2,1}^{m_1,m_2} = \frac{C^{(1,1)} I_{1,1}^{m_1,m_2}}{|\vec{x}_{12}|^{2\Delta}}, \qquad \Delta_1 = \Delta_2 = \Delta. \qquad (4.30)$$

The action of $B$ on the highest-level one requires $C \equiv C^{(0,0)} = C^{(1,1)}$ and

$$f_{2,2}^{m_1,0} = \frac{(C\, t_{12}/|\vec{x}_{12}| I_{1,0}^{m_1} + C_0^{(1,0)} \tilde{I}_{1,0}^{m_1})}{|\vec{x}_{12}|^{2\Delta}}. \qquad (4.31)$$

The Ward identities of $K$'s further require $C^{(1,0)} = 0$. In the end, we have the following nontrivial result

$$\begin{array}{lll}
\text{Level 3:} & f_{2,2}^{m_1,0} = \dfrac{C\, t_{12}/|\vec{x}_{12}|\, I_{1,0}^{m_1}}{|\vec{x}_{12}|^{2\Delta}}, & \begin{array}{cc} (1) & (0) \\ \mathcal{O}_1 \in \downarrow, & \mathcal{O}_2 \in \downarrow, \\ (0) & (1) \end{array} \\
\text{Level 2:} \quad f_{1,2}^{0,0} = \dfrac{C}{|\vec{x}_{12}|^{2\Delta}}, & f_{2,1}^{m_1,m_2} = \dfrac{C\, I_{1,1}^{m_1,m_2}}{|\vec{x}_{12}|^{2\Delta}}, & \\
\text{Level 1:} & f_{1,1}^{0,m_2} = 0, & \text{with } \Delta_1 = \Delta_2 = \Delta.
\end{array} \qquad (4.32)$$

Note that the highest-level correlator $f_{2,2}^{m_1,0}$ is linear in $t_{12}/|\vec{x}_{12}|$.

The discussions above applies to all the correlators of rank-2 operators with the structures (3.25). The non-trivial 2-pt correlators are

$$\begin{array}{lll}
\text{Level 3:} & f_{2,2} = \dfrac{C\, t_{12}/|\vec{x}_{12}|\, I_{j+1,j}}{|\vec{x}_{12}|^{2\Delta}}, & \begin{array}{cc} (j+1) & (j) \\ \mathcal{O}_1 \in \downarrow, & \mathcal{O}_2 \in \downarrow, \\ (j) & (j+1) \end{array} \\
\text{Level 2:} \quad f_{1,2} = \dfrac{C\, I_{j,j}}{|\vec{x}_{12}|^{2\Delta}}, & f_{2,1} = \dfrac{C\, I_{j+1,j+1}}{|\vec{x}_{12}|^{2\Delta}}, & \\
\text{Level 1:} & f_{1,1} = 0, & \text{with } \Delta_1 = \Delta_2 = \Delta,
\end{array} \qquad (4.33)$$

where $f_{1,2} = f_{j,j}^{(\text{CFT})}$ and $f_{2,1} = f_{j+1,j+1}^{(\text{CFT})}$ are the same with the 2-pt correlators in CFT. The details for this result can be found in Appendix B.

### 4.2.3 Longer chains

With the case of rank-2 chains being well discussed, let us now consider the longer chains. The long chain representations have the form of (3.26) which we repeat here for convenience:

$$\begin{array}{c}
(0) \to (1) \to (0), \\
\cdots \to (j) \to (j+1) \to (j+2) \to \cdots, \\
\cdots \to (j) \to (j-1) \to (j-2) \to \cdots
\end{array} \qquad (4.34)$$

We start from the increasing and the decreasing chains and get back to the special case $(0) \to (1) \to (0)$ later. Following the standard algorithm and the previous discussions, we find that the 2-pt correlators for both the operators in the increasing or the decreasing chains are trivial: non-zero for top-level correlator with the highest-order operators having the same spin, and vanishing for all other cases. This leaves us the following possible cases for nontrivial 2-pt correlators:

- Case 1: Two operators whose representations are of entirely _inverse_ pattern;

- Case 2: Two operators whose representations are at least partially _inverse_ in the sense that the representation of one operator have the _inverse_ pattern to the _leading_ sub-sector of the representations of the other operator.

We refer to the first one as the entirely inverse case, and refer to the second one as the partially inverse case. However, there are two exceptions:

- For $\mathcal{O}_1, \mathcal{O}_2 \in (j) \to (j)$, their 2-pt correlator is trivial;

- For $\mathcal{O}_1 \in (j) \to \cdots, \mathcal{O}_2 \in (j) \to \cdots$, their 2-pt correlator is also trivial.

For instance, $\mathcal{O}_i$ in the same non-self-inverse representation belongs to the second exception. Note that the two exceptional cases has the same top sub-sector which has only one $SO(3)$ representation. Therefore we refer to the entirely inverse case or the partially inverse case as the case in which the inverse pattern involves at least two $SO(3)$ representations related by $B$.

For the following example in the entirely inverse case, we have

$$
\text{Level 5:} \qquad f_{3,3}^{m_1,0} = \frac{C\, t_{12}^2/|\vec{x}_{12}|^2\, I_{2,0}^{m_1}}{|\vec{x}_{12}|^{2\Delta}},
$$

$$
\text{Level 4:} \quad f_{2,3}^{m_1,0} = \frac{C\, t_{12}/|\vec{x}_{12}|\, I_{1,0}^{m_1}}{|\vec{x}_{12}|^{2\Delta}}, \qquad f_{3,2}^{m_1,m_2} = \frac{C\, t_{12}/|\vec{x}_{12}|\, I_{2,1}^{m_1,m_2}}{|\vec{x}_{12}|^{2\Delta}},
$$

$$
\text{Level 3:} \quad f_{1,3}^{0,0} = \frac{C}{|\vec{x}_{12}|^{2\Delta}}, \qquad f_{2,2}^{m_1,m_2} = \frac{C\, I_{1,1}^{m_1,m_2}}{|\vec{x}_{12}|^{2\Delta}}, \qquad f_{3,1}^{m_1,m_2} = \frac{C\, I_{2,2}^{m_1,m_2}}{|\vec{x}_{12}|^{2\Delta}}, \tag{4.35}
$$

$$
\text{Level 2:} \qquad f_{1,2}^{0,m_2} = 0, \qquad f_{2,1}^{m_1,m_2} = 0,
$$

$$
\text{Level 1:} \qquad f_{1,1}^{0,m_2} = 0,
$$

$$
\mathcal{O}_1 \in (2) \to (1) \to (0), \quad \mathcal{O}_2 \in (0) \to (1) \to (2), \quad \text{with } \Delta_1 = \Delta_2 = \Delta. \tag{4.36}
$$

The non-zero 2-pt correlators start from the middle level where $j_1 = j_2$, having the same form with those in CFT$_3$. The higher-level ones are of power laws in $t_{12}/|\vec{x}_{12}|$ with the power increasing along with the level. The closed form of the 2-pt correlators for generic $\mathcal{O}_1 \in (j+n) \to \cdots \to (j)$ and $\mathcal{O}_2 \in (j) \to \cdots \to (j+n)$ can be found in (B.36).

Let us turn to a partially inverse case. Consider the operator $\mathcal{O}_1$ in the representation $(1) \to (0)$ and the operator $\mathcal{O}_2$ in the representation $(0) \to (1) \to (2)$. Their 2-pt correlators have the following structure

$$
\begin{aligned}
&\text{Level 4:} \qquad f_{2,3}^{m_1,0} = \frac{C\, t_{12}/|\vec{x}_{12}|\, I_{1,0}^{m_1}}{|\vec{x}_{12}|^{2\Delta}}, && (0) \\
& && (1) \qquad \downarrow \\
&\text{Level 3:} \quad f_{1,3}^{0,0} = \frac{C}{|\vec{x}_{12}|^{2\Delta}}, \quad f_{2,2}^{m_1,m_2} = \frac{C\, I_{1,1}^{m_1,m_2}}{|\vec{x}_{12}|^{2\Delta}}, && \mathcal{O}_1 \in \downarrow, \quad \mathcal{O}_2 \in (1), \\
& && (0) \qquad \downarrow \\
&\text{Level 2:} \qquad f_{1,2}^{m_1,m_2} = 0, \quad f_{2,1}^{m_1,m_2} = 0, && (2) \\
&\text{Level 1:} \qquad f_{1,1}^{m_1,m_2} = 0, && \text{with } \Delta_1 = \Delta_2 = \Delta,
\end{aligned} \tag{4.37}
$$

where the correlators at the top level are exactly the same as (4.32). It turns out that the $(2)$ sub-sector of $\mathcal{O}_2$ gives no contribution. The lower-order sub-sector of the representation giving no contribution is typical for the 2-pt correlators in the partially inverse case.

The requirement that the inverse pattern must start from the leading sub-sectors is necessary and can be easily understood. We build the 2-pt correlators from bottom to top, and the

Ward identities on the higher-level correlators give constraints back to the lower-level ones. If the higher-level sub-sectors does not have non-zero solution, the lower-level sub-sectors are also vanishing, resulting in vanishing 2-pt correlators for the whole representations. For example,

$$
\begin{array}{lll}
\text{Level 4:} & f_{3,2}^{m_1,0} = 0\,, & (2) \\[4pt]
\text{Level 3:} & f_{2,2}^{m_1,0} = 0\,, \quad f_{3,1}^{m_1,m_2} = 0\,, & \downarrow \qquad (0) \\[4pt]
\text{Level 2:} \quad f_{1,2}^{0,0} = 0\,, & f_{2,1}^{m_1,m_2} = 0\,, & \mathcal{O}_1 \in (1)\,, \quad \mathcal{O}_2 \in \;\downarrow\;\,. \\[4pt]
\text{Level 1:} & f_{1,1}^{0,m_2} = 0\,, & \downarrow \qquad (1) \\[4pt]
& & (0)
\end{array} \tag{4.38}
$$

In this example, without considering $f_{3,1}^{m_1,m_2}$ directly, we have $f_{1,1}^{0,m_2} = 0$ at level 1, and $f_{1,2}^{0,0} = C^{(0,0)}|\vec{x}_{12}|^{-2\Delta}, f_{2,1}^{m_1,m_2} = C^{(1,1)}I_{1,1}^{m_1,m_2}|\vec{x}_{12}|^{-2\Delta}$ at level 2, then the Ward identities require $f_{3,1}^{m_1,m_2} = 0$ and $C^{(0,0)} = C^{(1,1)} = 0$, resulting in vanishing 2-pt correlators.

The discussions also apply to the non-vanishing 2-pt correlators with one operator in $\mathcal{O} \in (0) \to (1) \to (0)$ and the other operator in a decreasing chain, as the increasing chains do not have the inverse pattern to $(0) \to (1)$. The special case is the 2-pt correlators of the operators both in $\mathcal{O} \in (0) \to (1) \to (0)$. Using the Ward identities only, we have the following result

$$
\begin{array}{lll}
\text{Level 5:} & f_{3,3}^{0,0} = \dfrac{-2C\,t_{12}^2/|\vec{x}_{12}|^2 + C'}{|\vec{x}_{12}|^{2\Delta}}\,, & \\[10pt]
& & (0) \qquad\quad (0) \\[4pt]
\text{Level 4:} \quad f_{2,3}^{m_1,0} = \dfrac{C\,t_{12}/|\vec{x}_{12}|\,I_{1,0}^{m_1}}{|\vec{x}_{12}|^{2\Delta}}\,, \quad f_{3,2}^{0,m_2} = \dfrac{C\,t_{12}/|\vec{x}_{12}|\,I_{0,1}^{m_2}}{|\vec{x}_{12}|^{2\Delta}}\,, & \downarrow \qquad\quad \downarrow \\[8pt]
& & \mathcal{O}_1 \in (1)\,, \quad \mathcal{O}_2 \in (1)\,, \\[4pt]
\text{Level 3:} \quad f_{1,3}^{0,0} = \dfrac{C}{|\vec{x}_{12}|^{2\Delta}}\,, \quad f_{2,2}^{m_1,m_2} = \dfrac{C\,I_{1,1}^{m_1,m_2}}{|\vec{x}_{12}|^{2\Delta}}\,, \quad f_{3,1}^{0,0} = \dfrac{C}{|\vec{x}_{12}|^{2\Delta}}\,, & \downarrow \qquad\quad \downarrow \\[8pt]
& & (0) \qquad\quad (0) \\[4pt]
\text{Level 2:} & f_{1,2}^{0,m_2} = 0\,, \quad f_{2,1}^{m_1,0} = 0\,, & \text{with } \Delta_1 = \Delta_2 = \Delta\,. \\[4pt]
\text{Level 1:} & f_{1,1}^{0,0} = 0\,, &
\end{array}
$$
$$\tag{4.39}$$

It is obvious that $f_{3,3}^{0,0}$ is a polynomial rather than a power law of $t_{12}/|\vec{x}_{12}|$. But with a change of basis

$$
\begin{array}{ccc}
(0)_3 & & (0)_3 - \dfrac{C'}{2C}(0)_1 \\[6pt]
\downarrow & & \downarrow \\[4pt]
(1)_2 & \longrightarrow & (1)_2 \\[4pt]
\downarrow & & \downarrow \\[4pt]
(0)_1 & & (0)_1\,.
\end{array} \tag{4.40}
$$

The coefficient $C'$ can be cancelled, where the subscripts label the orders of the original representation. This is a relatively special basis change that mixes the lower order operators and the higher order ones while keeping the chain structure. After having gone through all the 2-pt correlators for chain representations, we found that $(0) \to (1) \to (0)$ representation is the unique example that is possible to do this type of basis change. However for net representations, there exist numerous cases that we can apply this type of basis change and thus we leave the detailed discussions for this type of basis changes in section 4.3.

### 4.2.4 An indelible stain: lack of selection rules

One interesting question is if we can do some basis change such that the structure of the correlators could be simplified. For example, we can do Schmidt orthogonalization on the operators

$$\rho_A = \bullet \longrightarrow \bullet \longrightarrow \bullet \longrightarrow \bullet \longrightarrow \bullet \longrightarrow \bullet \longrightarrow \bullet$$

$$\rho_{A'} = \blacksquare \longrightarrow \blacksquare \longrightarrow \blacksquare \longrightarrow \blacksquare$$

$$\rho_B = \triangle \longrightarrow \triangle \longrightarrow \triangle \longrightarrow \triangle$$

$$\rho_A \rightarrow \widetilde{\rho_A} = \begin{pmatrix}\bullet\\+\\\blacksquare\end{pmatrix} \longrightarrow \begin{pmatrix}\bullet\\+\\\blacksquare\end{pmatrix} \longrightarrow \begin{pmatrix}\bullet\\+\\\blacksquare\end{pmatrix} \longrightarrow \begin{pmatrix}\bullet\\+\\\blacksquare\end{pmatrix} \longrightarrow \bullet \longrightarrow \bullet \longrightarrow \bullet$$

Figure 10: The representation $\rho_{A'}$ have the same structure as the leading parts of the representation $\rho_A$, and $\rho_{A'} = \rho_B^{\text{dual}}$. The basis change we consider here is $\rho_A \rightarrow \rho_A + c\rho_{A'}$.

with the same quantum numbers such that the 2-pt correlators are diagonalized. In the CCFT, the sub-sectors of the representations of $d \geq 3$ CCA rotation are irreducible $SO(d-1)$ representations. The basis change should respect the CCA rotation group, and thus the mixing of the operators could only happen between the operators carrying the same $SO(d-1)$ representations. There are only three types of basis changes that keeps the representation structure, namely,

- Type 1: the basis changes between the operators of different orders in the same representation, for example, (4.40);

- Type 2: the basis changes between different representations with the same leading part, defined as (4.41) and shown in Figure 10;

- Type 3: the basis changes between different operators in the same representations, i.e. Schmidt orthogonalization.

In practice, Type 1 basis change will help us to set the 2-pt correlators to be of power-law structure, as mentioned in the last section. For chain representations, (4.40) is the unique example, so we leave the discussions on Type 1 basis changes to section 4.3 after we briefly discuss the 2-pt correlators of net representations. Besides, we can always do the Schmidt orthogonalization first, which has nothing to do with the 2-pt correlators of operators in different representations.

The only possibility to simplify the 2-pt correlators of different representations is to use Type 2 basis changes. Thus, in the following, we try to discuss the implication of Type 2 basis change on the nonvanishing 2-pt correlators of the operators in different representations. As we shown before, the nonvanishing 2-pt correlators of chain representations come from two classes, the entirely inverse pattern which actually require the two operators to be in the dual representations, and the partially inverse pattern. The former one can be simply normalized, and we only need to consider the latter one. It turns out that Type 2 basis changes are not enough to give rise to the selection rules.

Consider the 2-pt correlator of two operators in the representations $\rho_A$ and $\rho_B$, which are in partially inverse pattern. The representation $\rho_{A'}$ have the same structure as the leading parts of representation $\rho_A$, and is entirely inverse to $\rho_B$. Following the earlier discussion, we have non-vanishing 2-pt correlators $\langle \mathcal{O}_1 \mathcal{O}_2 \rangle$ for $\mathcal{O}_1 \in \rho_A$ and $\mathcal{O}_2 \in \rho_B$, which are the same as the 2-pt correlators $\langle \mathcal{O}_1' \mathcal{O}_2 \rangle$ for $\mathcal{O}_1' \in \rho_{A'}$ and $\mathcal{O}_2 \in \rho_B$ up to 2-pt coefficients. It is indeed the case because that in the partially inverse case, the lower-order operators do not give any

contribution. Therefore, we can always make a basis change

$$
\begin{aligned}
\rho_A \Rightarrow \tilde{\rho}_A &= \rho_A + c\rho_{A'}, \\
\text{with } \mathcal{O}_1 &= \mathcal{O}_1^{(q)} \rightarrow \cdots \rightarrow \mathcal{O}_1^{(q-q'+1)} \rightarrow \mathcal{O}_1^{(q-q')} \rightarrow \cdots \rightarrow \mathcal{O}_1^{(1)} \\
\Rightarrow \tilde{\mathcal{O}}_1 &= \mathcal{O}_1 + c\mathcal{O}_1' \\
&\equiv (\mathcal{O}_1^{(q)} + c\mathcal{O}_1'^{(q')}) \rightarrow \cdots \rightarrow (\mathcal{O}_1^{(q-q'+1)} + c\mathcal{O}_1'^{(1)}) \rightarrow \mathcal{O}_1^{(q-q')} \rightarrow \cdots \rightarrow \mathcal{O}_1^{(1)},
\end{aligned}
\tag{4.41}
$$

with a suitable $c$ such that $\langle \tilde{\mathcal{O}}_1 \mathcal{O}_2 \rangle = 0$. It seems that Type 2 basis changes can indeed make some 2-pt correlators for the operators in different representations vanish.

However, Type 2 basis changes can not fix all the 2-pt coefficients. For a specific example, consider the following representations:

$$
\rho_1 = (2), \quad \rho_2 = (1), \quad \rho_3 = (2) \rightarrow (1), \quad \rho_4 = (1) \rightarrow (2).
\tag{4.42}
$$

We list all the 2-pt correlators between these representations in Table 4. Obviously, there are seven independent 2-pt coefficients $C_{\rho_i \rho_j}$ if we require the exchange symmetry $\langle \mathcal{O}_1 \mathcal{O}_2 \rangle = \langle \mathcal{O}_2 \mathcal{O}_1 \rangle$. But, there are only two options on Type 2 basis change

$$
\rho_3 \rightarrow \rho_3 + c_{5,1}\rho_1, \quad \rho_4 \rightarrow \rho_4 + c_{6,2}\rho_2.
\tag{4.43}
$$

They are not enough to fix all the 2-pt coefficients, even after taking into account of the renormalizations on four operators. We can actually make the 2-pt correlators of partially inverse pattern vanishing in this example, the price we pay is that other 2-pt correlators could become complicated. Therefore, we do not do any Type 2 basis change in the following, as it doe not lead to much simplification.

Table 4: The 2-pt correlators between the operators in (4.43), where we have left out the scaling part $|\vec{x}|^{-\Delta_i - \Delta_j}$ for simplicity. Here we require the exchange symmetry $\langle \mathcal{O}_1 \mathcal{O}_2 \rangle = \langle \mathcal{O}_2 \mathcal{O}_1 \rangle$, and thus there are seven independent 2-pt coefficients.

| | | $\rho_1$ (2) | $\rho_2$ (1) | $\rho_3$ (2) $\rightarrow$ (1) | $\rho_4$ (1) $\rightarrow$ (2) |
|---|---|---|---|---|---|
| $\rho_1$ | (2) | $C_{\rho_1\rho_1}[I_{2,2}]$ | $[0]$ | $C_{\rho_1\rho_3}\begin{bmatrix} I_{2,2} & 0 \end{bmatrix}$ | $\begin{bmatrix} 0 & 0 \end{bmatrix}$ |
| $\rho_2$ | (1) | $[0]$ | $C_{\rho_2\rho_2}[I_{1,1}]$ | $\begin{bmatrix} 0 & 0 \end{bmatrix}$ | $C_{\rho_2\rho_4}\begin{bmatrix} I_{1,1} & 0 \end{bmatrix}$ |
| $\rho_3$ (2)$\downarrow$(1) | | $C_{\rho_1\rho_3}\begin{bmatrix} I_{2,2} \\ 0 \end{bmatrix}$ | $\begin{bmatrix} 0 \\ 0 \end{bmatrix}$ | $C_{\rho_3\rho_3}\begin{bmatrix} I_{2,2} & 0 \\ 0 & 0 \end{bmatrix}$ | $C_{\rho_3\rho_4}\begin{bmatrix} \frac{t}{|\vec{x}|}I_{2,1} & I_{2,2} \\ I_{1,1} & 0 \end{bmatrix}$ |
| $\rho_4$ (1)$\downarrow$(2) | | $\begin{bmatrix} 0 \\ 0 \end{bmatrix}$ | $C_{\rho_2\rho_4}\begin{bmatrix} I_{1,1} \\ 0 \end{bmatrix}$ | $C_{\rho_3\rho_4}\begin{bmatrix} \frac{t}{|\vec{x}|}I_{1,2} & I_{1,1} \\ I_{2,2} & 0 \end{bmatrix}$ | $C_{\rho_4\rho_4}\begin{bmatrix} I_{1,1} & 0 \\ 0 & 0 \end{bmatrix}$ |

Things become worse if we consider longer chains. For example, consider the following nine[18] chain representations with rank$\leq 3$ and $j = 1, 2, 3$

$$
\begin{aligned}
\text{rank} = 1: \quad & (3), \quad (2), \quad (1), \\
\text{rank} = 2: \quad & (3) \to (2), \quad (2) \to (1), \quad (1) \to (2), \quad (2) \to (3), \\
\text{rank} = 3: \quad & (3) \to (2) \to (1), \quad (1) \to (2) \to (3),
\end{aligned}
\tag{4.44}
$$

we even can not make all the 2-pt correlators of partially inverse pattern vanishing in this case. Roughly speaking, if we consider chain representations with rank $\leq r$ and $j = 1, \ldots, r$, the following numbers grow with $r$ as

$$
\begin{aligned}
(\#\text{representation}) &= (\#\text{re-normalization d.o.f.}) = r^2, \\
(\#\text{Type 2 basis change d.o.f.}) &= \frac{1}{3} r(r+1)(r-1), \\
(\#\text{non-zero 2-pt coefficients}) &= \frac{1}{6} r(5r^2 + 1),
\end{aligned}
\tag{4.45}
$$

$$
\Rightarrow \quad (\#\text{unfixed 2-pt coefficients}) = \frac{1}{6} r(5r^2 + 1) - \frac{1}{3} r(r+1)(r-1) - r^2 = \frac{1}{2} r(r-1)^2.
$$

The number of unfixed 2-pt coefficients grows as $O(r^3)$, which means we can not get a simple selection rule.

### 4.2.5 Short summary for 2-pt correlators of chain representations

In the above discussions, we have dealt with all possible 2-pt correlators for the operators in chain representations and obtained the constraints on non-trivial 2-pt correlators. The non-trivial 2-pt correlators with $t$ dependence come from the operators with entirely or partially inverse chain representations. The trivial 2-pt correlators include three types: (a) The operators of the same $SO(3)$ representations for rank-1 singlets; (b) Both operators $\mathcal{O}_1, \mathcal{O}_2 \in (j) \to (j)$; (c) The operators with the same $SO(3)$ representations in the top sub-sector. The 2-pt correlators in other cases are all vanishing. The non-zero 2-pt correlators consist of some power of $|\vec{x}_{12}|$ representing the scaling behavior, the tensor structure $I_{j_1, j_2}^{m_1, m_2}$ representing the $SO(3)$ structure, and the power law of $t_{12}/|\vec{x}_{12}|$

$$
\langle \mathcal{O}_1 \mathcal{O}_2 \rangle = \frac{C \, (t_{12}/|\vec{x}_{12}|)^n \, I_{j_1, j_2}^{m_1, m_2}}{|\vec{x}_{12}|^{(\Delta_1 + \Delta_2)}}, \quad \text{with } \Delta_1 = \Delta_2.
\tag{4.46}
$$

The tensor structure $I_{j_1, j_2}^{m_1, m_2}$ including the relative coefficients within are also fixed. The 2-pt coefficients $C$, however, are not generally fixed.[19] It should be stressed that different from the CFT case where we have the selection rule that the 2-pt correlators for the operators in different representations vanish, the 2-pt correlators for the operators in different representations in CCFT can be non-zero. This will cause some difficulties when we try to do basis change. See section 4.3 for more details.

---

[18]Here we only consider the increasing or decreasing chain representations. The other possibilities are $(j) \to (j)$ and $(0) \to (1) \to (0)$. These representations contribute to the counting of d.o.f. in (4.45) as the subleading terms.

[19]This result seems to be in mismatch with the results in [38]. However, it should be in mind that our discussions are purely based on the symmetry, requiring all the operators to be primary operators. In that paper, the authors calculated 2-pt functions of covariant and contravariant vector gauge fields (electric and magnetic sectors, as discussed in section 3.6). In fact, the gauge fields in that paper are not primary operators, and moreover their correlators were studied based on Carrollian electrodynamics within an unfixed gauge, leading to two unfixed coefficients in the electric-electric correlator:

$$
\langle \phi_i(t_1, x_1) \phi_j(t_2, x_2) \rangle = \frac{\gamma_1}{r^2} \delta_{ij} + \frac{\gamma_2}{r^4} x_i x_j, \quad \text{with } \Delta = 1,
$$

where $r = |\vec{x}_{12}|$.

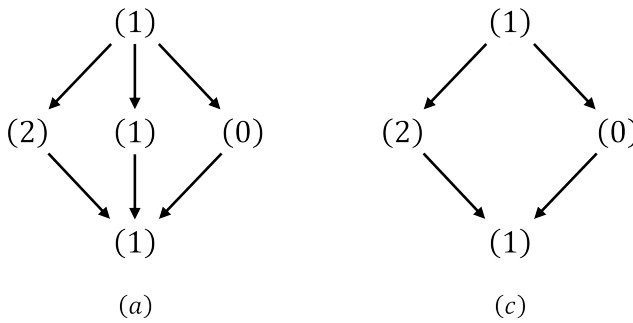

Figure 11: Representation (a) and (c) in Figure 7 repeated here.

## 4.3 Two-point correlators for net representations

The 2-pt correlation functions of net representations are quite involved, although the symmetries are good enough to fix the 2-pt correlators for net representations. The difficulty comes from that the $B$ generators link a $SO(3)$ sub-sector to the lower order sub-sectors in a multiplet. Omitting the coefficients, we have: $[B^i, \mathcal{O}^{(j,q)}] = \mathcal{O}^{(j+1,q-1)} + \mathcal{O}^{(j,q-1)} + \mathcal{O}^{(j-1,q-1)}$, resulting multiple terms in Ward identities

$$\langle (B^i \mathcal{O}^{(j+1,q-1)}) \ldots \rangle = \langle \mathcal{O}^{(j+1,q-1)} \ldots \rangle + \langle \mathcal{O}^{(j,q-1)} \ldots \rangle + \langle \mathcal{O}^{(j-1,q-1)} \ldots \rangle. \tag{4.47}$$

Generally, these three terms are all non-vanishing, and the calculation in chain representation could not directly apply here.

Here we present a non-trivial examples that two operators $\mathcal{O}_1, \mathcal{O}_2$ in (a) representation in Figure 7. We recall the representation here in Figure 11 for convenience. As discussed in section 3.3, the relative coefficients between middle three $SO(3)$ sub-sectors obey the constraint $c_1 = c_2 + 5c_3$ from (3.23). Re-normalizing the lowest sub-sector (1) rescales the three coefficients and thus there left one degree of freedom. This gives a representation family labeled by one parameter. We can set the coefficients for $\mathcal{O}_1$ and $\mathcal{O}_2$ differently, denoting as $c_{1,a}$ and $c_{2,a}$. We choose the rescaling that $c_{i,3} = 1$ to fix the coefficients and the resulting 2-pt correlators satisfying the Ward identities are[20]

Level 5:
$$f_{3,3}^{(1),(1)} = \frac{C\delta\, t_{12}^2/|\vec{x}_{12}|^2 \tilde{I}_{1,1}^{m_1,m_2} + C' I_{1,1}^{m_1,m_2}}{|\vec{x}_{12}|^{2\Delta}},$$

Level 4:
$$f_{2,3}^{(2),(1)} = \frac{C\delta\, t_{12}/|\vec{x}_{12}| I_{2,1}^{m_1,m_2}}{|\vec{x}_{12}|^{2\Delta}}, \qquad f_{2,3}^{(0),(1)} = \frac{C\delta\, t_{12}/|\vec{x}_{12}| I_{0,1}^{m_2}}{|\vec{x}_{12}|^{2\Delta}}, \qquad f_{2,3}^{(1),(1)} = 0,$$
$$f_{3,2}^{(1),(2)} = \frac{C\delta\, t_{12}/|\vec{x}_{12}| I_{1,2}^{m_1,m_2}}{|\vec{x}_{12}|^{2\Delta}}, \qquad f_{3,2}^{(1),(1)} = 0, \qquad f_{3,2}^{(1),(0)} = \frac{C\delta\, t_{12}/|\vec{x}_{12}| I_{1,0}^{m_2}}{|\vec{x}_{12}|^{2\Delta}},$$

Level 3:
$$f_{1,3}^{(1),(1)} = \frac{C\delta\, I_{1,1}^{m_1,m_2}}{|\vec{x}_{12}|^{2\Delta}}, \qquad f_{2,2}^{(2),(2)} = \frac{C\delta\, I_{2,2}^{m_1,m_2}}{|\vec{x}_{12}|^{2\Delta}}, \qquad f_{2,2}^{(2),(1)} = 0, \qquad f_{2,2}^{(2),(0)} = 0,$$
$$f_{2,2}^{(1),(2)} = 0, \qquad f_{2,2}^{(1),(1)} = 0, \qquad f_{2,2}^{(1),(0)} = 0,$$
$$f_{2,2}^{(0),(2)} = 0, \qquad f_{2,2}^{(0),(1)} = 0, \qquad f_{2,2}^{(0),(0)} = \frac{C\delta}{|\vec{x}_{12}|^{2\Delta}}, \qquad f_{3,1}^{(1),(1)} = \frac{C\delta\, I_{1,1}^{m_1,m_2}}{|\vec{x}_{12}|^{2\Delta}},$$

---

[20]This rescaling dose not work when $c_{i,3} = 0$. However this case won't cause any difference. The non-trivial 2-pt correlators with $C \neq 0$ in (4.48) only appear for $c_{i,2} = 0$, while other cases are trivial with $C = 0$, and particularly $c_{i,3} = 0$ also gives $C = 0$. The $C'$ can be chosen arbitrarily no matter how we rescale the lowest (1) sub-sectors.

$$\text{Level 2:} \qquad f_{1,2}^{(1),(2)} = 0\,, \quad f_{1,2}^{(1),(1)} = 0\,, \quad f_{1,2}^{(1),(0)} = 0\,,$$
$$f_{2,1}^{(2),(1)} = 0\,, \quad f_{2,1}^{(1),(1)} = 0\,, \quad f_{2,1}^{(0),(1)} = 0\,,$$

$$\text{Level 1:} \qquad f_{1,1}^{(1),(1)} = 0\,,$$

$$\text{with } \Delta_1 = \Delta_2 = \Delta\,, \ \mathcal{O}_1, \mathcal{O}_2 \in \text{(a) in Figure 11,}$$
$$\delta = 1\,, \text{ for } c_{1,2} = c_{2,2} = 0\,, \quad \delta = 0\,, \text{ otherwise}\,. \tag{4.48}$$

The tensor structure $I_{1,1}^{m_1,m_2}$ and $\tilde{I}_{1,1}^{m_1,m_2}$ are both the tensor structures for $\mathcal{O}_1, \mathcal{O}_2 \in$ (1), but with different coefficients. For $\mathcal{O}_1$ and $\mathcal{O}_2$ both in (c) representation in Figure 11, the 2-pt correlators are polynomials of $t_{12}/|\vec{x}_{12}|$ since the $C'$ term in the former case can be cancelled by basis change, similar to the case in (4.39). For either one of $\mathcal{O}_1$ and $\mathcal{O}_2$ in (a) with non-zero $c_{i,2}$, the correlators are trivial with $C = 0$. The big difference comes from the fact that there is no Ward identities applied on the middle (1) sub-sector for the former case, and thus we have extra solutions proportional to $C$.

In the above discussions, we have used the trick of changing basis. The purpose using the basis changes are different from the one in CFT case, and here we give some brief comments. This basis-changing technique is frequently used in the calculations of the CFT 2-pt correlators. The purpose there is to make the 2-pt correlators diagonalized. To be more specific, the 2-pt correlators from the Ward identities are non-vanishing for the operators in the same representations: $f^{(\text{CFT})} \propto \delta_{\Delta_1,\Delta_2} \delta_{\rho_1,\rho_2}$, where $\mathcal{O}_i \in \rho_i$ and $\rho_i$'s are the representations of $SO(d)$, the rotation part of CFT. Particularly, the 2-pt correlators for different $\mathcal{O}_1$ and $\mathcal{O}_2$ carrying the same scaling dimension $\Delta_1 = \Delta_2$ and in the same $SO(d)$ representation are non-vanishing. Using the basis change, or Schmidt orthogonalization, we can set the correlator diagonalized with $f^{(\text{CFT})} \propto \delta_{\mathcal{O}_1,\mathcal{O}_2}$.

In the case of LogCFT [57], the 2-pt correlators are non-vanishing for different multiplet representations. However, the sub-sectors are all 1-dimensional, and we can always do the basis change between the multiplet representations. Taking $\mathcal{O}_1 = A$ of rank-3 and $\mathcal{O}_2 = B$ of rank-2[21] for example, the constraints from the Ward identities would not require the 2-pt correlators of different multiplets $\langle AB \rangle$ to be vanishing. However, we can do the basis change

$$
\begin{pmatrix} A_3 \\ \downarrow \quad B_2 \\ A_2 \ , \ \downarrow \\ \downarrow \quad B_1 \\ A_1 \end{pmatrix}
\Rightarrow
\begin{pmatrix} A_3 + a_1 A_2 + a_2 A_1 + c_1 B_1 + c_2 B_2 \\ \downarrow \qquad\qquad\qquad B_2 + b_1 B_2 \\ A_2 + a_1 A_1 + c_1 B_2 \quad , \quad \downarrow \\ \downarrow \qquad\qquad\qquad B_1 \\ A_1 \end{pmatrix}, \tag{4.49}
$$

such that the 2-pt correlators of the same multiplets $\langle AA \rangle$ and $\langle BB \rangle$ are proportional to the powers of $(\ln x)^m$ with carefully chose parameters $\{a_i, b_i\}$, and the 2-pt correlators of different multiplets $\langle AB \rangle$ vanishes with carefully chose $\{c_i\}$. The proof can be reached in Appendix A in [57], and the argument can be generalized to 2d GCFT.

These basis changes in LogCFT that keeps the structure of the representations intact can be classified in two types, namely the basis change carried by $\{a_i, b_i\}$ corresponding to the basis change between the operators at different order in the same representation, i.e. Type

---

[21]We borrow our notation introduced in Figure 2. The real log-multiplet representation is slightly different in the sense that the multiplet is defined with respect to the dilatation operator. Thus the arrow linking $A_r$ to $A_{r-1}$ means $[D, A_r] = \Delta A_r + A_{r-1}$. The case for 2d GCFT is similar, but with the multiplet representation is defined with respect to the boost operator.

1, and the basis change carried by the parameters $\{c_i\}$ corresponding to the basis change between different representations, i.e. Type 2. Both of them appear for chain representations as mentioned in previous section. They are sure to appear for net representations. The Type 2 basis changes for chain representations have been discussed in section 4.2.4. The discussions naturally apply to general net representations, leading to the conclusion that there is no selection rule for the 2-pt correlators. Here we focus on the Type 1 basis changes which is helpful to express the 2-pt correlators in terms of powers of $t_{12}/|\vec{x}_{12}|$. Such basis exchanges exist for the representations with self-symmetric structure. By self-symmetric structure, we mean some sub-representations match the form of the top part, shown in Figure 12. The example (a) is clearly self-symmetric that the red (1) sub-sector is a sub-representation and matches the form of top blue (1) sub-sector. For the example (b), the sub-representation shown in red matches the top blue part thus it is a self-symmetric representation. More generally, a representation may have multiple top parts, and it is referred to as self-symmetric if the sub-representation match some top part, as shown in (c). And there may exist multiple self-symmetric structures in one representation, as shown in (d).

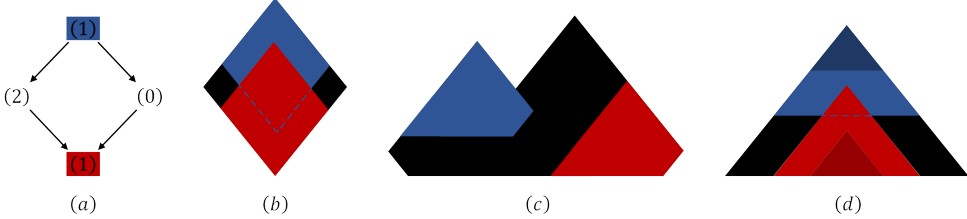

Figure 12: Some examples of self-symmetric representations.

It appears that the 2-pt correlators of the operators in the representations having self-symmetric structure would generally not obey the power laws by the constraints from the symmetry, but there always exist a basis change such that the correlators of the operators in the changed representations obey the power laws.

### 4.4 Three-point correlators of chain representations

In this subsection, we would like to give some observations on the 3-point correlators for the operators all in chain representations. The algorithm of calculating the 3-pt corrlators is similar to the 2-pt case. At the lowest level, which have the same Ward identities as the ones in $CFT_3$, there is no selection rule for the spin representations, which means there are non-zero correlators among the operators of different spins. For instance, $\langle \mathcal{O}_1 \mathcal{O}_2 \mathcal{O}_3 \rangle \neq 0$ with $\mathcal{O}_1, \mathcal{O}_2 \in (0)$, and $\mathcal{O}_3 \in (1)$. The general 3-pt correlators are non-zero, and the restrictions are much looser in CCFT.

Some ground rules can still be given by considering the Ward identities. One rule similar to the 2-pt case is that there must be one non-decreasing chain for the 3-pt correlators to be nontrivial. The argument is the same as the 2-pt case: since there are three degrees of freedom of time coordinates, if there are three independent differential equations on time coordinates, then the 3-pt correlator is independent of all the time coordinates and thus is trivial. It is obvious that for all the three operators being in some decreasing chain representations, the 3-pt correlators are trivial. The simplest example is that

$$
\begin{array}{ll}
\text{Level 2: } f_{1,1,2}^{0,0,m_3}, & \\
\text{Level 1: } f_{1,1,1}^{0,0,0}, & \mathcal{O}_1, \mathcal{O}_2 \in (0), \quad \mathcal{O}_3 \in \begin{matrix} (1) \\ \downarrow \\ (0) \end{matrix} .
\end{array}
\tag{4.50}
$$

In this case, the generators $B^+, B^3$ annihilate $\mathcal{O}_i^{(m_i=1,q_i=2)}$, and the Ward identities of $B^+, B^3, P^0$ for $f_{1,1,2}^{0,0,1}$ give the following differential equations

$$
\begin{aligned}
P^0: & \quad (\partial_{t_1} + \partial_{t_2} + \partial_{t_3}) f_{1,1,2}^{0,0,1} = 0, \\
B^3: & \quad (x_1^3 \partial_{t_1} + x_2^3 \partial_{t_2} + x_3^3 \partial_{t_3}) f_{1,1,2}^{0,0,1} = 0, \\
B^+: & \quad (x_1^+ \partial_{t_1} + x_2^+ \partial_{t_2} + x_3^+ \partial_{t_3}) f_{1,1,2}^{0,0,1} = 0, \quad x_i^+ = i x_i^1 + x_i^2.
\end{aligned}
\tag{4.51}
$$

With suitable combinations, the above equations suggest that $f_{1,1,2}^{0,0,1}$ and then $f_{1,1,2}^{0,0,m_3}$ are independent of time coordinates. And furthermore the Ward identities of $B^-$ on $f_{1,1,2}^{0,0,1}$ requires $f_{1,1,1}^{0,0,0} = 0$ such that the 3-pt correlator is trivial with only the highest level one being non-zero.

Here we present the simplest example of non-trivial 3-pt correlators:

$$
\text{Level 2:} \quad f_{1,1,2}^{0,0,0} = \frac{C_3\, g + C_3'}{|\vec{x}_{12}|^{\Delta_1+\Delta_2-\Delta_3} |\vec{x}_{13}|^{\Delta_1-\Delta_2+\Delta_3} |\vec{x}_{23}|^{-\Delta_1+\Delta_2+\Delta_3}}, \tag{0}
$$

$$
\mathcal{O}_1, \mathcal{O}_2 \in (0), \quad \mathcal{O}_3 \in \downarrow,
$$

$$
\text{Level 1:} \quad f_{1,1,1}^{0,0,m_3} = \frac{C_3\, I_{0,0,1}^{m_3}}{|\vec{x}_{12}|^{\Delta_1+\Delta_2-\Delta_3} |\vec{x}_{13}|^{\Delta_1-\Delta_2+\Delta_3} |\vec{x}_{23}|^{-\Delta_1+\Delta_2+\Delta_3}}, \tag{1}
$$

$$
\tag{4.52}
$$

where

$$
\begin{aligned}
g = & \frac{t_{12}(x_{23}x_{13} + y_{23}y_{13} + z_{23}z_{13})}{|\vec{x}_{12}||\vec{x}_{13}||\vec{x}_{23}|} + \frac{t_{23}(x_{12}x_{13} + y_{12}y_{13} + z_{12}z_{13})}{|\vec{x}_{12}||\vec{x}_{13}||\vec{x}_{23}|} \\
& - \frac{t_{13}(x_{12}x_{23} + y_{12}y_{23} + z_{12}z_{23})}{|\vec{x}_{12}||\vec{x}_{13}||\vec{x}_{23}|},
\end{aligned}
\tag{4.53}
$$

and

$$
I_{0,0,1}^{m_3} =
\begin{cases}
\dfrac{(x_{13} - i y_{13})\vec{x}_{23}^2 - (x_{23} - i y_{23})\vec{x}_{13}^2}{|\vec{x}_{12}||\vec{x}_{13}||\vec{x}_{23}|}, \\[3mm]
\dfrac{-i\sqrt{2}(z_{13}\vec{x}_{23}^2 - z_{23}\vec{x}_{13}^2)}{|\vec{x}_{12}||\vec{x}_{13}||\vec{x}_{23}|}, \\[3mm]
\dfrac{(x_{13} + i y_{13})\vec{x}_{23}^2 - (x_{23} + i y_{23})\vec{x}_{13}^2}{|\vec{x}_{12}||\vec{x}_{13}||\vec{x}_{23}|}.
\end{cases}
\tag{4.54}
$$

There are two 3-pt coefficients: $C_3$ and $C_3'$. The level-2 correlator $f_{1,1,2}^{0,0,0}$ is a polynomial in $t$ coordinates as expected.

## 5 Correlation functions in GCFT

Since the finite representations of GCA are exactly the same with the ones of CCA as discussed in section 3.5, we can use the same strategy as introduced in section 4.2 to calculate the correlation functions in GCFT. The results are even much simpler for GCFT.

The Ward identities in GCFT are different from the ones in CCFT, since the differential operators and commutation relations of GCA are different. Here we present some useful Ward identities. The BCH formula gives

$$
\begin{aligned}
U^{-1}(x) B^i U(x) &= B^i - t P^0, \\
U^{-1}(x) K^0 U(x) &= K^0 + 2(tD - x^i B^i) + t^2 P^0 + 2t x^i P^i, \\
U^{-1}(x) K^i U(x) &= K^i + 2t B^i - t^2 P^i,
\end{aligned}
\tag{5.1}
$$

which lead to the Ward identities:

$$
\begin{aligned}
B^i : \ 0 =& (-t_1\partial_1^i - t_2\partial_2^i + \dots)\langle \mathcal{O}_1\mathcal{O}_2\dots\rangle + \langle (B^i\mathcal{O}_1)\mathcal{O}_2\dots\rangle + \langle \mathcal{O}_1(B^i\mathcal{O}_2)\dots\rangle + \dots, \\
K^0 : \ 0 =& ((t_1^2\partial_{t_1} + 2t_1 x_1^i\partial_1^i) + (t_2^2\partial_{t_2} + 2t_2 x_2^i\partial_2^i) + \dots)\langle \mathcal{O}_1\mathcal{O}_2\dots\rangle \\
& + 2\big(t_1\Delta_1\langle \mathcal{O}_1\mathcal{O}_2\dots\rangle - x_1^i\langle (B^i\mathcal{O}_1)\mathcal{O}_2\dots\rangle\big) \\
& + 2\big(t_2\Delta_2\langle \mathcal{O}_1\mathcal{O}_2\dots\rangle - x_2^i\langle \mathcal{O}_1(B^i\mathcal{O}_2)\dots\rangle\big) + \dots, \\
K^i : \ 0 =& (-t_1^2\partial_1^i - t_2^2\partial_2^i + \dots)\langle \mathcal{O}_1\mathcal{O}_2\dots\rangle + 2t_1\langle (B^i\mathcal{O}_1)\mathcal{O}_2\dots\rangle + 2t_2\langle \mathcal{O}_1(B^i\mathcal{O}_2)\dots\rangle + \dots
\end{aligned}
\tag{5.2}
$$

Let us start from the 2-pt correlators of the singlets. The Ward identities of $\{P^i, B^i\}$ give $2(d-1)$ differential equations. As $B^i$ act trivially on the singlets, we find

$$
\begin{aligned}
P^i : \quad & (\partial_{x_1^i} + \partial_{x_2^i})f = 0, \\
B^i : \quad & (-t_1\partial_{x_1^i} - t_2\partial_{x_2^i})f = 0.
\end{aligned}
\tag{5.3}
$$

These equations are enough to fix the form of the correlators. More explicitly, similar to the CCFT case, there are actually two independent solutions of the Ward identities in GCFT. For the scalar type operator, the 2-pt correlator is of the form

$$
\langle \mathcal{O}_1\mathcal{O}_2\rangle = c_1\frac{1}{|t_{12}|^{2\Delta}} + c_2\frac{\delta(t_{12})}{|\vec{x}_{12}|^{2\Delta-1}}.
\tag{5.4}
$$

For the operators $\mathcal{O}_i \in (j)$, there will be additionally non-trivial tensor structures $I$ in the second type solution proportional to $\delta(t_{12})$. However, as it is $\delta$-functional distribution, the second type solution is less interesting. For simplicity, we discard this kind of solution in the following discussion. Consequently, the 2-pt correlators of the singlets can not have any tensor structure, i.e.

$$
\langle \mathcal{O}_1\mathcal{O}_2\rangle = \frac{1}{|t_{12}|^{2\Delta}}\delta_{\Delta_1,\Delta_2}, \quad \mathcal{O}_1,\mathcal{O}_2 \in (0),
\tag{5.5}
$$

and all other 2-pt correlators for the singlet operators vanish.

Next, we consider the 2-pt correlators of the operators in the multiplet representations. It turn out that there is no non-trivial 2-pt correlators for 4$d$ GCFT. To get non-trivial 2-pt correlators, the higher level correlators must be related to non-zero lower level correlators by the Ward identities of $B$ generators, and thus we should focus on the representations (no matter a chain or a net representations) with an $SO(3)$ spin-0 representation $(0)$ at the lowest level.[22] Besides, from the constraints of the representations introduced in section 3.3, we know that the representations with a sub-sector $(0)$ at the lowest level must be of the form as in Figure 13 that only a $(1)$ sub-sector links to the ending $(0)$.

Consider the following 2-pt correlators of chain representations

$$
\begin{array}{ccc}
\vdots \quad \vdots & & \mathcal{O}_1 \in \dots \rightarrow (0), \\
\text{Level 2:} \quad f_{1,2}^{0,m_2}, & & \mathcal{O}_2 \in \cdots \rightarrow (1) \rightarrow (0), \\
\text{Level 1:} \quad f_{1,1}^{0,0}, & & \text{with } \Delta_1 = \Delta_2 = \Delta.
\end{array}
\tag{5.6}
$$

with the $B$ generators acting on $\mathcal{O}_2^{(m_2,q=2)}$ as its representation matrices $\mathbf{B}^i$: $[B, \mathcal{O}_2^{(m_2,q=2)}] = (\mathbf{B}^i)_{m_2,0}\,\mathcal{O}_2^{(0,q=1)}$. The Ward identities of $B^i$ and $K^i$ generators on $f_{1,2}^{0,m_2}$ require

---

[22]In general net representations, there may exist multiple ending $(0)$ sub-sectors. A sub-sector $(j)$ is referred to as an ending, if there is no sub-sector $(j')$ such that $B$ generators link $(j)$ to $(j')$, i.e. $[B^i, (j)] = 0$. In this case, the analysis also applies to every ending $(0)$ sub-sectors.

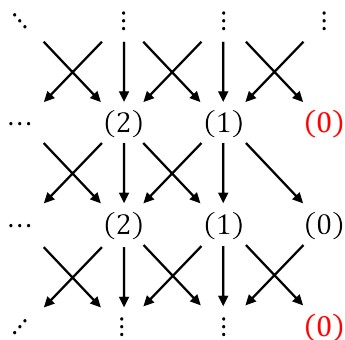

Figure 13: An example with two ending (0) sub-sectors (red ones), and there is only one (1) sub-sector links to each one. Noticing the middle (0) sub-sector is not an ending.

$$-t_{12}\partial_{x_{12}^i} f_{1,2}^{0,m_2} + 0 + (\mathbf{B}^i)_{m_2,0}\, f_{1,1}^{0,0} = 0, \tag{5.7}$$

$$(-t_1^2\partial_1^i - t_2^2\partial_2^i)f_{1,2}^{0,m_2} + 0 + 2t_2(\mathbf{B}^i)_{m_2,0}\, f_{1,1}^{0,0} = 0. \tag{5.8}$$

Multiplying (5.7) by $(t_1 + t_2)$ and minus (5.8), we find

$$t_{12}(\mathbf{B}^i)_{m_2,0}\, f_{1,1}^{0,0} = 0, \quad \forall i = 1,2,3,\ m_2 = \pm 1, 0. \tag{5.9}$$

This means that we have to set $f_{1,1}^{0,0} = 0$ for $f_{1,2}^{0,m_2}$ satisfying its Ward identities. With the lowest level correlator vanishing, the level-2 correlators including $f_{1,2}^{0,m_2}$ behave like the correlators for the singlet operators, and thus the non-vanishing level-2 correlators have the same form as (5.5). Using this argument recursively, we conclude that there is no non-trivial GCFT 2-pt correlators and the non-zero 2-pt correlators must be of the form

$$
\begin{aligned}
\text{Highest Level:} \quad & f_{\text{2-pt}} = \frac{C}{(t_{12})^{2\Delta}}, & & \mathcal{O}_1 \in (0) \to \dots, \\
& & & \mathcal{O}_2 \in (0) \to \dots, \\
\text{Lower Levels:} \quad & 0, & & \text{with } \Delta_1 = \Delta_2 = \Delta,
\end{aligned}
\tag{5.10}
$$

where the representations of $\mathcal{O}_1$ and $\mathcal{O}_2$ can be different general net representations that starting from a (0) sub-sector. In general net representations, there can be multiple starting (0) sub-sectors. In this case, there are multiple non-zero 2-pt correlators $C(t_{12})^{-2\Delta}$ between those starting (0) sub-sectors with possibly different coefficients. Similar to the cases in CCFT, the coefficients are generally not able to be normalized by basis change of the representations.

The 3-pt correlators of singlet operators are also independent of spacial coordinates by the Ward identities of $\{P^i, B^i\}$ generators. Actually the non-zero 3-pt correlators for the singlets are of the form

$$\langle \mathcal{O}_1\mathcal{O}_2\mathcal{O}_3 \rangle = \frac{C_3}{|t_{12}|^{\Delta_1+\Delta_2-\Delta_3}|t_{13}|^{\Delta_1-\Delta_2+\Delta_3}|t_{23}|^{-\Delta_1+\Delta_2+\Delta_3}}, \quad \mathcal{O}_1,\mathcal{O}_2,\mathcal{O}_3 \in (0), \tag{5.11}$$

and others vanish. However, for general multiplet representations, the 3-pt correlators could depend on the spacial coordinates. A non-trivial example is:

$$
\begin{aligned}
\text{Level 2:} \quad & f_{1,1,2}^{0,0,m_3} = \frac{C_3\, I_{0,0,1}^{m_3}}{|t_{12}|^{\Delta_1+\Delta_2-\Delta_3}|t_{13}|^{\Delta_1-\Delta_2+\Delta_3}|t_{23}|^{-\Delta_1+\Delta_2+\Delta_3}}, & & \hspace{1em} (1) \\
& & & \mathcal{O}_1,\mathcal{O}_2 \in (0),\ \mathcal{O}_3 \in\ \downarrow, \\
\text{Level 1:} \quad & f_{1,1,1}^{0,0,0} = \frac{C_3}{|t_{12}|^{\Delta_1+\Delta_2-\Delta_3}|t_{13}|^{\Delta_1-\Delta_2+\Delta_3}|t_{23}|^{-\Delta_1+\Delta_2+\Delta_3}}, & & \hspace{1em} (0)
\end{aligned}
$$

$$\tag{5.12}$$

where the tensor structure is

$$
I_{0,0,1}^{m_3} = \begin{cases} \dfrac{1}{\sqrt{2}}\left(\dfrac{ix_{12}+y_{12}}{t_{12}}+\dfrac{ix_{13}+y_{13}}{t_{13}}+\dfrac{ix_{23}+y_{23}}{t_{23}}\right), \\[3mm] \dfrac{(t_{13})^2 z_{12}+t_{12}(t_{12}-2t_{13})z_{13}}{t_{12}t_{13}t_{23}}, \\[3mm] \dfrac{1}{\sqrt{2}}\left(\dfrac{ix_{12}-y_{12}}{t_{12}}+\dfrac{ix_{13}-y_{13}}{t_{13}}+\dfrac{ix_{23}-y_{23}}{t_{23}}\right). \end{cases} \tag{5.13}
$$

# 6 Conclusion and Discussion

In this paper, we studied CCFT and GCFT in higher dimensions. We paid more attention to $CCFT_4$. As the symmetry algebras in these theories are quite different from the ones in usual CFT, we had to study the highest weight representations of $\mathfrak{cca}_d$ and $\mathfrak{gca}_d$ carefully. It turns out that the representations can be of complicated multiplet structures: the relative simple ones are chain representations, and the more general ones are net representations. Another remarkable feature of the representations is that they are reducible but indecomposable, as the CCA/GCA rotation is not semisimple. We managed to classify all the allowed chain representations, which include the singlets, the increasing chains, the decreasing chains and two exceptional cases of the following forms

$$
(j)\to(j),\ \ j\neq 0,\quad \text{and}\quad (0)\to(1)\to(0). \tag{6.1}
$$

Furthermore we discussed the 2-pt and 3-pt correlators in CCFT and GCFT. In principle, these correlators can be determined by the Ward identities of the symmetry transformations, but due to the complicated structure of the representations, the correlators have some remarkable features. For simplicity we focused on the correlators of the operators in chain representations. Even for this simple case, the 2-pt correlators in CCFT present some novel features.

- Not all the correlators have time-dependence. The non-trivial correlators with time dependence appear only for the representations of certain structure. This is because the temporal and spatial coordinates have different behaviors under symmetry transformations;

- Due to the multiplet structure of the representations, the correlators present multi-level structures. At each level, there are more than one 2-pt coefficients. Even if considering the basis change and renormalization of the operators, not all 2-pt coefficients can be fixed by the Ward identities;

- As the representations are reducible, there is short of selection rule on the representations. This means that the 2-pt correlators of the operators in different representations could be nonvanishing.

More precisely, the non-vanishing trivial 2-pt correlators of chain representations have only three types, and the non-trivial 2-pt correlators comes from the two operators in entirely or partially inverse chain representations. In spite of the multi-level structure, the structure of the 2-pt correlators at each level in CCFT is quite similar with the one in CFT. It consists of a scaling factor $|\vec{x}_{12}|^{-(\Delta_1+\Delta_2)}$ representing the scaling behavior, a tensor structure $I(\vec{x})$ representing the

behavior under spacial rotations $\{J^{ij}\}$ and a factor being of powers of $t_{12}/|\vec{x}_{12}|$ representing the behavior under the Carrollian boosts $\{B^i\}$

$$f_{2\text{-pt}}^{(\text{CCFT})} \propto \frac{(t_{12}/|\vec{x}_{12}|)^n \, I}{|\vec{x}_{12}|^{(\Delta_1+\Delta_2)}}, \quad \text{with } \Delta_1 = \Delta_2. \tag{6.2}$$

We explored the 2-pt correlators of net representations and the 3-pt correlators of chain representations. It turns out that the constraints from the Ward identities are quite loose, and we had to compute them case by case. We found that not all the 2-pt correlators of net representations obey the power laws of $t_{12}/|\vec{x}_{12}|$. Nevertheless, for the operator in a representation with self-symmetric structure, its two-point function with itself obey the power-law, after suitable basis change. We have to admit that our investigation was not thorough, and further studies are needed.

In the present study on CCFT, we focused on the correlators with nontrivial dependence on spatial coordinates. As we discussed in section 4.1, there actually exists a $\delta$-distribution in spacial directions in the 2-pt correlator. Such correlators do appear in a class of Carrollian free theories. Moreover, it was pointed out in [60, 61] that this kind of correlation functions is related to celestial holography. It seems to us that the two kinds of correlators correspond to different quantization scheme. The $\delta$-distribution correlators certainly deserves further studies. It is also important to construct specific theories whose correlators have nontrivial dependence on spatial coordinates.

In contrast, for the GCFT, the structure for the 2-pt correlators are relatively simple due to the action of Galilean boosts. For GCFT, all the 2-pt and 3-pt correlators of chain representations are independent of spatial coordinates. It turns out that the only non-vanishing 2-pt correlator for GCFT appear for the representations with leading (0) sub-sector(s), and it is independent of spacial coordinates

$$f_{2\text{-pt}}^{(\text{GCFT})} \propto \frac{1}{|t|^{(\Delta_1+\Delta_2)}}. \tag{6.3}$$

The structure of the 3-pt correlators of GCFT generally have non-trivial dependence on spacial coordinates, since there are less constraints for the 3-pt correlators than the ones for the 2-pt correlators.

One important question is whether the conformal bootstrap is valid for higher dimensional CCFTs and GCFTs. The first step is to determine the operator product expansion (OPE)

$$\mathcal{O}_1(x_1)\mathcal{O}_2(x_2) = \sum_i f_{12i} P(x_{12}, \partial) \mathcal{O}_i(x), \tag{6.4}$$

in which the differential operator $P(x_{12}, \partial)$ resums the contributions of descendant operators in a highest weight representation and should be fixed by the CCA and GCA respectively.[23] Assuming the convergence of OPE and inserting it into the 3-pt function, we get the relation between the OPE coefficients $f_{12i}$, the 2-pt coefficients $C_{ij}^{(2)}$ and the 3-pt coefficients $C_{ijk}^{(3)}$,

$$\langle \mathcal{O}_1 \mathcal{O}_2 \mathcal{O}_3 \rangle = \sum_i f_{12i} P(x_{12}, \partial_2) \langle \mathcal{O}_i(x_2)\mathcal{O}_3(x_3) \rangle. \tag{6.5}$$

Inserting the OPE into the 4-pt function we get the expansion depending on the squared OPE coefficients and the 2-pt coefficients

$$\langle \mathcal{O}_1 \mathcal{O}_2 \mathcal{O}_3 \mathcal{O}_4 \rangle = \sum_{i,j} f_{12i} f_{34j} P(x_{12}, \partial_2) P(x_{34}, \partial_3) \langle \mathcal{O}_i(x_2)\mathcal{O}_j(x_3) \rangle. \tag{6.6}$$

---

[23]In CFT, this differential operator is called the OPE block, see e.g. [62, 63].

For compact CFT whose spectrum is discrete and positive definite, the 2-pt coefficients can be diagonalized to $C_{ij}^{(2)} = \delta_{ij}$, and the relation (6.5) gives the identification $f_{ijk} = C_{ijk}^{(3)}$. The 4-pt function (6.6) can be simply expanded in terms of the conformal blocks, each of which encodes the contribution of an exchanged conformal family. The diagonalizability of 2-pt coefficients is crucial for establishing the bootstrap equations.

For CCFT, the 2-pt correlators in (6.5) can not be diagonalized, since different representations may have non-vanishing 2-pt correlators and there are 2-pt coefficients that cannot be determined by the symmetry. The relation between the OPE coefficients and the 3-pt coefficients is not simple. For the 4-pt functions we have the expansion

$$\langle \mathcal{O}_1 \mathcal{O}_2 \mathcal{O}_3 \mathcal{O}_4 \rangle = \sum_{i,j} f_{12i} f_{34j} P(x_{12}, \partial_2) P(x_{34}, \partial_3) \langle \mathcal{O}_i(x_2) \mathcal{O}_j(x_3) \rangle \,, \qquad (6.7)$$

in which the "conformal block" $P_1 P_3 \langle \mathcal{O}_i(x_2) \mathcal{O}_j(x_3) \rangle$ depends on two conformal families instead of one. This makes the equation much more involved and hard to analyze.

For GCFT, the only non-vanishing 2-pt correlators are from the operators in the representations having (0) starting sub-sector(s), such that the matrix of the 2-pt coefficients $C_{ij}^{(2)}$ for GCFT is degenerate. Meanwhile, the 3-pt correlators are generally non-vanishing, which means even if we assume

$$C_{123}^{(3)} = \sum_i f_{12i} C_{i3}^{(2)} \,, \qquad (6.8)$$

not all OPE coefficients appear in the relation. Thus it is not enough by only considering the OPE limit 3-pt$\xrightarrow{\text{OPE}}$2-pt to read the OPE coefficients.

Another interesting question is on quantization and the operator-state correspondence in higher dimensional CCFT and GCFT. As the space-time structure is a bit bizarre, the discussion of CFT does not fit here. Two naive guesses for the quantization are: (1) temporal quantization foliating the space-time with equal time slices; (2) spacial radial quantization foliating the space-time with equal spacial radius. Neither of them gives a well-defined operator-state correspondence.

It would be interesting to consider the super-symmetric version of Carrollian/Galilean conformal field theory. It is natural to expect that the super-Carrollian/Galilean conformal algebras (SSCA/SGCA) may show up by taking the ultua- or non-relativistic limit on the superconformal algebra. It is well known that there is no super conformal symmetry in $d > 6$ dimensions [64,65]. From this point of view, it seems that one cannot find $d > 6$ SSCA/SGCA. However, this restriction may break down for super-Carrollian/Galilean conformal symmetry.

## Acknowledgments

We are grateful to Hong-jie Chen, Peng-xiang Hao, Aditya Mehra, Akhila Mohan, Aditya Sharma, Zhe-fei Yu, Xi-nan Zhou for valuable discussions. The work is supported in part by NSFC Grant No. 11735001.

## A  Representation of $SO(d)$ and its limits

In this appendix, we try to interpret the multiplet representations of CCA/GCA rotation from the point of view of taking the limit, i.e. taking the Inonu-Wigner contraction, from irreducible representations of $SO(d)$ [66,67]. The main steps of taking such a limit goes as follows: firstly, separate the generators of $\mathfrak{so}_d$ into the spacial rotation $\{J^{ij}\}$ and the Euclidean boost $\{J^{i0}\}$;

then, decompose the representations of $SO(d)$ as a direct sum of irreducible representations of $SO(d-1)$ generated by $J^{ij}$, which are connected by the action of $J^{i0}$; finally, we get the multiplet representation by taking the Inonu-Wigner contraction, after carefully choosing the scaling behavior of $J^{i0}$'s action under the limit.

The first step is automatic, and the commutation relations of the generators separate into two groups, the ones among $J^{ij}$ and the ones involving $J^{i0}$

$$\begin{aligned}
[J^{ij}, J^{kl}] &= \delta^{ik} J^{jl} - \delta^{il} J^{jk} + \delta^{jl} J^{ik} - \delta^{jk} J^{il}, \\
[J^{ij}, J^{k0}] &= \delta^{ik} J^{j0} - \delta^{jk} J^{i0}, \\
[J^{i0}, J^{j0}] &= J^{ij},
\end{aligned} \tag{A.1}$$

where $J^{i0}$ acts as a vector under the action of $J^{ij}$.

The second step is the branching problem, i.e. decompose the irreducible representations of group $G$ into direct sum of the irreducible representations of its subgroup $H \subset G$. Here we decompose the irreducible representations of $SO(d)$ into the irreducible representations of $SO(d-1) \subset SO(d)$, and this problem is solved by using the Gelfand–Tsetlin patterns [68]. Recall that the irreducible representations of $SO(d)$ group are labeled by

$$[\lambda_1, \ldots, \lambda_n], \quad \lambda_i \in \mathbb{Z}/2, \quad \text{with} \begin{cases} \lambda_1 \geq \cdots \geq |\lambda_n| \geq 0, & d = 2n, \quad \text{even}, \\ \lambda_1 \geq \cdots \geq \lambda_n \geq 0, & d = 2n+1, \quad \text{odd}. \end{cases} \tag{A.2}$$

Here we only concern the decomposition of the irreducible representation of $SO(d)$ into the irreducible representations of $SO(d-1)$, and we have

$$\begin{aligned}
d = 2n: \quad & [\lambda_1, \ldots, \lambda_n] = \bigoplus_{\{\mu\}} M_{\{\mu\}} [\mu_1, \ldots, \mu_{n-1}], \\
& \text{with} \begin{cases} M_{\{\mu\}} = 1, & \text{for } \lambda_1 \geq \mu_1 \geq \lambda_2 \geq \mu_2 \geq \cdots \geq \mu_{n-1} \geq |\lambda_n|, \\ M_{\{\mu\}} = 0, & \text{otherwise}, \end{cases}
\end{aligned} \tag{A.3}$$

$$\begin{aligned}
d = 2n+1: \quad & [\lambda_1, \ldots, \lambda_n] = \bigoplus_{\{\mu\}} M_{\{\mu\}} [\mu_1, \ldots, \mu_n], \\
& \text{with} \begin{cases} M_{\{\mu\}} = 1, & \text{for } \lambda_1 \geq \mu_1 \geq \lambda_2 \geq \mu_2 \geq \cdots \geq \lambda_n \geq |\mu_n|, \\ M_{\{\mu\}} = 0, & \text{otherwise}, \end{cases}
\end{aligned} \tag{A.4}$$

where $M_{\{\mu\}}$ is the multiplicity representing the number of times the representation $[\{\mu\}]$ appearing in the decomposition.

To do the final step taking Inonu-Wigner contraction, we consider $d = 4$ for simplicity. The decomposition goes as

$$[\lambda_1, \lambda_2] = \bigoplus_\mu [\mu], \quad \text{with } \lambda_1 \geq \mu \geq |\lambda_2| \geq 0. \tag{A.5}$$

To match the notation in the main text, we identify the $SO(3)$ representation $[\mu]$ with the notation $(j)$. As for $SO(4)$, noticing that $SO(4) \cong SO(3) \times SO(3)/\mathbb{Z}_2$, the irreducible representation $[\lambda_1, \lambda_2]$ can also be labeled by two $SO(3)$ spin $j_i$. We have the identifications

$$\begin{aligned}
[\mu] &= (j), \quad \text{with } \mu = j, \\
[\lambda_1, \lambda_2] &= (j_1, j_2), \quad \text{with } \lambda_1 = j_1 + j_2, \quad \lambda_2 = j_1 - j_2.
\end{aligned} \tag{A.6}$$

In this notation, (A.5) is equivalent to

$$(j_1, j_2) = \bigoplus_{j=|j_1-j_2|}^{j_1+j_2} (j). \tag{A.7}$$

Since $J^{i0}$ is a vector representation, their action on $(j)$ gives combination of $(1) \otimes (j) = (j+1) \oplus (j) \oplus (j-1)$. The explicit action of $J^{i0}$ on $(j)$ is denoted as

$$J^{i0}|j,m\rangle = a_{j+1}\mathbf{J}^{i0}_{j \to j+1}|j+1,m'\rangle + a_j\mathbf{J}^{i0}_{j \to j}|j,m'\rangle + a_{j-1}\mathbf{J}^{i0}_{j \to j-1}|j-1,m'\rangle\,, \tag{A.8}$$

in which $a_j$'s are essentially the reduced matrix elements of $J^{i0}$ determined by $[J^{i0},J^{j0}] = J^{ij}$, and here we do not need their explicit expressions. Taking the Inonu-Wigner contraction of the Carrollian limit $J^{i0} = \lim_{c \to 0} \frac{1}{c}B^i$ trivializes the third commutations in (A.1) and yields the CCA rotation group with generators $\{J^{ij},B^i\}$. The actions of $B^i$'s on the $(j)$ representations (A.8) are non-vanishing only if we renormalize the states $|j,m\rangle \to f_j(c)|j,m\rangle$ properly

$$\begin{aligned}
\frac{1}{c}B^i f_j(c)|j,m\rangle = {} & a_{j+1}f_{j+1}(c)\mathbf{J}^{i0}_{j \to j+1}|j+1,m'\rangle \\
& + a_j f_j(c)\mathbf{J}^{i0}_{j \to j}|j,m'\rangle + a_{j-1}f_{j-1}(c)\mathbf{J}^{i0}_{j \to j-1}|j-1,m'\rangle\,.
\end{aligned} \tag{A.9}$$

There are three apparent options on $f_j(c)$ without modifying the action of $J$'s: $f_j(c) = c^{\pm j}$ and $f_j(c) = 1$.[24] The different choice of $f_j(c)$, and correspondingly the contracted matrix elements of $B$'s and the resulting representations, are respectively:

$$f_j(c) = c^j \quad \Longrightarrow \quad B^i|j,m\rangle = a_{j-1}\mathbf{B}^i_{j \to j-1}|j-1,m'\rangle\,, \tag{A.10}$$

leading to the increasing chain representations $(j_1 + j_2) \to \cdots \to (|j_1 - j_2|)$;

$$f_j(c) = 1 \quad \Longrightarrow \quad B^i|j,m\rangle = 0\,, \tag{A.11}$$

with the representations decomposing into the singlet representations $(j_1+j_2) \oplus \cdots \oplus (|j_1-j_2|)$;

$$f_j(c) = c^{-j} \quad \Longrightarrow \quad B^i|j,m\rangle = a_{j+1}\mathbf{B}^i_{j \to j+1}|j+1,m'\rangle\,, \tag{A.12}$$

giving rise to the decreasing chain representations $(|j_1-j_2|) \to \cdots \to (j_1+j_2)$. Thus we get the singlet representations and the multiplet representations (3.26). This result is compatible with the fact that taking the limit and doing tensor product decomposition are non-commutative, since the product of pre-factors $f_j(c)$ from the tensor product of two different $(j)$'s can get cancelled which leads to additional non-vanishing matrix elements of $B$'s.

It is difficult but possible to get general net representations from the Inonu-Wigner contraction. This involves taking the limit of reducible representations of $SO(4)$. One example which was discussed in the main text is the first part of Figure 4. It can be obtained by taking the limit of $(0,0) \oplus (1,1)$ representation of $SO(4)$, as shown in Figure 14.

The above discussions can be easily extend to the Galilean limit $J^{i0} = \lim_{c \to \infty} cB^i_G$ giving rise to similar results. And it is also easy to extend the study to read the representations of general $d$-dimensional CCA/GCA rotation group.

---

[24]In taking the limit $c \to 0$, only the leading-$c$ dependence of $f_j(c)$ matters. The only constraint to get a finite $\mathbf{B}$ matrix is that $c\frac{f_{j+1}(c)}{f_j(c)}$ and $c\frac{f_{j-1}(c)}{f_j(c)}$ should be finite for each $j$. It is possible to get more complicated representations by taking the limit with carefully chosen $f_j(c)$.

$$\square_4 \otimes \square_4 \;=\; \bullet_4 \;\oplus\; \square\square_4 \;\oplus\; \begin{array}{c}\square\\\square\end{array}_4$$

$$\left(\tfrac{1}{2},\tfrac{1}{2}\right) \otimes \left(\tfrac{1}{2},\tfrac{1}{2}\right) \;=\; (0,0) \;\oplus\; (1,1) \;\oplus\; \big((1,0)\oplus(0,1)\big)$$

Figure 14: The rank-2 tensor representation of $SO(4)$ and their Inonu-Wigner contraction of the Carrollian limit.

## B Calculation details of CCFT 2-pt correlation functions

In the main text, we omitted many details in calculating the 2-pt correlators of $4d$ CCFT, and here we present the details. First of all, we list all the Ward identities for the 2-pt correlators:

$$
\begin{aligned}
W(P^\mu) &\equiv (\partial_1^\mu + \partial_2^\mu)\langle\mathcal{O}_1\mathcal{O}_2\rangle = 0\,,\\
W(D) &\equiv (x_1^\mu\partial_1^\mu + x_2^\mu\partial_2^\mu)\langle\mathcal{O}_1\mathcal{O}_2\rangle + \Delta_1\langle\mathcal{O}_1\mathcal{O}_2\rangle + \Delta_2\langle\mathcal{O}_1\mathcal{O}_2\rangle = 0\,,\\
W(J^{ij}) &\equiv ((x_1^i\partial_1^j - x_1^j\partial_1^i) + (x_2^i\partial_2^j - x_2^j\partial_2^i))\langle\mathcal{O}_1\mathcal{O}_2\rangle + \langle(J^{ij}\mathcal{O}_1)\mathcal{O}_2\rangle + \langle\mathcal{O}_1(J^{ij}\mathcal{O}_2)\rangle = 0\,,\\
W(B^i) &\equiv (x_1^i\partial_{t_1} + x_2^i\partial_{t_2})\langle\mathcal{O}_1\mathcal{O}_2\rangle + \langle(B^i\mathcal{O}_1)\mathcal{O}_2\rangle + \langle\mathcal{O}_1(B^i\mathcal{O}_2)\rangle = 0\,,\\
W(K^0) &\equiv (-x_1^i x_1^i\partial_{t_1} - x_2^i x_2^i\partial_{t_2})\langle\mathcal{O}_1\mathcal{O}_2\rangle - 2x_1^i\langle(B^i\mathcal{O}_1)\mathcal{O}_2\rangle - 2x_2^i\langle\mathcal{O}_1(B^i\mathcal{O}_2)\rangle = 0\,,\\
W(K^i) &\equiv ((2x_1^i x_1^\mu\partial_1^\mu - x_1^j x_1^j\partial_1^i) + (2x_2^i x_2^\mu\partial_2^\mu - x_2^j x_2^j\partial_2^i))\langle\mathcal{O}_1\mathcal{O}_2\rangle\\
&\quad + 2\Big(\Delta_1 x_1^i\langle\mathcal{O}_1\mathcal{O}_2\rangle + t_1\langle(B^i\mathcal{O}_1)\mathcal{O}_2\rangle + x_1^j\langle(J^{ij}\mathcal{O}_1)\mathcal{O}_2\rangle\Big)\\
&\quad + 2\Big(\Delta_2 x_2^i\langle\mathcal{O}_1\mathcal{O}_2\rangle + t_2\langle\mathcal{O}_1(B^i\mathcal{O}_2)\rangle + x_2^j\langle\mathcal{O}_1(J^{ij}\mathcal{O}_2)\rangle\Big) = 0\,,
\end{aligned}
\tag{B.1}
$$

where $\mu = 0, 1, \ldots, d-1$, $i, j = 1, \ldots, d-1$. It is immediately noticed that if ignoring all $t_i$ and $\partial_{t_i}$ terms, the Ward identities of $\{D, P^i, K^i, J^{ij}\}$ are just the Ward identities for CFT$_3$.

It is worth mentioning that plugging the Ward identities of $\{D, P^\mu, J^{ij}, B^i\}$ into $W(K^\mu)$ results algebraic equations rather than differential equations. To be more specific, we have

$$
\left\{
\begin{aligned}
0 &= W(K^0) - x_1^j x_2^j W(P^0) + (x_1 + x_2)^i W(B^i)\,,\\
0 &= W(K^i) + \Big(x_1^i x_2^\mu W(P^\mu) + x_2^i x_1^\mu W(P^\mu) - x_1^j x_2^j W(P^0)\Big)\\
&\quad - \Big((x_1 + x_2)^i W(D) - (x_1 + x_2)^0 W(B^i) - (x_1 + x_2)^j W(J^{ij})\Big)\,,
\end{aligned}
\right.
\tag{B.2}
$$

$$
\Rightarrow
\left\{
\begin{aligned}
0 &= -x_{12}^i\big(\mathbf{B}_1^i f_{q_1-1,q_2} - \mathbf{B}_2^i f_{q_1,q_2-1}\big)\,,\\
0 &= x_{12}^i(\Delta_1 - \Delta_2)f_{q_1,q_2} + x_{12}^j\big(\mathbf{J}_1^{ij} f_{q_1,q_2} - \mathbf{J}_2^{ij} f_{q_1,q_2}\big) + t_{12}\big(\mathbf{B}_1^i f_{q_1-1,q_2} - \mathbf{B}_2^i f_{q_1,q_2-1}\big)\,,
\end{aligned}
\right.
\tag{B.3}
$$

where $f$ denotes the 2-pt correlators and $\mathbf{B}_{1,2}$ and $\mathbf{J}_{1,2}$ are the representation matrix acting on $\mathcal{O}_i$ respectively. Thus the Ward identities of $K$'s mostly play the role of giving the constraints or the selection rules after other Ward identities having determined the form of the 2-pt correlators.

In the rest of this appendix, we show how to apply these Ward identities, following the strategy introduced in section 4.2. As in 4d case, we discard the distributional solutions. The discussions could be viewed as a generalization of the discussions of the Ward identities in CFT$_3$. For completeness, we present all the analysis here including the well-known facts on CFT$_3$.

### B.1  Ward identities of $P$ and $D$ generators

To calculate the 2-pt function of a given representation, we first apply $W(P^\mu) = 0$, which requires the 2-pt function to be translational invariant, i.e. it depends only on $x_{12}^\mu = x_1^\mu - x_2^\mu$. Moreover, $W(D) = 0$ requires that the 2-pt function has the form

$$\left\langle \mathcal{O}_1^{(m_1,q_1)}(x_1)\mathcal{O}_2^{(m_2,q_2)}(x_2) \right\rangle = \frac{f_{q_1,q_2}^{m_1,m_2}(\varphi,\theta,\phi)}{R^{\Delta_1+\Delta_2}}, \tag{B.4}$$

where $R \equiv \sqrt{(t_{12})^2 + (\vec{x}_{12})^2}$ and the angular coordinates $\{\varphi, \theta, \phi\}$ are defined as

$$\cot\varphi \equiv t_{12}/|\vec{x}_{12}|, \quad \tan\theta \equiv x_{12}^1/x_{12}^2, \quad \tan\phi \equiv x_{12}^3/\sqrt{(x_{12}^1)^2 + (x_{12}^2)^2}. \tag{B.5}$$

### B.2  Ward identities of $J$ generators — tensor structure $I_{j_1,j_2}^{m_1,m_2}$

In terms of $(R, \varphi, \theta, \phi)$, the differential part of $W(J^{ij})$ are the derivatives of $\theta$ and $\phi$, and applying $W(J^{ij}) = 0$, we can factorize out the dependence of $(\theta, \phi)$ as

$$f_{q_1,q_2}^{m_1,m_2}(\varphi,\theta,\phi) = f_{q_1,q_2}(\varphi) I_{j_1,j_2}^{m_1,m_2}(\theta,\phi). \tag{B.6}$$

We call $I_{j_1,j_2}^{m_1,m_2}$ the tensor structure of the 2-pt correlators, which can be fixed up to some relative coefficients by requiring $W(J^{ij}) = 0$. The Ward identities are reduced to

$$-i\partial_\theta(I_{j_1,j_2}^{m_1,m_2}) + (\mathbf{J}_{j_1})_{m_1,m_1'} I_{j_1,j_2}^{m_1',m_2} + (\mathbf{J}_{j_2})_{m_2,m_2'} I_{j_1,j_2}^{m_1,m_2'} = 0, \tag{B.7}$$

$$\frac{e^{-i\theta}}{\sqrt{2}}(i\partial_\phi \pm \cot\phi\,\partial_\theta)(I_{j_1,j_2}^{m_1,m_2}) + (\mathbf{J}_{j_1}^\pm)_{m_1,m_1'} I_{j_1,j_2}^{m_1',m_2} + (\mathbf{J}_{j_2}^\pm)_{m_2,m_2'} I_{j_1,j_2}^{m_1,m_2'} = 0. \tag{B.8}$$

Equation (B.7) leads to

$$I_{j_1,j_2}^{m_1,m_2} = e^{-i(m_1+m_2)\theta}\, h_{j_1,j_2}^{m_1,m_2}(\phi), \tag{B.9}$$

and then (B.8) further reduces to

$$\mathcal{J}^\pm(h_{j_1,j_2}^{m_1,m_2}) + (\mathbf{J}_{j_1}^\pm)_{m_1,m_1'} h_{j_1,j_2}^{m_1',m_2} + (\mathbf{J}_{j_2}^\pm)_{m_2,m_2'} h_{j_1,j_2}^{m_1,m_2'} = 0, \tag{B.10}$$

with

$$\mathcal{J}^\pm = \frac{i}{\sqrt{2}}(\partial_\phi \mp (m_1+m_2)\cot\phi\,\partial_\theta). \tag{B.11}$$

Let us first consider a simple case that $j_1 = 0$, and thus the actions of $\mathbf{J}_{j_1}$ and $\mathbf{J}_{j_1}^\pm$ vanish. The combinations of (B.10) gives

$$-\left(\mathcal{J}^+(\mathcal{J}^-(h_{0,j_2}^{m_2})) + \mathcal{J}^-(\mathcal{J}^+(h_{0,j_2}^{m_2}))\right) - \left(\mathbf{J}_{j_2}^+\mathbf{J}_{j_2}^- + \mathbf{J}_{j_2}^-\mathbf{J}_{j_2}^+\right)_{m_2,m_2'} h_{0,j_2}^{m_2'} = 0,$$

$$\implies \left(\partial_\phi^2 h_{0,j_2}^{m_2} + \cot\phi\,\partial_\phi h_{0,j_2}^{m_2} - m^2\cot^2\phi\, h_{0,j_2}^{m_2}\right) + \left(\mathbf{C}_{j_2} - \mathbf{J}_{j_2}^2\right)_{m_2,m_2'} h_{0,j_2}^{m_2'} = 0, \tag{B.12}$$

$$\implies \partial_\phi^2 h_{0,j_2}^{m_2} + \cot\phi\,\partial_\phi h_{0,j_2}^{m_2} + \left(j_2(j_2+1) - \frac{m^2}{\sin\phi}\right) h_{0,j_2}^{m_2} = 0.$$

This is exactly the equation for the associated Legendre polynomials and hence $h_{0,j_2}^{m_2} \propto P_j^m(\cos(\phi))$. In our convention, the solution for $I_{0,j_2}^{m_2}$ is similar to the spherical harmonics with different coefficients and dependence on $\theta$

$$I_{0,j_2}^{m_2} = \tilde{Y}_{j_2}^{m_2}, \qquad \tilde{Y}_j^m(\theta,\phi) = i^{(j-m)}\sqrt{\frac{(2j)!(j-m)!}{(j+m)!}}\frac{(-1)^j}{(2j-1)!!}P_j^m(\cos(\phi))e^{-im\theta}. \tag{B.13}$$

The pre-factors are chosen such that the first component is $I_{0,j_2}^{j_2} = e^{-ij_2\theta}\sin^{j_2}\phi$. For example, we have

$$I_{0,1}^{m_2} = \left(e^{-i\theta}\sin\phi, \quad -i\sqrt{2}\cos\phi, \quad e^{i\theta}\sin\phi\right), \tag{B.14}$$

and

$$I_{0,2}^{m_2} = \left(e^{-2i\theta}\sin^2\phi, \quad -2ie^{-i\theta}\sin\phi\cos\phi, \quad \sqrt{\tfrac{2}{3}}(1-3\cos^2\phi), \quad -2ie^{i\theta}\sin\phi\cos\phi, \quad e^{2i\theta}\sin^2\phi\right). \tag{B.15}$$

The tensor structure $I_{0,j_2}^{m_2}$ can be viewed as an $SO(3)$ representation ($j = j_2$). Thus in fact, $I_{j_1,j_2}^{m_1,m_2}$ should be viewed as the tensor product representation $(j_1)\otimes(j_2) = \bigoplus_j(j)$. Using the Clebsch-Gordan coefficients, we can decompose $I_{j_1,j_2}^{m_1,m_2}$ into irreducible ($j$) representations $I_j^m$ with every $I_j^m$ satisfying $J$'s Ward identities for $I_{0,j_2=j}^{m_2=m}$. And thus, $I_j^m = \tilde{Y}_j^m$, and $I_{j_1,j_2}^{m_1,m_2}$ is composed of $\tilde{Y}_j^m$ by using the CG coefficients

$$I_{j_1,j_2}^{m_1,m_2} = \sum_{j,m}\langle j_1,m_1;j_2,m_2|j,m\rangle c_{j_1,j_2}^j \tilde{Y}_j^m, \tag{B.16}$$

where $c_{j_1,j_2}^j$ are the relative coefficients to be determined later.[25] Notice that the tensor product $(j_1)\otimes(j_2) = (j_2)\otimes(j_1)$, we have the exchange symmetry $I_{j_1,j_2}^{m_1,m_2} = I_{j_2,j_1}^{m_2,m_1}$.

Taking $j_1 = j_2 = 1$ as a concrete example, we have

$$I_{1,1} = c_{1,1}^{j=2}\begin{pmatrix} e^{-2i\theta}\sin^2\phi & -i\sqrt{2}e^{-i\theta}\sin\phi\cos\phi & -\frac{1}{6}(3\cos2\phi+1) \\ -i\sqrt{2}e^{-i\theta}\sin\phi\cos\phi & -\frac{1}{3}(3\cos2\phi+1) & -i\sqrt{2}e^{i\theta}\sin\phi\cos\phi \\ -\frac{1}{6}(3\cos2\phi+1) & -i\sqrt{2}e^{i\theta}\sin\phi\cos\phi & e^{2i\theta}\sin^2\phi \end{pmatrix}$$

$$+ c_{1,1}^{j=1}\begin{pmatrix} 0 & -\frac{1}{\sqrt{2}}e^{-i\theta}\sin\phi & i\cos\phi \\ \frac{1}{\sqrt{2}}e^{-i\theta}\sin\phi & 0 & -\frac{1}{\sqrt{2}}e^{i\theta}\sin\phi \\ -i\cos\phi & \frac{1}{\sqrt{2}}e^{i\theta}\sin\phi & 0 \end{pmatrix} + c_{1,1}^{j=0}\begin{pmatrix} 0 & 0 & \frac{1}{\sqrt{3}} \\ 0 & -\frac{1}{\sqrt{3}} & 0 \\ \frac{1}{\sqrt{3}} & 0 & 0 \end{pmatrix}. \tag{B.17}$$

## B.3 Ward identities of $B$ and $K$ generators

Generally speaking, the relative coefficients in $I_{j_1,j_2}^{m_1,m_2}$, and moreover the relative coefficients between different levels of the 2-pt correlators can be totally fixed by considering the Ward identities of $B$ and $K$ generators. Firstly, let us consider the $W(B^i)$. After plugging in the ansatz $f(\varphi) = g(\varphi)\cos^{(\Delta_1+\Delta_2)}\varphi$, the differential parts of $W(B^i)$ are proportional to $\partial_{\cot\varphi}$, which constrain the form of 2-pt correlators

$$\left\langle \mathcal{O}_1^{(m_1,q_1)}(x_1)\mathcal{O}_2^{(m_2,q_2)}(x_2)\right\rangle = \frac{g_{q_1,q_2}(\varphi)I_{j_1,j_2}^{m_1,m_2}(\theta,\phi)}{|\vec{x}_{12}|^{\Delta_1+\Delta_2}}. \tag{B.18}$$

---

[25]The number of $c_{j_1,j_2}^j$'s is $n_c = (j_1+j_2) - |j_1-j_2| + 1$, but the independent number of $c_{j_1,j_2}^j$ is $n_c - 1$ since an overall factor can be absorbed into $f_{q_1,q_2}(\varphi)$. We write down all $c_{j_1,j_2}^j$ for later convenience.

As

$$x^\pm \equiv (ix^1 \pm x^2)/\sqrt{2} = \frac{i}{\sqrt{2}} e^{\mp i\theta} \sin\phi \sin\varphi, \quad x^3 = \cos\phi \sin\varphi, \tag{B.19}$$

we have $(x^+, x^3, x^-) = i/\sqrt{2}\, \tilde{Y}^a_{j=1}$, $a = \pm, 3$, and get the Ward identities of $B^a$

$$\tilde{Y}^a_1 I^{m_1,m_2}_{j_1,j_2} \frac{\partial g_{q_1,q_2}(\varphi)}{\partial \cot\varphi} + (\mathbf{B}^a_1)^{j_1 \to j'_1}_{m_1,m'_1} I^{m'_1,m_2}_{j'_1,j_2} g_{q_1-1,q_2}(\varphi) + (\mathbf{B}^a_2)^{j_2 \to j'_2}_{m_2,m'_2} I^{m_1,m'_2}_{j_1,j'_2} g_{q_1,q_2-1}(\varphi) = 0. \tag{B.20}$$

This gives the constraints on the relative coefficients in the tensor structure $I$ and the explicit forms of $g_{q_1,q_2}$.

We first consider the 2-pt correlators in the trivial case. The 2-pt correlator, as discussed in the main text, is non-vanishing at the highest level, and this non-zero 2-pt correlator reduces to the one in $\mathrm{CFT}_3$ and is independent of $t$ coordinates

$$f = \frac{C\, I^{m_1,m_2}_{j_1,j_2}}{|\vec{x}_{12}|^{\Delta_1 + \Delta_2}}. \tag{B.21}$$

It is easy to check that $W(B^i) = 0$ are satisfied. We have to take into account of the Ward identities of $K^i$. For the trivial case, the first line in (B.3) is trivially satisfied, and the second line leads to

$$x^i_{12}(\Delta_1 - \Delta_2) I_{j_1,j_2} + x^j_{12} \left( \mathbf{J}^{ij}_{j_1} I_{j_1,j_2} - \mathbf{J}^{ij}_{j_1} I_{j_1,j_2} \right) = 0, \qquad \forall i = 1,2,3. \tag{B.22}$$

A combination of these equations gives

$$x^3_{12}(\Delta_1 - \Delta_2) I_{j_1,j_2} - x^-_{12} \left( \mathbf{J}^+_{j_1} I_{j_1,j_2} - \mathbf{J}^+_{j_1} I_{j_1,j_2} \right) - x^+_{12} \left( \mathbf{J}^-_{j_1} I_{j_1,j_2} - \mathbf{J}^-_{j_1} I_{j_1,j_2} \right) = 0. \tag{B.23}$$

Taking $\theta = 0$, i.e. $x^3 = |\vec{x}_{12}| \cos\phi$ and $x^\pm_{12} = 0$ and using the fact $I^{j_1,j_2}_{j_1,j_2} = c^{j_1+j_2}_{j_1,j_2} \left( e^{-i\theta} \sin\phi \right)^{(j_1+j_2)}$, the above constraint becomes

$$c^{j_1+j_2}_{j_1,j_2}(\Delta_1 - \Delta_2) |\vec{x}_{12}| \sin^{(j_1+j_2)}\phi \cos\phi = 0, \tag{B.24}$$

leading to the selection rule $\Delta_1 = \Delta_2$. Furthermore solving the residual equation

$$x^j_{12} \left( \mathbf{J}^{ij}_{j_1} I_{j_1,j_2} - \mathbf{J}^{ij}_{j_1} I_{j_1,j_2} \right) = 0, \qquad \forall i = 1,2,3, \tag{B.25}$$

gives rise to the selection rule $j_1 = j_2$, and fix all the relative coefficients in $I_{j_1,j_2}$ as

$$c^j_{j_1=j_2} = (-1)^{j_1} \left( \frac{i}{\sqrt{2}} \right)^j \sqrt{\frac{(2j+1)!!}{j!} \frac{(-j+2j_1-1)!! \, (j+2j_1)!!}{(-j+2j_1)!! \, (j+2j_1+1)!!}}, \quad \text{for } 0 \le j \le 2j_1 \text{ and } j \text{ even},$$

$$c^j_{j_1=j_2} = 0, \quad \text{otherwise}. \tag{B.26}$$

The coefficients are chosen such that $c^{j=2j_1}_{j_1=j_2} = 1$. This is exactly the form of the 2-pt correlators in $\mathrm{CFT}_3$. Especially, taking $(c^0_{j_1=j_2=1} = -1/\sqrt{3},\ c^1_{j_1=j_2=1} = 0,\ c^2_{j_1=j_2=1} = 1)$ in (B.17), $I^{m_1,m_2}_{1,1}$ matches exactly with $I^{\mu\nu} = \delta^{\mu\nu} - 2\frac{x^\mu x^\nu}{x^2}$ in [6] for $d = 3$. [26]

Next, we turn to a little more complicated case and try to prove (4.26). For simplicity, consider $\mathcal{O}_1 \in (j_0 + 1)$ and $\mathcal{O}_2 \in (j_0) \to (j_0 + 1)$. From the above discussions, we learn that $f_{1,1} = C I_{j_0+1,j_0+1} |\vec{x}|^{-2\Delta}$ with $\Delta \equiv \Delta_1 = \Delta_2$, and thus the $B$'s Ward identities on $f_{1,2}$ are

$$\tilde{Y}^a_1 I^{m_1,m_2}_{j_0,j_0+1} \frac{\partial g_{1,2}(\varphi)}{\partial \cot\varphi} + C(\mathbf{B}^a_2)^{j_0+1 \to j_0}_{m_2,m'_2} I^{m_1,m'_2}_{j_0+1,j_0+1} = 0. \tag{B.27}$$

---

[26] This could be easily checked using a basis change: $I^{m_1=\pm 1,*}_{1,1} = \frac{1}{\sqrt{2}}(iI^{\mu=1,*} \pm I^{\mu=2,*})$ and $I^{m_1=0,*}_{1,1} = iI^{\mu=3,*}$, and a similar basis change between $m_2$ and $\nu$.

Since both $\tilde{Y}$'s and $I$'s depend only on $(\theta, \phi)$ and $\mathbf{B}$'s are purely numbers, the equation could be satisfied only if

$$g_{1,2} = C \cot \varphi + C' \, . \tag{B.28}$$

If $C \neq 0$, the above equation becomes

$$\tilde{Y}_1^a I_{j_0, j_0+1}^{m_1, m_2} + (\mathbf{B}_2^a)_{m_2, m_2'}^{j_0+1 \to j_0} I_{j_0+1, j_0+1}^{m_1, m_2'} = 0 \, . \tag{B.29}$$

Similar to the spherical harmonics, $\tilde{Y}$'s enjoy the construction rule

$$\tilde{Y}_1^a \, \tilde{Y}_j^m = \tilde{Y}_{j+1}^{m+a} + \frac{2j}{\sqrt{4j^2-1}} \, \tilde{Y}_{j-1}^{m+a} \, , \tag{B.30}$$

and thus by (B.16), the above equations become

$$\begin{aligned}
&\sum_{j,m} \langle j_0, m_1; j_0 + 1, m_2 | j, m \rangle \, c_{j_0, j_0+1}^j \left( \tilde{Y}_{j+1}^{m+a} + \frac{2j}{\sqrt{4j^2-1}} \, \tilde{Y}_{j-1}^{m+a} \right) \\
&+ (\mathbf{B}_2^a)_{m_2, m_2'}^{j_0+1 \to j_0} \sum_{j,m} \langle j_0 + 1, m_1; j_0 + 1, m_2 | j, m \rangle \, c_{j_0+1, j_0+1}^j \, \tilde{Y}_j^m = 0 \, ,
\end{aligned} \tag{B.31}$$

where $c_{j_0+1, j_0+1}^j$ are determined as (B.26) and $c_{j_0, j_0+1}^j$ are to be determined. However, there are $2j_0+3$ equations for each $j$ in the second term in (B.31), and thus it is over-constrained for $c_{j_0, j_0+1}^j$ which has $2j_0+1$ components. In fact, there is indeed no solution to (B.31). Therefore $C$ must be vanishing in (B.28), which requires $f_{1,2}$ being independent of $t$. Using the $K$'s Ward identities, we find $f_{1,2} = 0$, and thus we have proved (4.26).

In short summary, the differential parts of $B$'s Ward identities give the constraints on the $\varphi$ dependence, and together with the $B$'s unidirectional action on the representations, makes $g_{q_1, q_2}(\varphi)$ be the polynomials in $\cot \varphi = t_{12}/|\vec{x}_{12}|$ of order at most $(q_1 + q_2 - 2)$. The tensor structure part then gives the constraints on the relative coefficients in the tensor structure.

The simplest non-trivial case is that $\mathcal{O}_1 \in (1) \to (0)$ and $\mathcal{O}_2 \in (0) \to (1)$. As analysed in the main text, the Ward identities on the level-2 correlators suggest that $f_{1,2}$ and $f_{2,1}$ are reduced to the correlators of $\text{CFT}_3$ with $\Delta_1 = \Delta_2$, and $f_{1,1} = 0$. The Ward identities of $B$'s (B.20) on the level-3 correlator require $f_{2,2}$ to be linear in $t_{12}$

$$f_{2,2} = \frac{C \, t_{12}/|\vec{x}_{12}| I_{1,0} + C_0^{(1,0)} \, \tilde{I}_{1,0}}{|\vec{x}_{12}|^{\Delta_1 + \Delta_2}} \, , \tag{B.32}$$

and further fix the relative coefficients in $I_{1,0}$ as $c_{1,0}^1 = i\sqrt{2}$. It follows that the Ward identities of $K^i$ (B.3) lead to

$$x_{12}^j \left( \mathbf{J}_1^{ij} f_{2,2} - \mathbf{J}_2^{ij} f_{2,2} \right) + t_{12}^i \left( \mathbf{B}_1^i f_{1,2} - \mathbf{B}_2^i f_{2,1} \right) = 0 \, . \tag{B.33}$$

It is obvious that $C_0^{(1,0)}$ should vanish, and the resulting 2-pt correlators are (4.32).

We can further extend these discussions to the chain representation case that

$$\begin{aligned}
\mathcal{O}_1 &\in (j_0 + n) \to \cdots \to (j_0 + 1) \to (j_0) \, , \\
\mathcal{O}_2 &\in (j_0) \to (j_0 + 1) \to \cdots \to (j_0 + n) \, , \quad j_0 \in \mathbb{N} \, .
\end{aligned} \tag{B.34}$$

The same argument on the 2-pt correlators level by level by using (B.33) suggests that the 2-pt correlators are the power-law functions of $t_{12}/|\vec{x}_{12}|$ and thus the Ward identities require the 2-pt correlators being

$$f_{q_1, q_2} = \frac{C}{|\vec{x}_{12}|^{\Delta_1 + \Delta_2}} \left( \frac{t_{12}}{|\vec{x}_{12}|} \right)^{(q_1 + q_2 - n)} I_{j_1, j_2} \, , \qquad \text{for } q_1 + q_2 \geq n \, , \tag{B.35}$$

with the relative coefficients in $I_{j_1,j_2}$ being

$$
c_{j_1,j_2}^j = \left(\frac{i}{\sqrt{2}}\right)^j \frac{(-1)^{j_2}}{|j_1-j_2|!} \sqrt{\frac{(2j+1)!!}{j!}\frac{(j+|j_1-j_2|)!}{(j-|j_1-j_2|)!}\frac{(-j+j_1+j_2-1)!!}{(-j+j_1+j_2)!!}\frac{(j+j_1+j_2)!!}{(j+j_1+j_2+1)!!}},
$$
$$
\text{for } |j_1-j_2| \leq j \leq j_1+j_2, \text{ and } (j-|j_1-j_2|) \text{ even,}
$$
$$
c_{j_1,j_2}^j = 0, \quad \text{otherwise.}
$$
$$(B.36)$$

One explicit example was shown in (4.35). Together with the special case (4.39), we have given rigorous proof for all 2-pt correlators of chain representations.

However it is hard to give a general result for the operators in net representations. In the case of CFT, by the (finite) Ward identity on the 2-pt correlator, $I_{j_1,j_2}$ is invariant in the sense that

$$
I_{j_1,j_2}^{m_1,m_2} = (\rho_1)_{m_1}^{m_1'} \left(I_{j_1,j_2}^{m_1',m_2'}\right) (\rho_2)_{m_2}^{m_2'}, \tag{B.37}
$$

with suitable choice of coordinates, where $\rho_i$ is the irreducible $SO(d)$ representation matrix on $\mathcal{O}_i$, and $\rho^R$ represents "reflected" representation under inversion $i_E : x^\mu \to \frac{x^\mu}{x^2}$ [6]. Using the Schur lemma, we have $\rho_1 = (\rho_2^R)^{-1}$ and thus the selection rule $j_1 = j_2$. In the CCFT case, $g_{q_1,q_2} I_{j_1,j_2}^{m_1,m_2}$ is invariant similarly

$$
g_{q_1,q_2} I_{j_1,j_2}^{m_1,m_2} = (\rho_1)_{q_1',m_1}^{q_1,m_1'} \left(g_{q_1',q_2'} I_{j_1,j_2}^{m_1',m_2'}\right) (\rho_2)_{q_2',m_2}^{q_2,m_2'}, \tag{B.38}
$$

with suitable choice of coordinates, where $\rho_i$ is the representation matrix of CCA rotation on $\mathcal{O}_i$. However, the Schur lemma can not be applied since $\rho_i$'s are not irreducible. The forms of 2-pt correlators of net representations have to be determined case by case, without a general form.

## C  Three-dimensional CCFT

In the main text, we focus on $d = 4$ CCFT, and we can extend the discussions to other dimensions as well. As we can expect, the representations of CCA rotation in $d$ dimension are generally the multiplet representations that every sub-sector is an $SO(d-1)$ irreducible representation, and the boost generators $B$'s map the sub-sectors to the sub-sectors. In particular, for the dimension higher than four, the general representations of CCA rotation are intrinsically higher dimensional net representations. To be more specific, taking $d = 5$ for example, there are two spins $(j_1, j_2)$ labeling the $SO(4)$ representations, thus the $B$ generators form a $(\frac{1}{2}, \frac{1}{2})$ representation of $SO(4)$. Since we have

$$
(j_1,j_2) \otimes \left(\frac{1}{2},\frac{1}{2}\right) = \left(j_1+\frac{1}{2},j_2+\frac{1}{2}\right) \oplus \left(j_1+\frac{1}{2},j_2-\frac{1}{2}\right) \oplus \left(j_1-\frac{1}{2},j_2+\frac{1}{2}\right) \oplus \left(j_1-\frac{1}{2},j_2-\frac{1}{2}\right), \tag{C.1}
$$

the actions of $B$ generators should be organized as a three-dimensional net as shown in Figure 15. Together with the non-diagonalizable structure to be introduced later in this appendix, the general net representation of $d$-dimensional CCA rotation could be wildly complicated.

In this appendix we will discuss CCFT$_3$. The three-dimensional ($3d$) CCA is of special interest, as its infinitely extension is isomorphic to BMS$_4$ [32]. Firstly, we try to construct the finite dimensional representations following the strategy in the main text, where the representations should be based on $SO(2)$ representations. It should be reminded that since the irreducible representations of $SO(2)$ are 1-dimensional, the appearance of a $j = -1$ operator

is not guaranteed even if there is a $j = 1$ operator. For example,

$$
\begin{aligned}
B^+ \mathcal{O}^{(0,2)} &= \mathcal{O}^{(1,1)}, & B^- \mathcal{O}^{(0,2)} &= 0, \\
B^+ \mathcal{O}^{(1,1)} &= 0, & B^- \mathcal{O}^{(1,1)} &= 0,
\end{aligned}
\qquad
\begin{aligned}
&\mathcal{O}^{(0,2)} \\
&\;\downarrow B^+. \\
&\mathcal{O}^{(1,1)}
\end{aligned}
\tag{C.2}
$$

Here the notation $\mathcal{O}^{(j,q)}$ following the main text means that the operator $\mathcal{O}$ has spin $j$ under the rotation $J = -iJ^{12}$ and is at order $q$ in a multiplet. As shown in Figure 16, one may view the irreducible finite representations $\mathcal{O}^{j,b^+,b^-}$ as a set of points on a $2d$ grid connected by $B^\pm$ with the relations

$$
\mathcal{O}^{j,b^+,b^-} = \left(B^+\right)^{b^+} \left(B^-\right)^{b^-} \mathcal{O}^{j,0,0}, \qquad J \mathcal{O}^{j,b^+,b^-} = (j + b^+ - b^-) \mathcal{O}^{j,b^+,b^-}. \tag{C.3}
$$

However, the story does not end here, and there are much more complicated representations for the 3-dimensional CCA rotation. For example, consider the representation consisting of $\{\mathcal{O}^{(1,3)}, \mathcal{O}^{(-1,3)}, \mathcal{O}^{(2,2)}, \mathcal{O}_1^{(0,2)}, \mathcal{O}_2^{(0,2)}, \mathcal{O}^{(-2,2)}, \mathcal{O}^{(1,1)}, \mathcal{O}^{(-1,1)}\}$ with the relations

$$
\begin{aligned}
B^+ \mathcal{O}^{(1,3)} &= \mathcal{O}^{(2,2)}, & B^- \mathcal{O}^{(1,3)} &= \frac{1}{\sqrt{2}}(-\mathcal{O}_1^{(0,2)} + \sqrt{3}\mathcal{O}_2^{(0,2)}), \\[2mm]
B^+ \mathcal{O}^{(-1,3)} &= \frac{1}{\sqrt{2}}(\mathcal{O}_1^{(0,2)} + \sqrt{3}\mathcal{O}_2^{(0,2)}), & B^- \mathcal{O}^{(-1,3)} &= -\mathcal{O}^{(-2,2)}, \\[2mm]
B^+ \mathcal{O}^{(2,2)} &= 0, & B^- \mathcal{O}^{(2,2)} &= \mathcal{O}^{(1,1)}, \\[1mm]
B^+ \mathcal{O}_1^{(0,2)} &= \frac{1}{\sqrt{2}} \mathcal{O}^{(1,1)}, & B^- \mathcal{O}_1^{(0,2)} &= \frac{1}{\sqrt{2}} \mathcal{O}^{(-1,1)}, \\[2mm]
B^+ \mathcal{O}_2^{(0,2)} &= \sqrt{\frac{3}{2}} \mathcal{O}^{(1,1)}, & B^- \mathcal{O}_2^{(0,2)} &= -\sqrt{\frac{3}{2}} \mathcal{O}^{(-1,1)}, \\[1mm]
B^+ \mathcal{O}^{(-2,2)} &= \mathcal{O}^{(-1,1)}, & B^- \mathcal{O}^{(-2,2)} &= 0, \\[2mm]
B^+ \mathcal{O}^{(1,1)} &= 0, & B^- \mathcal{O}^{(1,1)} &= 0, \\[1mm]
B^+ \mathcal{O}^{(-1,1)} &= 0, & B^- \mathcal{O}^{(-1,1)} &= 0.
\end{aligned}
\tag{C.4}
$$

In the graphic language, this representation is shown in Figure 17. It is impossible to redefine $\mathcal{O}_1^{(0,2)}$ and $\mathcal{O}_2^{(0,2)}$ by basis change to fit them into the grid in Figure 16, and this representation is essentially a 3-dimensional net. We call this a non-diagonalizable structure in the sense that

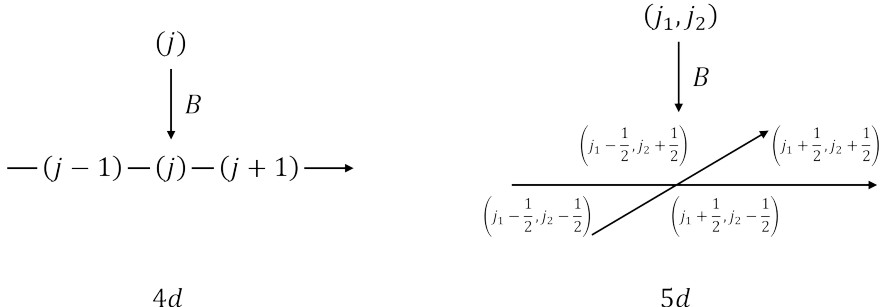

$4d$ $\qquad\qquad\qquad\qquad\qquad\qquad$ $5d$

Figure 15: Since the $SO(4)$ representations are labeled by $(j_1, j_2)$, which form a 2-dimensional structure, the general net structure of $5d$ CCA representation should be three dimensional.

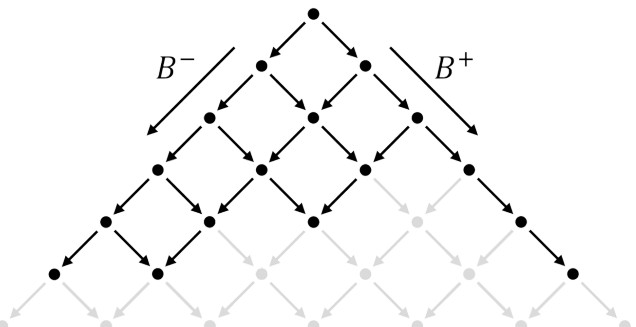

Figure 16: One example of 3$d$ CCA representation where the black dots representing the operators of this representation fitting in the 2$d$ grid shown in gray.

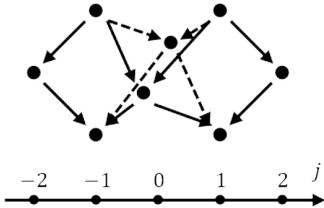

Figure 17: The representation in (C.4). The points represent the operators and the axis labels their spin. This is exactly the second part in the rank-3 tensor representation shown in Figure 20.

we can not diagonalize $\mathcal{O}_1^{(0,2)}$ and $\mathcal{O}_2^{(0,2)}$ to simplify the structure of the representation. More generally, a representations of 3-dimensional CCA rotation could be a higher dimensional net if there is nesting of such non-diagonalizable structures.

The reason this non-diagonalizable structure appearing is that $B$ generators form a vector representation under $SO(d-1)$, and for $d=3$, $B^{\pm}$ are two independent 1-dimensional representations. Thus the coefficients $c^{\pm}$ in $B^{\pm}\mathcal{O}^q = c^{\pm}\mathcal{O}^{q-1}$ are arbitrary, leading to the non-diagonalizable structure. This is a general feature for $d>2$. It is easy to construct such a non-diagonalizable structure in the representations of general $d>2$ CCA rotation. A concrete example for $d=4$ appears in the rank-4 tensor representation, as shown in Figure 18.

The general structure of the representation of 3-dimensional CCA rotation is loosely constrained by the structure of the algebra. The relation $[J,B^{\pm}]=\pm B^{\pm}$ requires that the action of $B^{\pm}$ increase or decrease $j$ by 1, which is satisfied by construction. The relation $[B^+,B^-]=0$ requires a diamond structure $\diamond$ or a straight chain structure for three successive levels with suitable coefficients for the $B$ actions. This means that there do not exist the folded-line structure like $\langle$ or $\rangle$ in the net representations. For example, if there exist a $\mathcal{O}^{j+1}$ that satisfies $B^-B^+\mathcal{O}_2^j \propto B^-\mathcal{O}^{j+1} \propto \mathcal{O}_1^j$, there must exist a $\mathcal{O}^{j-1}$ satisfying $B^+B^-\mathcal{O}_2^j \propto B^+\mathcal{O}^{j-1} \propto \mathcal{O}_1^j$, and together forming a diamond shape to respect the commutator $[B^+,B^-]=0$, as shown in Figure 19.

As in the example given in (C.4), the non-diagonalizable structure are allowed. In this case, although the structure of the representation is not strictly a diamond, we call it a generalized diamond in the sense that projecting the second representation in Figure 17 to the grid in Figure 16, it has the structure of diamonds.

The matrix representation of $J$ and $B^{\pm}$ could be chosen to be equal to the case in 4$d$ and have the form given in section 3.3. However, since the $SO(2)$ sub-sector is 1-dimensional, there is a redefining degree of freedom which may lead to the change of the matrices of $B^{\pm}$

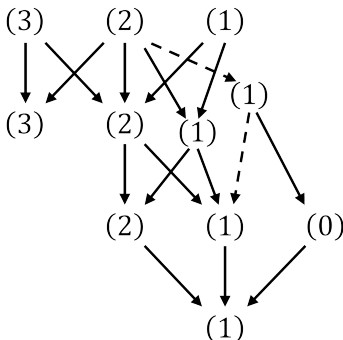

Figure 18: The 45-dimensional indecomposable rank-4 tensor representation of $4d$ CCA rotations. There are two (1) sectors at order 3 that can not be diagonalized. The rank-4 tensor representation appears in the decomposition of the tensor product: $((3) \rightarrow (1))^{\otimes 4} = [35] \oplus 3 \times [45] \oplus 2 \times [20] \oplus 3 \times [15] \oplus [1]$, where $[dim]$ represents a $dim$-dimensional representation.

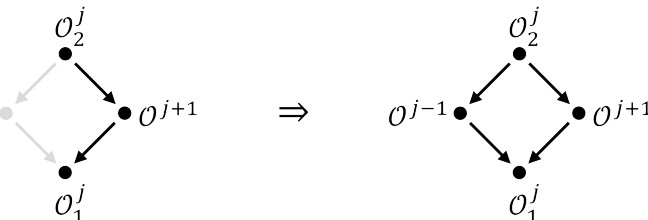

Figure 19: The commutation relation $[B^+, B^-] = 0$ completes the folded-line structure to a diamond structure.

generators. Thus the $J$ matrix of a representation is diagonal with the elements being the spins, and the $B^\pm$ matrices do not admit a fixed form. They could be of the same form as in section 3.3 or not as long as they are consistent with the algebra structure.

In summary, the representations of $3d$ CCA can be organized into a higher dimension net with 1-dimensional representations living on the grid points, and unidirectionally connected by $B$ generators. The commutation relation of $[B^+, B^-] = 0$ should be respect, leading to the generalized diamond structure or straight line structure.

Similar to the $4d$ case, the calculation of the correlation functions of chain representations are easier and would give us guide to the computations for net representations. For the short chains of rank 2, different from the $4d$ case, there are only increasing or decreasing chains since there is no $B^3$ generator. For the longer chains, the constraints from $[B^+, B^-] = 0$ require that only the increasing and decreasing chains (straight chains fitting in Figure 16) are acceptable, otherwise if it contains the increasing and decreasing structure at the same time, it would be completed to a net representation for self-consistency as Figure 19.

For physical application, the vector representation can be defined by taking the limit from the vector representation of $SO(3)$, and further using tensor product one can determine the higher-rank tensor representations.[27] Although the vector representation of $SO(2)$ itself is reducible and decomposable, with the help of $B$ generators, the vector representation of $3d$ CCA is indecomposable. The first three lowest-rank tensor representations are shown in Figure 20. And as discussed above, at rank 3 the second irreducible representation have a 3-dimensional structure, where the actions of $B$ operators are exactly the same as (C.4), leading to the 3-dimensional structure.

---

[27]The tensor representations could also be get by taking ultra-relativistic limit as we did in $d = 4$ in Appendix A. Although similarly, the tensor representations may come from a reducible representation of $SO(3)$.

Figure 20: The first three lowest-rank tensor representations. The axes beneath every irreducible representations show the spins, and the arrows label the connection by $B$ generators.

Based on the representations of $3d$ CCA rotation $\{J, B^\pm\}$, the highest weight representations and the local operators can be defined naturally. The primary operators of $\text{CCFT}_3$ are labeled by four quantum numbers: $\{\Delta, j, r, q\}$ with $\Delta$ and $j$ being the quantum numbers of $D$ and $J = -iJ^{12}$ respectively, $q$ being the order in the multiplet, and $r$ being the total rank of the multiplet. The actions of symmetry generators on a primary operator are

$$\text{primary } \mathcal{O}^{(m,q)}: \quad [D, \mathcal{O}^{(m,q)}] = \Delta \mathcal{O}^{(m,q)}, \quad [J, \mathcal{O}^{(m,q)}] = m\mathcal{O}^{(m,q)}, \quad [K^\mu, \mathcal{O}^{(m,q)}] = 0,$$

$$\text{descendants}: \quad [P^\mu, \mathcal{O}^{(m,q)}] = \partial^\mu \mathcal{O}^{(m,q)}, \quad \text{multiplet index } [B^\pm, \mathcal{O}^{(m,q)}] \propto \mathcal{O}^{(m\pm 1, q-1)}, \tag{C.5}$$

and the local operators are defined as

$$\mathcal{O}(x) = U\mathcal{O}(0)U^{-1}, \quad U = \exp(x^\mu P^\mu), \quad \mu = 0, 1, 2. \tag{C.6}$$

With the primary operators being well defined, we can further consider the correlators by the constraints from the Ward identities. It turns out that the 2-point correlators of $\text{CCFT}_3$ have very similar structures with the ones in $4d$, except the tensor structures are much simpler. The non-zero 2-pt of chain representations are those with the operators in the representations having inverse increasing and decreasing structures at the top levels. The 2-pt correlators consist of the powers of $|\vec{x}_{12}|$ representing the scaling behavior, the tensor structure $I^{j_1, j_2}$, and the powers of $t_{12}/|\vec{x}_{12}|$ with the order being related to the levels of the 2-pt correlators[28]

$$\langle \mathcal{O}_1 \mathcal{O}_2 \rangle = \frac{C \, (t_{12}/|\vec{x}_{12}|)^n \, I^{j_1, j_2}}{|\vec{x}_{12}|^{(\Delta_1 + \Delta_2)}}, \quad \text{with } \Delta_1 = \Delta_2, \tag{C.7}$$

where $C$ is the 2-pt coefficient. The tensor structure can be fixed by $J$ generator

$$I^{j_1, j_2} = \exp(-i(j_1 + j_2)\theta), \tag{C.8}$$

where $\theta$ is the angular coordinate with $\cot\theta = x_{12}^1/x_{12}^2$. For example, for the operators

$$\mathcal{O}_1 \in j+2 \to j+1 \to j, \quad \mathcal{O}_2 \in j \to j+1 \to j+2, \quad \text{with } \Delta_1 = \Delta_2 = \Delta, \tag{C.9}$$

---

[28]Here we discard the distributional solution, as in $4d$ case.

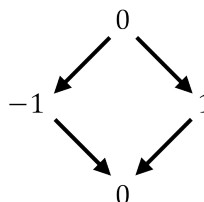

Figure 21: The representation behaves similar to $(0) \to (1) \to (0)$ representation in $4d$.

we have the following correlators

Level 5:
$$f_{3,3}^{j+2,j} = \frac{-C\, t_{12}^2/|\vec{x}_{12}|^2\, e^{-i(2j+3)\theta}}{|\vec{x}_{12}|^{2\Delta}},$$

Level 4:
$$f_{2,3}^{j+1,j} = \frac{C\, i\sqrt{2}\, t_{12}/|\vec{x}_{12}|\, e^{-i(2j+1)\theta}}{|\vec{x}_{12}|^{2\Delta}}, \qquad f_{3,2}^{j+2,j+1} = \frac{C\, i2\sqrt{3}\, t_{12}/|\vec{x}_{12}|\, e^{-i(2j+3)\theta}}{|\vec{x}_{12}|^{2\Delta}},$$

Level 3:
$$f_{1,3}^{j,j} = \frac{C\, e^{-i(2j)\theta}}{|\vec{x}_{12}|^{2\Delta}}, \qquad f_{2,2}^{j+1,j+1} = \frac{C\, e^{-i(2j+2)\theta}}{|\vec{x}_{12}|^{2\Delta}}, \qquad f_{3,1}^{j+2,j+2} = \frac{C\, e^{-i(2j+4)\theta}}{|\vec{x}_{12}|^{2\Delta}},$$

Level 2:
$$f_{1,2}^{j,j+1} = 0, \qquad f_{2,1}^{j+1,j+2} = 0,$$

Level 1:
$$f_{1,1}^{j,j+2} = 0.$$

(C.10)

Here the notation follow the one in the main text, where $f_{q_1,q_2}^{j_1,j_2}$ is the 2-pt correlator, $j$ (without bracket) represents an 1-dimensional representation of $SO(2)$ with spin-$j$ and $j_1 \to j_2 \to \dots$ labels the chain representation with the arrows labeling the action of $B$ generators.

In CCFT$_3$, there exists a special case similar to the $(0) \to (1) \to (0)$ representation in $4d$ that admit basis change. It is now a net representation, as shown in Figure. 21. The highest level 2-pt correlator of the operators in this representation contains an extra undetermined 2-pt coefficient $C'$ as well.

Level 5:
$$f_{3,3}^{0,0} = \frac{-2C + C'}{|\vec{x}_{12}|^{2\Delta}},$$

Level 4:
$$f_{2,3}^{1,0} = \frac{C\, i\sqrt{2}\, t_{12}/|\vec{x}_{12}|\, e^{-i\theta}}{|\vec{x}_{12}|^{2\Delta}}, \quad f_{2,3}^{-1,0} = \frac{C\, i\sqrt{2}\, t_{12}/|\vec{x}_{12}|\, e^{i\theta}}{|\vec{x}_{12}|^{2\Delta}},$$
$$f_{3,2}^{0,1} = \frac{C\, i\sqrt{2}\, t_{12}/|\vec{x}_{12}|\, e^{-i\theta}}{|\vec{x}_{12}|^{2\Delta}}, \quad f_{3,2}^{0,-1} = \frac{C\, i\sqrt{2}\, t_{12}/|\vec{x}_{12}|\, e^{i\theta}}{|\vec{x}_{12}|^{2\Delta}},$$

Level 3:
$$f_{1,3}^{0,0} = \frac{C}{|\vec{x}_{12}|^{2\Delta}}, \quad f_{2,2}^{1,1} = \frac{C\, e^{-2i\theta}}{|\vec{x}_{12}|^{2\Delta}}, \quad f_{2,2}^{1,-1} = 0,$$
$$f_{2,2}^{-1,1} = 0, \qquad f_{2,2}^{-1,-1} = \frac{C\, e^{2i\theta}}{|\vec{x}_{12}|^{2\Delta}}, \quad f_{3,1}^{0,0} = \frac{C}{|\vec{x}_{12}|^{2\Delta}},$$

(C.11)

Level 2:
$$f_{1,2}^{0,1} = 0, \quad f_{1,2}^{0,-1} = 0, \quad f_{2,1}^{1,0} = 0, \quad f_{2,1}^{-1,0} = 0,$$

Level 1:
$$f_{1,1}^{0,0} = 0.$$

$C'$ can be canceled by a change of basis similar to the equation (4.40). There are in fact many other similar cases, in which the structure of 2-pt correlator can take the standard power-law form as long as one carefully define the operators.

As discussed in section 3.5, the representation of the CCA rotation could be applied to the $3d$ GCA rotation, and consequently define the highest weight representation of $\text{GCFT}_3$. Moreover we can also discuss the correlation functions of $\text{GCFT}_3$. The discussion is quite similar, and we would not like to go into the details.

## D CCFTs as null defects of Lorentzian CFT

The conformal defects in Euclidean CFT and equivalently the timelike conformal defects in Lorentzian CFT have been intensively studied in recent years, see e.g. [69–75] and the analytic studies in [76–79]. In this appendix, we show that the residual symmetry algebra of the null conformal defects in Lorentzian CFTs is composed of two parts $\mathbb{R} \ltimes \mathfrak{iso}(d, 1)$, one of which can be identified as the higher dimensional CCA, and the other of which is the outer automorphism of CCA.[29] Hence for a Lorentzian CFT coupled with null defect, the local excitations living on the defect can be described by a CCFT in principle.

We would like to emphasize that this perspective on CCA is different from those in the previous literature, including the asymptotic symmetries at the null infinity of flat spacetime - the BMS algebra, and the conformal transformations of the flat Carrollian manifold - the infinitely extended CCA, see e.g. [30, 32, 37]. There are two distinctions: firstly the residual symmetry algebra of the null defect is finite-dimensional, while the BMS algebra and the infinitely extended CCA are infinite-dimensional; secondly there is an additional generator inherited from the dilatation symmetry of the Lorentzian CFT, and it acts as the outer automorphism of the CCA.

Mathematically, this outer automorphism is due to the non-semisimple nature of the CCA, contrary to the fact that the outer automorphism group of the semisimple Lorentzian conformal algebra is a finite group. From the holography viewpoint, in AdS/CFT the AdS radius $R$ serves as an infrared regulator, leading to the discrete spectrum. The masses of the bulk states combined with $R$ are related to the scaling dimensions of CFT states by mass-dimension relations, see e.g. [80]. While in the proposed BMS holography [55], there is no intrinsic length scale playing the role of the AdS radius and the bulk geometry itself is dilatation invariant, then this bulk dilatation acts on the null infinity as the outer automorphism of the BMS algebra. We leave its physical implications for further study.

Starting from a $(d+1)$-dimensional Lorentzian CFT, there are two types of codimensional one null hyper-surface in $\mathbb{R}^{d,1}$: the null-plane and the null-cone, which are equivalent upto a bulk conformal transformation. We focus on the null-plane case and demonstrate that the residual symmetries on the null-plane are the CCA plus the outer automorphism. Following the convention of [6], we denote the vector field of $(d+1)$-dimensional Lorentzian conformal symmetry as in Table 5, where the superscript "$L$" of the generators indicates the Lorentzian signature $g = \text{diag}\{\overset{-1}{-1}, \overset{0}{1}, \overset{1}{1}, \ldots, \overset{d-1}{1}\}$ and $\alpha, \beta = -1, 0, 1, \ldots, d-1$.

We choose the null-plane as $\mathbf{P} : x^{-1} - x^0 = 0$ and use the lightcone coordinates $x^{\pm} = \frac{1}{\sqrt{2}}(x^{-1} \pm x^0)$. Apparently the following generators preserve this null-plane

---

[29]For a related discussion in two dimension, see [26].

Table 5: Action of Lorentzian conformal symmetry generators as vector fields on the spacetime.

| generator | vector field | finite transformation |
|:---------:|:------------:|:---------------------:|
| $\hat{d}^L$ | $x^\alpha \partial_\alpha$ | $\lambda x^\alpha$ |
| $\hat{p}^L_\alpha$ | $\partial_\alpha$ | $x^\alpha + x^\alpha_0$ |
| $\hat{k}^L_\alpha$ | $2x_\alpha x^\beta \partial_\beta - x^2 \partial_\alpha$ | $\frac{x^A - a^A x^2}{1 - 2a\cdot x + a^2 x^2}$ |
| $\hat{m}^L_{\alpha\beta}$ | $x_\alpha \partial_\beta - x_\beta \partial_\alpha$ | $\Lambda \cdot x$ |

$$
\begin{aligned}
\hat{p}^0 &\equiv \hat{p}^L_+ = \partial_+ := \frac{1}{\sqrt{2}}(\partial_{-1} + \partial_0), \\
\hat{p}^i &\equiv \hat{p}^L_i = \partial_i, \\
\hat{d} &\equiv (\hat{d}^L)|_{\mathbf{P}} = x^+ \partial_+ + 0\partial_- + x^i \partial_i, \\
\hat{m}^{ij} &\equiv \hat{m}^L_{ij} = x_i \partial_j - x_j \partial_i, \\
\hat{k}^i &\equiv (\hat{k}^L_i)|_{\mathbf{P}} = 2x_i(x^+ \partial_+ + 0\partial_- + x^i \partial_i) - (2x^+ \times 0 + \vec{x}^2)\partial_i.
\end{aligned}
\tag{D.1}
$$

There are other residual symmetries: the combinations of $\hat{m}^L_{-1i}$ and $\hat{m}^L_{0i}$ corresponding to the Carrollian boosts

$$
\left.\begin{aligned}
(\hat{m}^L_{-1i})|_{\mathbf{P}} &= \frac{1}{\sqrt{2}}(-x^+ \partial_i - 0\partial_i - x^i \partial_+ - x^i \partial_-), \\
(\hat{m}^L_{0i})|_{\mathbf{P}} &= \frac{1}{\sqrt{2}}(x^+ \partial_i - 0\partial_i - x^i \partial_+ + x^i \partial_-),
\end{aligned}\right\} \Rightarrow \hat{b}^i = -\frac{1}{\sqrt{2}}(\hat{m}^L_{-1i} + \hat{m}^L_{0i})|_{\mathbf{P}} = x^i \partial_+, \quad \text{(D.2)}
$$

and the combination of $\hat{k}^L_{-1}$ and $\hat{k}^L_0$ corresponding to the temporal SCT $\hat{k}^0$ in CCA

$$
\left.\begin{aligned}
(\hat{k}^L_{-1})|_{\mathbf{P}} &= \sqrt{2}(-x^+ - 0)(x^+ \partial_+ + 0\partial_- + x^i \partial_i) - \frac{1}{\sqrt{2}}(2x^+ \times 0 + \vec{x}^2)(\partial_+ + \partial_-), \\
(\hat{k}^L_0)|_{\mathbf{P}} &= \sqrt{2}(x^+ - 0)(x^+ \partial_+ + 0\partial_- + x^i \partial_i) - \frac{1}{\sqrt{2}}(2x^+ \times 0 + \vec{x}^2)(\partial_+ - \partial_-),
\end{aligned}\right\}
\tag{D.3}
$$

$$
\implies \hat{k}^0 = \frac{1}{\sqrt{2}}(\hat{k}^L_{-1} + \hat{k}^L_0)|_{\mathbf{P}} = -\vec{x}^2 \partial_t.
$$

Identifying the coordinates of the flat Carrollian spacetime and the null-plane by $t_{\text{CCFT}} = x^+_{\text{CFT}}$ and $x^i_{\text{CCFT}} = x^i_{\text{CFT}}$, the above generators form the $d$-dimensional CCA. Thus we have matched the Carrollian conformal symmetries with the Lorentzian conformal symmetries preserving the null-plane.

However, there is an extra residual symmetry not in the CCA,

$$
\hat{b}^0 := (\hat{m}^L_{0,-1})|_{\mathbf{P}} = x^+ \partial_+ - 0\partial_-,
\tag{D.4}
$$

and the adjoint action of the related generator $B^0$ on the CCA can be worked out as

$$
\begin{aligned}
[B^0, D] &= [B^0, J^{ij}] = [B^0, P^i] = [B^0, K^i] = 0, \\
[B^0, P^0] &= P^0, \quad [B^0, K^0] = K^0.
\end{aligned}
\tag{D.5}
$$

Recall that the CCA is isomorphic to the Poincare algebra, together with $B^0$ they are isomorphic to $\mathbb{R} \ltimes \mathfrak{iso}(d, 1)$. On the other hand, the automorphism algebra of the Poincare algebra is exactly the same: $\text{Aut}(\mathfrak{iso}(d, 1)) = \mathbb{R} \ltimes \mathfrak{iso}(d, 1)$, hence $B^0$ generates the non-trivial outer automorphism.

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
