# Peer review of "On Higher-dimensional Carrollian and Galilean Conformal Field Theories"

_SciPost Physics, doi:SciPost Phys. 14, 088 (2023)_

## Round 1 · Referee Report · Anonymous (Referee 2) · 2022-10-12

Report

The authors have replied to my comments in a satisfactory way. There is only one point I would like to add:

Regarding point 4 in the attached pdf, the terms electric and magnetic Galilean action that you use are swapped with respect to the definitions in the reference that I mentioned in my report, 2206.12177. Their magnetic Galilean action in equation 7.4 of 2206.12177 corresponds to what you call electric in your file. Moreover, I am not sure if their electric Galilean action in equation 7.9 is not the same as your magnetic one coming from the Hamiltonian action. There is also an extra field in the action playing the role of a Lagrange multiplier.

In the electric case it is clear that the magnetic action involving a Lagrange multiplier, 7.15 in 2206.12177, is the same as 5.3 in 2109.06708. The Lagrange multiplier is simply what was the momentum in the Hamiltonian action before taking the limit. At first glance I don't see this equivalence in the Galilean case. This could be worth to check.

Since the authors will treat electric and magnetic scalar actions in a further publication, I think this new version of their manuscript can be published in SciPost.

---

## Round 1 · Referee Report · Anonymous (Referee 1) · 2022-11-3

Report

In the new version of the paper, the authors attempt to answer several questions that were raised in my earlier report. Some of the errors have been removed and some explanations have been provided. However there are some further comments that need to be made.

• Point 4: This is still the main remaining issue. There is still no proper explanation why SU(n) Young Tableaux appeared. This seems ad hoc and I am not sure one can trust the constructions that follow from this assumption. The authors need to clarify this further.

• Point 7: The statement that the magnetic scalar action gives spatial delta function correlations is wrong. See e.g. 2207.03468, where it is shown that the theory can effectively be reduced to a lower dimensional Euclidean CFT. Since the authors don’t write anything else regarding the magnetic theory, this statement should be removed.

Once these points are addressed, the paper can be considered further.
  • validity: -
  • significance: -
  • originality: -
  • clarity: -
  • formatting: -
  • grammar: -

Author:  Yu-fan Zheng  on 2022-12-04  [id 3101]

(in reply to Report 2 on 2022-11-03)

Dear referee,

Thank you very much for your patience and valuable comments and suggestions. Please see the attachment for our detailed reply, and see the resubmission page for new version of the priprint.

link to resubmission: https://scipost.org/submissions/scipost_202210_00048v2/
link to new priprint: https://scipost.org/preprints/scipost_202210_00048v2/

Best regards,
Bin Chen, Reiko Liu and Yu-fan Zheng

Attachment:

Reply_to_report_1_2_on__2112_10514_.pdf

---

## Round 1 · List of Changes

In this of the preprint, we have added following comments comparing to 2112.10514v2:

  1. Added comments on the representation in the introduction. Stressed that the highest weight representation is not of the form of Wigner classification.

  2. Added comments on Carrollian-BMS symmetry in the introduction, as well as added citations of related papers.

  3. Corrected a statement in section 2.2. The Casimir operators of Galilean conformal algebra can be deduced by taking limit.

  4. Added a foot note on page 20 section 3.2 to explain the convenience using $SU$ rather than $SO$ Young tableau.

  5. Added a comment in section 3.5 about dual relation of Carrollian/Galilean representations is a result of the geometric structure.

  6. Changed the equation (4.5) to be the same with (B.1).

  7. Added on page 33 section 4.1 explicit theories with different correlation functions.

  8. Corrected some typos.

---

## Round 2 · Referee Report · Anonymous (Referee 2) · 2022-12-8

Report

In their previous submission the authors have replied to my comments in a satisfactory way. In my opinion, their new version of their manuscript can be published in SciPost.

---

## Round 2 · Referee Report · Anonymous (Referee 1) · 2022-12-20

Report

The authors have sufficiently addressed my concerns in this version of the paper. I recommend publication.

---

## Round 2 · List of Changes

In this version of our paper, we made revisions mainly in section 3.2. We explained why we used Young tableau and how it corresponds to the tensor representations in this section according to suggestions from the referee. We also modified some expressions in section 3.1 to match expressions in revised section 3.2 which does not change the original statement.

---

## Editorial Decision

published